# Inhibition of calcium-triggered secretion by hydrocarbon-stapled peptides

Ying Lai[1,6 ✉], Giorgio Fois[2], Jose R. Flores[3], Michael J. Tuvim[3], Qiangjun Zhou[1,7], Kailu Yang[1,4], Jeremy Leitz[1], John Peters[1,8], Yunxiang Zhang[1,9], Richard A. Pfuetzner[1,4], Luis Esquivies[1,4], Philip Jones[5], Manfred Frick[2 ✉], Burton F. Dickey[3 ✉] & Axel T. Brunger[1,4 ✉]

Membrane fusion triggered by Ca²⁺ is orchestrated by a conserved set of proteins to mediate synaptic neurotransmitter release, mucin secretion and other regulated exocytic processes[1–4]. For neurotransmitter release, the Ca²⁺ sensitivity is introduced by interactions between the Ca²⁺ sensor synaptotagmin and the SNARE complex[5], and sequence conservation and functional studies suggest that this mechanism is also conserved for mucin secretion[6]. Disruption of Ca²⁺-triggered membrane fusion by a pharmacological agent would have therapeutic value for mucus hypersecretion as it is the major cause of airway obstruction in the pathophysiology of respiratory viral infection, asthma, chronic obstructive pulmonary disease and cystic fibrosis[7–11]. Here we designed a hydrocarbon-stapled peptide that specifically disrupts Ca²⁺-triggered membrane fusion by interfering with the so-called primary interface between the neuronal SNARE complex and the Ca²⁺-binding C2B domain of synaptotagmin-1. In reconstituted systems with these neuronal synaptic proteins or with their airway homologues syntaxin-3, SNAP-23, VAMP8, synaptotagmin-2, along with Munc13-2 and Munc18-2, the stapled peptide strongly suppressed Ca²⁺-triggered fusion at physiological Ca²⁺ concentrations. Conjugation of cell-penetrating peptides to the stapled peptide resulted in efficient delivery into cultured human airway epithelial cells and mouse airway epithelium, where it markedly and specifically reduced stimulated mucin secretion in both systems, and substantially attenuated mucus occlusion of mouse airways. Taken together, peptides that disrupt Ca²⁺-triggered membrane fusion may enable the therapeutic modulation of mucin secretory pathways.

Membrane fusion is an essential step for many biological processes, including enveloped virus–host cell fusion, cell–cell fusion and intracellular vesicle–membrane fusion[12–14]. Fusion proteins supply the energy to overcome the high kinetic barrier to mediate lipid bilayer fusion. The underlying molecular mechanisms are highly similar for both class 1 viral fusion proteins and SNARE-mediated fusion, wherein the formation of helical bundles drives membranes together, leading to membrane fusion. However, for Ca²⁺-triggered membrane fusion, SNARE proteins must cooperate with Ca²⁺ sensors. For example, the neuronal SNARE complex (comprising synaptobrevin-2 (also known as VAMP2), syntaxin-1A (Stx1) and SNAP-25A) forms a specific interface with the Ca²⁺-binding domain C2B of synaptotagmin-1 (Syt1)[5]. This primary interface is conserved in all species and across the other fast isoforms for neurotransmitter release—synaptotagmin-2 (Syt2) and synaptotagmin-9. Many of the key residues that are involved in the primary interface are located in the target-SNARE protein SNAP-25A. Residues involved in the primary interface are critical for Ca²⁺-triggered

fusion in a reconstituted system and in neuronal cultures[5,15]. In the absence of Ca²⁺, Syt1 interacts simultaneously with the *trans*-SNARE (through the primary interface) and the anionic plasma membrane (through the poly-basic region of Syt1 C2B), greatly enhancing the affinity of the primary complex[16]. After Ca²⁺ binding to Syt1, the Syt1–SNARE–membrane assembly changes[5,16], ultimately leading to membrane fusion.

Primary sequence conservation suggests that the primary interface also exists in other systems that use SNARE proteins and synaptotagmins[5]. In particular, mucin exocytosis in airway secretory cells is mediated by SNARE proteins, the Ca²⁺ sensor Syt2 and other factors[1,2,11]. Syt2 is selectively expressed in airway secretory cells compared with ciliated cells, and it serves as a critical sensor for stimulated, but not baseline, mucin secretion[6]. Syntaxin-3 (Stx3) and SNAP-23 are also highly expressed in airway epithelial cells[17–19]. In stimulated mucin secretion, Ca²⁺ is released from the endoplasmic reticulum through the activated inositol triphosphate receptor. Inositol triphosphate is generated by

[1]Department of Molecular and Cellular Physiology, Stanford University, Stanford, CA, USA. [2]Institute of General Physiology, Ulm University, Ulm, Germany. [3]Department of Pulmonary Medicine, University of Texas MD Anderson Cancer Center, Houston, TX, USA. [4]Howard Hughes Medical Institute, Stanford University, Stanford, CA, USA. [5]Institute for Applied Cancer Science, University of Texas MD Anderson Cancer Center, Houston, TX, USA. [6]Present address: National Clinical Research Center for Geriatrics, West China Hospital, State Key Laboratory of Biotherapy and Collaborative Innovation Center of Biotherapy, Sichuan University, Chengdu, China. [7]Present address: Department of Cell & Developmental Biology, Vanderbilt Brain Institute, Center for Structural Biology, Vanderbilt University, TN, USA. [8]Present address: Department of Biological Chemistry and Molecular Pharmacology, Harvard Medical School, Boston, MA, USA. [9]Present address: Department of Chemistry, Fudan University, Shanghai, China. ✉e-mail: ylai@scu.edu.cn; manfred.frick@uni-ulm.de; bdickey@mdanderson.org; brunger@stanford.edu

phospholipase C after binding of agonists (such as ATP or acetylcholine) to hepta-helical receptors in the plasma membrane coupled to $G_q$. The released $Ca^{2+}$ in turn binds to Syt2 on the granule vesicle and then triggers SNARE-mediated fusion of the granule with the plasma membrane, leading to mucin secretion[1,2,6].

Both baseline and stimulated mucin secretion are impaired in *SNAP-23*-heterozygous-mutant mice[18], *Vamp8*-knockout mice[20] and *Unc13b*-knockout (also known as *Munc13-2*) mice[21]. By contrast, stimulated mucin secretion is selectively impaired in *Munc18-2*-mutant (also known as *Stxbp2*) and *Syt2*-mutant mice[6,19,22]. Stx3 binds to and colocalizes with Munc18-2 (refs. [17,19]) and overexpression of *Munc18-2* reduces the level of Stx3–SNAP-23 binary complex[17]. These findings suggested that, similar to neurotransmitter release, SNARE proteins (Stx3, SNAP-23 and VAMP8), Syt2, Munc13-2 and Munc18-2 are among the key components that drive stimulated membrane fusion between mucin-containing granules and the plasma membrane[1,2,11].

## *Syt2* deletion prevents mucus occlusion

Stimulated secretion of mucins that are highly produced in response to inflammation (together, mucus hypersecretion) is a major cause of airway obstruction in the pathophysiology of respiratory viral infection, asthma, chronic obstructive pulmonary disease and cystic fibrosis[7–10,23]. To validate Syt2 as a target of pharmacologic inhibition, we tested whether the deletion of *Syt2* specifically in airway epithelial cells might protect against mucus occlusion. To accomplish this, we crossed mice carrying a floxed *Syt2* allele (*Syt2^F/F*) (ref. [24]) with *Scgb1a-1^Cre*-knockin mice[25,26], as *Syt2*-knockout mice die from complications of ataxia by postnatal day 24 (ref. [27]), precluding the study of the pathophysiologic role of airway mucin secretion in adult *Syt2*-knockout mice. The airway-specific deletant progeny of this cross (*Syt2^D/D*) were born at a Mendelian ratio and appeared to be healthy, and the efficiency of deletion was essentially complete (Extended Data Fig. 1). There was no spontaneous intracellular mucin accumulation in *Syt2^D/D* mice that would indicate impairment of baseline mucin secretion[1,21]. To test impairment of stimulated mucin secretion in these mice, mucous metaplasia was induced by intrapharyngeal instillation of IL-13, then secretion was stimulated with an ATP aerosol. Fractional secretion of intracellular mucin in response to ATP by wild-type (WT) and *Syt2^F/F* mice was 71% and 65%, respectively, whereas it was only 30% in *Syt2^D/D* mice (Fig. 1a, b). To test protection against airway lumenal mucus occlusion, mucous metaplasia was first induced with IL-13, then mucin secretion and bronchoconstriction were stimulated by a methacholine aerosol. The scattered sites of airway mucus occlusion observed in WT and *Syt2^F/F* mice were reduced in *Syt2^D/D* mice, and the cross-sectional area of airway lumenal mucus in a systematic sample of the left lung was reduced in *Syt2^D/D* mice by 74% compared with the WT and by 69% compared with *Syt2^F/F* mice (Fig. 1c, d). Together these data validated Syt2 as a therapeutic target in muco-obstructive airway disease.

## Stapled peptide design

Helical peptides, such as a fragment of SNAP-25A involved in the primary interface (Fig. 2a and Extended Data Fig. 2), could theoretically be used to selectively interfere with this synaptotagmin–SNARE interaction and thereby disrupt the process of $Ca^{2+}$-triggered membrane fusion. From a therapeutic perspective, peptide-based strategies have successfully been applied to inhibit virus–host membrane fusion[28–32]. However, the molecular mechanisms are quite different as viral membrane fusion inhibitors act by interfering with the formation of a six-helix bundle[12], whereas specific $Ca^{2+}$-triggered membrane fusion inhibitors must act in a different manner, for example, by disrupting the interaction between SNARE proteins and the $Ca^{2+}$ sensor synaptotagmin[5,15]. Moreover, peptide-based viral inhibitors function extracellularly whereas peptide-based secretory/synaptic vesicle inhibitors need to act

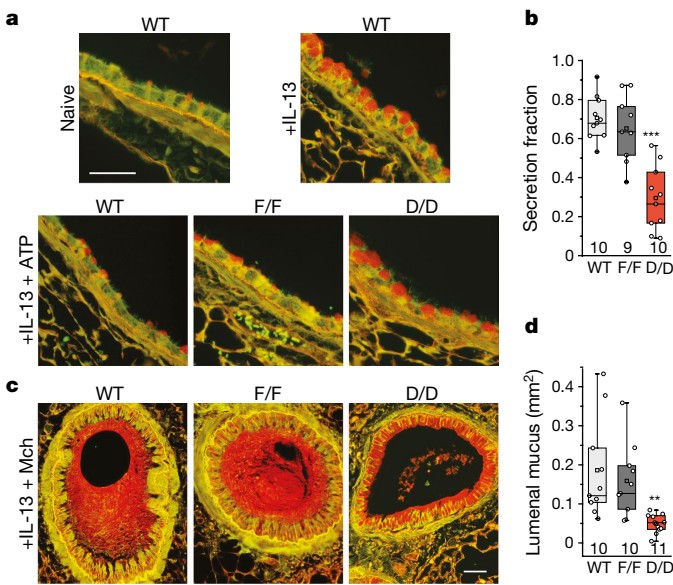

**Fig. 1 | Mucin secretion defects in *Syt2*-mutant mice. a**, Transverse sections of bronchial airways of mice stained with periodic acid fluorescent Schiff (PAFS) to demonstrate mucin with red fluorescence. Top, in naive mice without airway inflammation, scant intracellular mucin is visible. Treatment with IL-13 increases mucin synthesis, resulting in abundant intracellular mucin. Bottom, subsequent treatment with ATP induces mucin secretion, reducing intracellular mucin in WT C57Bl/6J mice (WT) and *Syt2^F/F* mice (F/F), but not in *Syt2^D/D* mice (D/D). Scale bar, 50 μm. **b**, Fractional mucin secretion was measured by analysing images of airways of mice treated with IL-13 alone and comparing those with those of mice treated with IL-13 followed by ATP, as shown in **a**. Individual data points and box plots are shown for two independent sets of experiments combined to give a total *n* mice (indicated below each box plot) per group (Supplementary Table 1). Comparison with the *Syt2^F/F* group of mice was performed using two-tailed unpaired Student's *t*-tests; ***P = 0.00026. **c**, Transverse sections of bronchial airways of mice treated with IL-13, then with methacholine (Mch) to induce smooth muscle contraction and mucin secretion, and fixed with methacarn and stained with PAFS to demonstrate lumenal mucus and residual intracellular mucin. Scale bar, 50 μm. **d**, The sum of the lumenal mucus cross-sectional area in the left lung measured at 500 μm intervals. Individual data points and box plots are shown for two independent sets of experiments combined to give a total of *n* mice (indicated below each box plot) per group (Supplementary Table 1). Comparison with the *Syt2^F/F* group of mice was performed using two-tailed unpaired Student's *t*-tests; **P = 0.0012.

intracellularly. Thus, this research addresses whether a peptide inhibitor strategy can be applied to disrupt $Ca^{2+}$-triggered membrane fusion.

Peptides typically have little secondary structure in solution when taken out of the context of the intact system. Thus, their efficacy as in vivo reagents may be limited by their loss of secondary structure. Considering the successful use of stapled peptides[33] to inhibit HIV virus infection[34] and for p53-dependent cancer therapy[35], we also used non-natural amino acids containing olefin-bearing groups to generate hydrocarbon-stapled peptides by a ring closing metathesis reaction using Grubbs catalyst[36] to interfere with the primary interface. The residues that are at or near to the primary interface are identical for Syt1 and Syt2 except for V292C, and identical for SNAP-25A and SNAP-23 except for K40Q, L47I and V48T (Extended Data Fig. 2a, b). As the crystal structure is known for the neuronal system, we first used it for the design of stapled peptides that disrupt this interaction, and subsequently tested the peptides in the airway epithelial system. We examined a series of stapled peptides[37] and ultimately designed a hydrocarbon-stapled peptide consisting of a SNAP-25A fragment that included many of the key residues involved in the primary interface (named SP9; Fig. 2a–c and Extended Data Fig. 2a–c). It will be desirable to

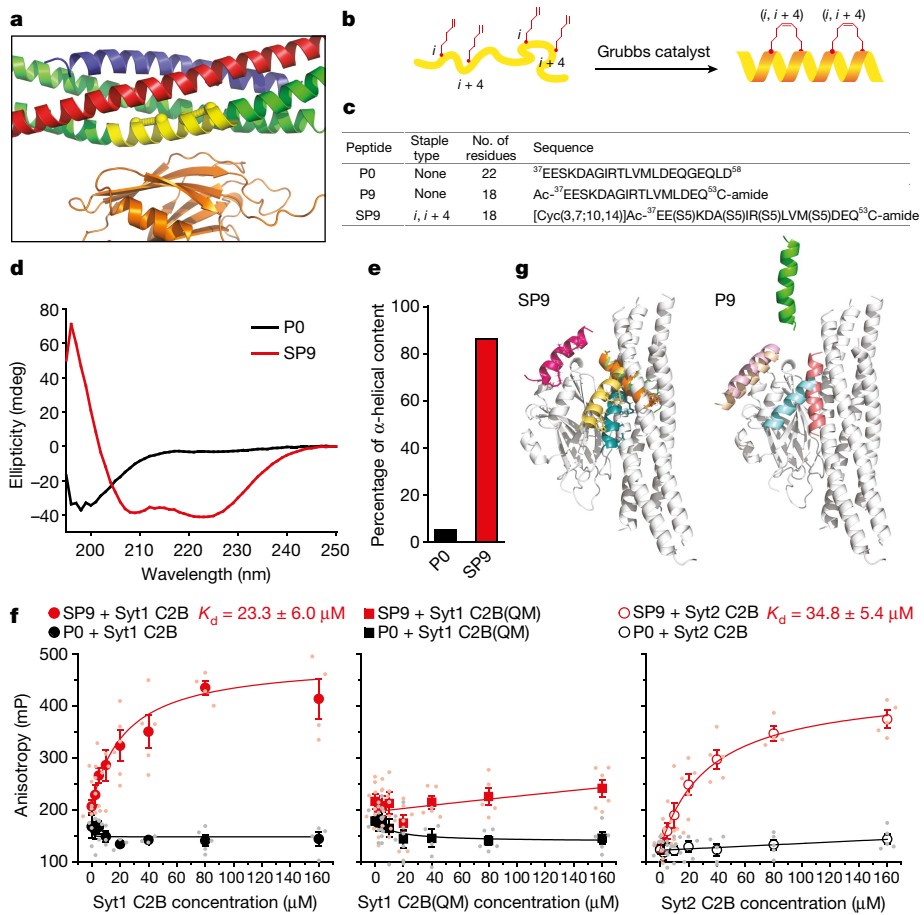

**Fig. 2 | Characterization of SP9. a**, Magnified view of the primary interface between the neuronal SNARE complex (VAMP-2 (blue), Stx1 (red) and SNAP-25A (green)) and the C2B domain of Syt1 (orange) (Protein Data Bank (PDB): 5W5C), indicating the region (yellow) that corresponds to the stapled peptide SP9 with staples shown as dumbbells. **b**, Schematic of the synthesis of SP9. Hydrocarbon-stapled peptides are formed by cross-linking residues at the specified positions. **c**, Sequences of peptides. *S*5 indicates *S* stereochemistry at the α-carbon, with 5 carbon atoms in the olefinic side chains. The superscripts denote the start and end positions of the SNAP-25A sequence. **d**, Circular dichroism (CD) spectra of 100 mM peptides measured at pH 7.4 and at 25 ± 1 °C. **e**, The percentage of α-helical content in these peptides was estimated by dividing the mean residue ellipticity $[\varphi]222_{obs}$ by the reported $[\varphi]222_{obs}$ for a model helical decapeptide. **f**, Interactions between Cy3-labelled SP9 or P0 and unlabelled Syt1 C2B, the quintuple Syt1 C2B(QM) mutant and Syt2 C2B as measured by bulk fluorescence anisotropy (Methods). Data are mean ± s.e.m. along with individual data points from $n$ = 3–7 independent experiments. Hill equations were fit to estimate the dissociation constant $K_d$, where the Hill coefficients were constrained to 1. **g**, Peptide conformations (colours) after five independent 1 μs molecular dynamics simulations of SP9–Syt1 C2B (left) and P9–Syt1 C2B (right) superimposed onto the structure of the primary interface (grey). The simulations started from a conformation (Extended Data Fig. 2f, g) that was derived from the crystal structure PDB 5W5C (Supplementary Videos 1 and 2). For one simulation of P9–Syt1 C2B, the P9 peptide dissociated around 168 ns.

test other sequences and strategies in future work, including the SNAP-23 sequence itself, that might strengthen (or weaken) the interaction with Syt1 or Syt2 considering the K40Q, L47I and V48T substitutions (Extended Data Fig. 2a). For SP9, four non-natural amino acids were incorporated into the sequence (Methods). Two hydrocarbon staples were made to flank three (substitution positions $i$ and $i + 4$) amino acids within the SNAP-25A fragment (Fig. 2b, c). The positions of these substitutions were chosen to be away from the primary interface (Fig. 2a). As a control, we also used a non-stapled SNAP-25A fragment, named P0, that displays only 5% helicity in solution, indicating that P0 is largely a random coil (Fig. 2d, e). By contrast, SP9 has substantial helical content (Fig. 2d, e) and it does not aggregate, as assessed using size-exclusion chromatography (Extended Data Fig. 3a). Single-molecule counting experiments with surface-immobilized SP9–Cy3 suggest that it is primarily monomeric with minor fractions of higher-order oligomers (Extended Data Fig. 3b–d).

To determine whether SP9 specifically interacts with the C2B domains of Syt1 and Syt2, we labelled it with the fluorescent dye Cy3 at the C terminus. We recorded bulk fluorescence anisotropy after

mixing Cy3 labelled SP9 with varying concentrations of Syt1 C2B, Syt2 C2B or a quintuple mutant of Syt1 C2B (C2B(QM)) (Fig. 2f and Extended Data Fig. 2c). SP9 binds to both Syt1 C2B and Syt2 C2B with a similar dissociation constant ($K_d$) of 24 μM and 35 μM, respectively, which is comparable to the $K_d$ between Syt1 and the SNARE complex (~20 μM) (ref. [5]). As a control and as expected, SP9 does not bind to the quintuple mutant of Syt1 C2B(QM) as that mutation disrupts the primary interface[5]. Moreover, we did not observe binding of the non-stapled peptide P0 to the C2B domain of either Syt1 or Syt2 in the conditions of these experiments. Taken together, SP9 binds specifically to the C2B domain of Syt1 or Syt2.

These binding experiments suggest that the stabilization of SP9 peptide by staples is important for binding. To corroborate this finding, we assessed the stability of the SP9 peptide interactions with the C2B domain of Syt1 using molecular dynamics simulations starting with a conformation derived from the crystal structure of the primary complex (Extended Data Fig. 2f, g). Four out of the five simulations of SP9 adopt binding poses at the end of the 1 μs simulations that would interfere with the formation of the primary interface (Fig. 2g (left) and Supplementary

Video 1). By contrast, 1 µs simulations of this peptide without staples (referred to as P9) produced only one binding pose that would interfere with primary complex formation (Fig. 2g (right) and Supplementary Video 2). These simulations show that the interaction between SP9 and Syt1 is more dynamic at the N-terminal end of SP9, suggesting that there are opportunities for strengthening the interaction in the future.

## SP9 inhibits Ca²⁺-triggered vesicle fusion

Next, we tested whether SP9 specifically disrupts membrane fusion with reconstituted neuronal SNARE proteins and Syt1. We first tested the effect of SP9 in a single-vesicle content mixing assay with reconstituted neuronal SNARE proteins and Syt1 (Extended Data Fig. 4a). SP9 had no effect on vesicle association (Extended Data Fig. 4b), but reduced both $Ca^{2+}$-triggered fusion and $Ca^{2+}$-independent fusion (Extended Data Fig. 4c–h). As a control and to test specificity, when Syt1 was left out (that is, vesicles with VAMP2 only, referred to as VAMP2 vesicles), or replaced by the quintuple mutant of Syt1 (Syt1(QM)) that disrupts binding to the SNARE complex[5], SP9 had little effect on either $Ca^{2+}$-independent or $Ca^{2+}$-triggered fusion using the single-vesicle content mixing assay (Extended Data Fig. 5). The specific inhibitory effect of SP9 observed in this assay is probably due to peptide binding to Syt1, in competition with the primary complex.

To test whether SP9 has a specific inhibitory effect on the process of mucin secretion, we first performed a 'simple' reconstitution with two types of vesicles to mimic mucin secretion: we used vesicles with reconstituted Stx3 and SNAP-23 that mimic the plasma membrane of epithelial cells (airway PM vesicles), and vesicles with reconstituted VAMP8 and Syt2 that mimic mucin-containing secretory granules (SG vesicles) (Methods) (Extended Data Figs. 6 and 7a). Inclusion of 10 µM of the P0 unstapled control peptide in the fusion assay had no effect on the intrinsic $Ca^{2+}$-independent fusion probability or $Ca^{2+}$-triggered cumulative fusion probability (Extended Data Fig. 7c–h). By contrast, 10 µM SP9 reduced both $Ca^{2+}$-triggered and, to a lesser degree, $Ca^{2+}$-independent fusion (Extended Data Fig. 7c–h). As with the neuronal system, SP9 had no effect on vesicle association (Extended Data Fig. 7b). As a control and to test specificity, when Syt2 was left out (that is, vesicles with VAMP8 only, referred to as VAMP8 vesicles), this inhibitory effect of SP9 was eliminated (Extended Data Fig. 7i–k).

As Munc13 catalyses the transition of syntaxin from the syntaxin–Munc18 complex into the ternary SNARE complex[38–40] and promotes proper SNARE complex formation[41], we next tested the effect of the SP9 stapled peptide in a more complete reconstitution that includes airway epithelial SNAREs, Syt2, the C1C2B_MUN2 fragment of Munc13-2 (referred to as Munc13-2*) (ref. [21]) and Munc18-2 (refs. [19,22]) (Fig. 3a). Following previous research with neuronal proteins[41], to form the Stx3–Munc18-2 complex (that is, SM vesicles), we first added the 'disassembly factors' (NSF, αSNAP, ATP and Mg²⁺) along with Munc18-2 to tethered airway PM vesicles. We next added Munc13-2* and SNAP-23 along with SG vesicles in the flow chamber above the tethered SM vesicles (Fig. 3b).

As a control, in the absence of Munc13-2*, essentially neither SG $Ca^{2+}$-dependent nor $Ca^{2+}$-independent fusion events were observed because the Stx3–Munc18-2 complex is in the closed conformation and, therefore, ternary SNARE complex formation cannot occur[42] (Supplementary Table 2). By contrast, in the presence of Munc13-2*, robust $Ca^{2+}$-triggered fusion was observed at $Ca^{2+}$ concentrations of both 50 µM and 500 µM (Fig. 3f–i). When comparing this more complete reconstituted fusion assay with the simple reconstitution (SR) assay that uses only airway SNAREs and Syt2, the $Ca^{2+}$-independent fusion probability is similar (Fig. 3d, e), but the 500 µM $Ca^{2+}$-triggered fusion amplitude and the cumulative fusion probability are significantly larger (Fig. 3g, h). Taken together, the more complete reconstitution that includes Syt2, Munc13-2*, Munc18-2, NSF and αSNAP improved $Ca^{2+}$-triggered fusion by an order of magnitude compared with the SR

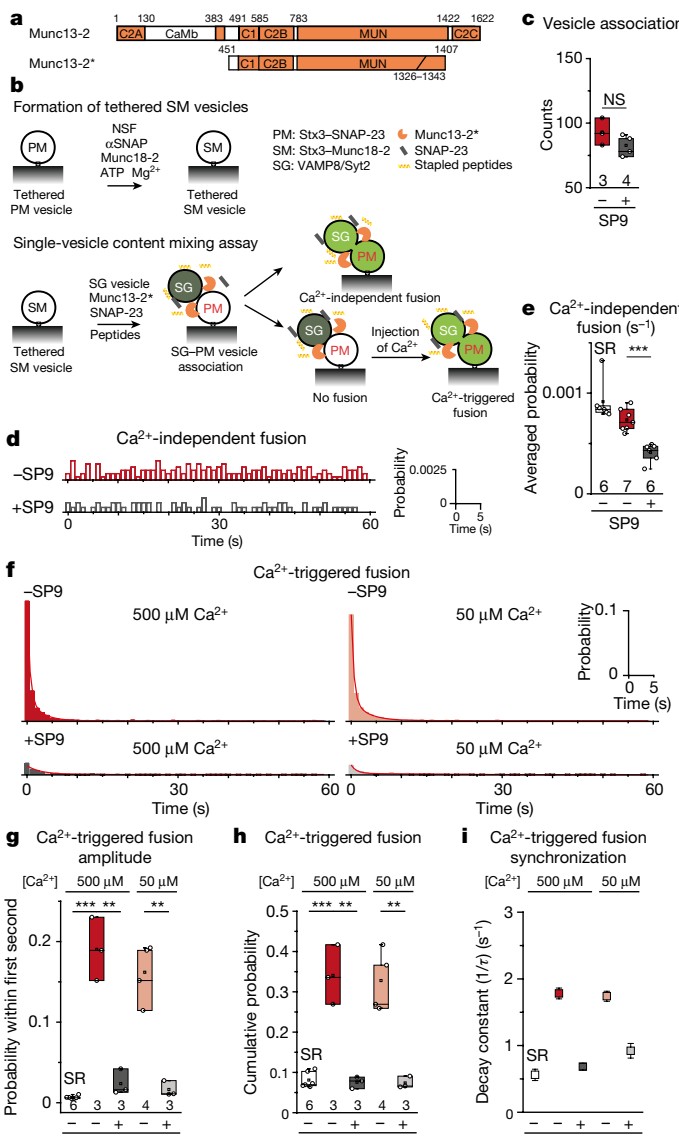

**Fig. 3 | SP9 inhibits triggered fusion in a reconstituted system. a**, The domain structure of Munc13-2 and its fragment (Munc13-2*). **b**, Single-vesicle content mixing assay with complete reconstitution (Methods). Stapled peptide (10 µM) was added together with SG vesicles and was present during all of the subsequent stages. **c**, The effect of SP9 on vesicle association. **d**, Corresponding $Ca^{2+}$-independent fusion probabilities. **e**, Corresponding average probabilities of $Ca^{2+}$-independent fusion events per second. For comparison, the result for the SR assay (Extended Data Fig. 7) is also shown. ***$P = 0.00016$. **f**, Corresponding $Ca^{2+}$-triggered fusion probabilities at 500 µM and 50 µM $Ca^{2+}$. **g–i**, Corresponding $Ca^{2+}$-triggered fusion amplitudes of the first 1 s time bin after injection with 500 µM $Ca^{2+}$ (**g**) (from left to right, ***$P = 0.0000053$, **$P = 0.0024$, **$P = 0.0012$); the cumulative $Ca^{2+}$-triggered fusion probability within 1 min (**h**) (from left to right, ***$P = 0.000058$, **$P = 0.0037$, **$P = 0.0026$); and the decay rate ($1/\tau$) of the $Ca^{2+}$-triggered fusion histogram (**i**). The fusion probabilities and amplitudes were normalized to the number of analysed SG–airway PM vesicle pairs (Supplementary Table 2). For comparison, the results for the SR assay (Extended Data Fig. 7) are also shown. For **c**, **e**, **g** and **h**, box plots and data points are shown for $n$ (indicated below each box plot) independent repeat experiments (Supplementary Table 2). For **c**, **e**, **g** and **h**, statistical analysis was performed using two-tailed Student's $t$-tests. Decay constants (boxes) and error estimates (bars) in **i** were computed from the covariance matrix after fitting the corresponding histograms combining all repeats with a single exponential decay function using the Levenberg–Marquardt algorithm.

reconstitution with just epithelial airway SNARE proteins and Syt2. Moreover, the $Ca^{2+}$ sensitivity of the more complete reconstitution is closer to the physiological range[43,44].

When 10 µM SP9 was added in the more complete reconstitution assay, the 50 µM and 500 µM $Ca^{2+}$-triggered fusion amplitude, the cumulative fusion probability and the synchronization were strongly inhibited (Fig. 3g–i). By contrast, the $Ca^{2+}$-independent fusion probability (Fig. 3e) was only moderately reduced compared with the $Ca^{2+}$-triggered amplitude and cumulative fusion probabilities. Moreover, SP9 had no effect on vesicle association (Fig. 3c). Taken together, SP9 specifically inhibits $Ca^{2+}$-triggered membrane fusion in our reconstituted system that includes epithelial airway SNARE proteins, Syt2, Munc13-2*, Munc18-2, NSF and αSNAP. From a mechanistic perspective, our results further solidify the critical and active role of the conserved primary (synaptotagmin–SNARE) interface for $Ca^{2+}$-triggered membrane fusion.

## SP9 inhibits stimulated mucin secretion in cells

We next examined whether the selected stapled peptides could also inhibit mucin secretion in primary human airway epithelial (HAE) cells. To facilitate cellular entry of the stapled peptides, we conjugated the N terminus of SP9 with cell-penetrating peptides (CPPs) (ref. [45]), and the C terminus with Cy3 fluorescent dye for assessing cellular entry (Fig. 4a). Moreover, we conjugated SP9–Cy3 with biotin and bound it to streptavidin-conjugated bacterial toxins (non-toxic mutants of clostridial C2 toxin or diphtheria toxin (CRM197)) as a possible alternative for intracellular delivery. These toxins can deliver biotin-conjugated peptides into mammalian cells through endocytosis[46,47].

Confocal imaging of fixed HAE cells treated with SP9–Cy3, conjugated to either bacterial toxins or CPPs, indicated that only CPP-modified SP9–Cy3 penetrated into the cell interior (Fig. 4b–d and Extended Data Fig. 8a). Analysis of the intracellular localization in MUC5AC-positive airway secretory cells (MUC5AC+ cells) confirmed that CPP-modified SP9–Cy3 was delivered into the cytoplasm of secretory cells, whereas biotin–SP9–Cy3 bound to streptavidin-conjugated bacterial toxins was mainly localized to apical mucin granules (Extended Data Fig. 8b). Quantification of cumulative, intracellular Cy3 fluorescence intensities indicated that both penetratin (PEN) (ref. [48]) and TAT[49] efficiently delivered SP9–Cy3 into MUC5AC+ cells (Fig. 4b–d). Intact epithelial morphology was observed under all of the conditions (Fig. 4c and Extended Data Fig. 8a). We therefore used SP9–Cy3 conjugated to these CPPs for all of the subsequent experiments.

To investigate the effect of CPP-conjugated SP9–Cy3 on baseline and stimulated secretion under control and mucous metaplastic conditions, we cultured HAE cells in the absence or presence of 10 ng ml$^{-1}$ IL-13, respectively. IL-13 treatment induces goblet cell hyperplasia and metaplasia in vitro[50,51], mimicking IL-13-induced mucous metaplasia in vivo[52,53]. Consistently, MUC5AC expression was upregulated in IL-13 treated cells (Extended Data Fig. 8c). We next performed experiments under metaplastic conditions with a 30 min peptide pre-incubation at a peptide concentration of 10 µM (Fig. 4e). Baseline secretion was low under metaplastic conditions and was not affected by peptide treatment (Fig. 4f, g). Cell morphology also appeared normal. Short-term treatment with PEN–SP9–Cy3 and TAT–SP9–Cy3 substantially reduced stimulated (that is, IL-13 + ATP) MUC5AC secretion by 48% and 86%, respectively (Fig. 4f, h). The unconjugated SP9–Cy3 peptide had no effect on stimulated secretion. We also tested the non-stapled P9–Cy3 peptides (Fig. 4a, f, h): 10 µM PEN–P9–Cy3 or TAT–P9–Cy3 had no effect on stimulated secretion, consistent with a specific action of the stapled SP9 on stimulated mucin secretion.

As the inhibitory effect of TAT–SP9–Cy3 was statistically significant, whereas PEN–SP9–Cy3 was just below being statistically significant, we next tested the effect of the CPP-conjugated SP9–Cy3 peptides at a higher concentration and longer duration. We incubated HAE cells

with 100 µM CPP-conjugated SP9–Cy3 for 24 h before stimulation (Extended Data Fig. 8c). Similar to the experiments at a lower peptide concentration, both PEN–SP9–Cy3 and TAT–SP9–Cy3 significantly reduced stimulated MUC5AC secretion by 73% and 83% in metaplastic HAE cells, respectively (Extended Data Fig. 8c, d). SP9–Cy3 (without CPP) and non-stapled P9–Cy3 peptides had no effect on ATP-stimulated secretion (Extended Data Fig. 8c, d). Taken together, these results suggest that CPP-conjugated SP9–Cy3 inhibits agonist-stimulated mucin secretion from secretory airway epithelial cells.

## SP9 inhibits stimulated mucin secretion in mice

To investigate whether SP9 could have a therapeutic benefit in vivo, we introduced SP9 conjugated to CPPs and fluorophores into mouse airways using a microsprayer inserted into the distal trachea under direct visualization with a laryngoscope. Initial pilot experiments using TAT–SP9–Cy3 and PEN–SP9–Cy3 showed that the forcefully injected peptide solutions mostly bypassed the left proximal axial bronchus between lateral branches L1 and L2, although labelled peptides entered epithelial cells in the distal bronchus and alveoli (Extended Data Fig. 9a). All of the subsequent studies of airway mucin secretion were therefore performed in the distal axial bronchus. Unexpectedly, when TAT–SP9–Cy3 was introduced into the airways of mice with mucous metaplasia, serial sections showed that Cy3-labelled cells had secreted their mucin stores without stimulation by a secretagogue such as ATP or methacholine (Extended Data Fig. 9b, c). This effect was not observed with the injection of buffer alone. Thus, secretion was not induced by the shear force of the microsprayer, but it appears to be a side effect of the TAT–SP9–Cy3 compound in this system. Fortunately, PEN–SP9–Cy3 did not show this problem (Fig. 5a, b), so it was used in all of the subsequent experiments. Exploratory dose-ranging experiments showed minimal Cy3 labelling below 20 µM PEN–SP9–Cy3, and an apparent plateau above 200 µM, so a peptide concentration of 200 µM in the microsprayer delivered 30 min before secretagogues was used in all subsequent experiments. Notably, labelling of ciliated cells with PEN–SP9–Cy3 was greater than that of secretory cells (Extended Data Fig. 9d, e), perhaps reflecting the greater apical surface area of ciliated cells.

Aerosol administration of 200 µM (microsprayer concentration) PEN–SP9–Cy3 labelled 76% of epithelial cells in the distal left axial bronchus, whereas 200 µM of the control peptide PEN–P9–Cy3 labelled 77% of epithelial cells (Fig. 5a, b). Pretreatment with PEN–SP9–Cy3 markedly reduced the fractional secretion of intracellular mucin stimulated by methacholine (by 82.3%), whereas PEN–P9–Cy3 had no effect (Fig. 5c, d), suggesting that the effect of PEN–SP9–Cy3 is specific. Pretreatment with PEN–SP9–Cy3 also significantly reduced airway lumenal mucus accumulation in the lungs (by 33.1%), whereas PEN–P9–Cy3 had no effect (Fig. 5e, f).

## Discussion

The primary interface between Syt1 and the SNARE complex is essential for fast synchronous $Ca^{2+}$-triggered neurotransmitter release[5]. Here we designed a hydrocarbon-stapled peptide, SP9, based on a fragment of SNAP-25A that participates in the interface as observed in the crystal structure of the neuronal SNARE–Syt1 complex, and used this both to examine Syt1-dependent stimulated neurotransmitter release and, more generally, to analyse cognate interactions of Syt isoforms with non-neuronal SNARE complexes.

The hydrocarbon staples promoted an α-helical conformation of SP9 and greatly enhanced the interaction with Syt1 (Fig. 2d, f). SP9 specifically inhibited $Ca^{2+}$-triggered single-vesicle fusion with reconstituted neuronal SNAREs and Syt1 (Extended Data Fig. 4). We next tested SP9 with a reconstituted simple single-vesicle fusion assay using airway SNAREs and Syt2, and found that SP9 also specifically inhibited $Ca^{2+}$-triggered single-vesicle fusion for the airway system

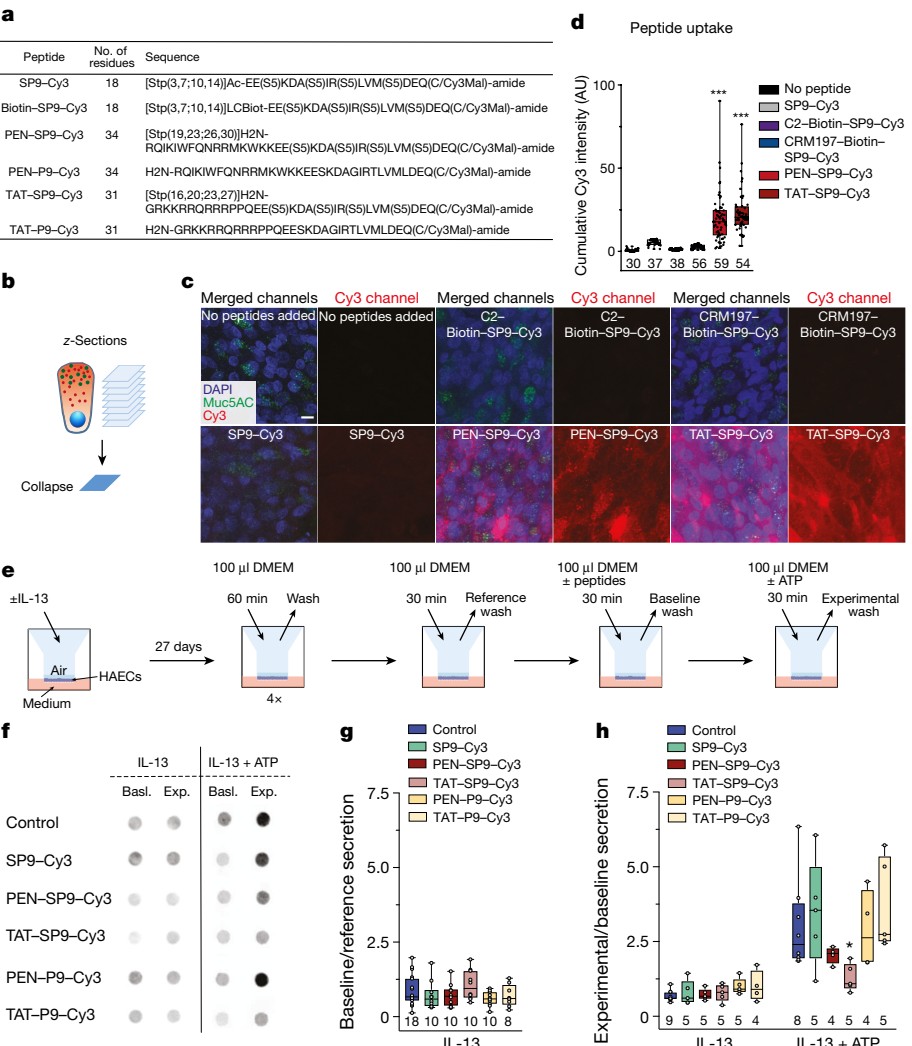

**Fig. 4 | SP9 inhibits mucin secretion from airway epithelial cells. a**, The design of synthesized SP9 with biotin or CPPs conjugated to the N terminus and Cy3 to the C terminus, respectively. Biotin–SP9–Cy3 was bound to streptavidin-conjugated C2 or CRM197. **b**, Diagram of the analysis of cumulated Cy3 intensities within individual HAE cells. **c**, Confocal collapse (projected) images of fixed HAE cells that were treated with SP9–Cy3 or SP9–Cy3 conjugated to bacterial toxins (C2, CRM197) or CPPs. Scale bar, 10 μm. The experiment was independently repeated twice with air–liquid interface (ALI) cultures from different donors with similar results. **d**, Quantitative analysis of intracellular Cy3 fluorescence for each peptide in MUC5AC⁺ HAE cells. Box plots and data points are shown for *n* cells (indicated below each box plot). Statistical analysis was performed using one-way analysis of variance (ANOVA) followed by post hoc Dunnett's test; ***P < 0.0001. **e**, Schematic of the peptide application and sample collection from HAE cells maintained under ALI conditions. **f**, Representative western blot immunofluorescence images for

MUC5AC on an apical surface of untreated HAE cells (control), HAE cells treated with 10 μM of SP9–Cy3, or 10 μM of either SP9–Cy3 or P9–Cy3 conjugated to CPPs for 30 min before stimulation. Basl., MUC5AC secretion during a 30 min period before stimulation (baseline). Exp., MUC5AC secreted within a 30 min experimental period with (IL-13 + ATP) or without (IL-13) stimulation of HAE cells with 100 μM ATP. Cells were treated with IL-13 to induce mucous metaplasia. All of the original blots are shown in Supplementary Fig. 1a. **g**, **h**, The ratio of baseline to reference wash secretion (fold increase in baseline secretion over reference secretion) (**g**) and the ratio of experimental to baseline (fold increase in stimulated secretion over baseline secretion) (**h**) for each condition in **f**. The numbers below the box plots indicate *n* for each condition, representing individual ALI cultures derived from four donors for each condition. Statistical analysis was performed using two-way ANOVA followed by post hoc Dunnett's test; *P = 0.013.

(Extended Data Fig. 7). Moreover, we observed even more pronounced inhibition of Ca²⁺-triggered vesicle fusion when both Munc13-2* and Munc18-2 were included in a more complete reconstitution with airway SNAREs and Syt2, resulting in around an eight to tenfold decrease in the Ca²⁺-triggered fusion amplitude (Fig. 3g). By contrast, the effect of SP9 on Ca²⁺-independent fusion was relatively modest (Fig. 3e). The observed inhibition of Ca²⁺-triggered fusion by SP9 is probably caused by SP9 binding to Syt2 in competition with the primary interface (Fig. 2g).

We next examined whether SP9 can be delivered into HAE cells. Peptides were applied to the apical side of the reconstituted airway epithelia, mimicking intratracheal aerosol delivery in vivo, a significant

advantage to minimize systemic off-target effects. Conjugation of SP9 with either PEN or TAT CPPs resulted in substantial peptide uptake into the cytoplasm of secretory airway cells (Fig. 4b–d). Treatment with 10 μM CPP-conjugated TAT–SP9–Cy3 or PEN–SP9–Cy3 reduced ATP-stimulated, Ca²⁺-triggered MUC5AC secretion (Fig. 4e, f, h and Extended Data Fig. 8c, d), whereas the non-stapled CPP-conjugated P9–Cy3 peptide did not exhibit inhibitory effects at this concentration, suggesting that the inhibitory action of the SP9 compound is specific. Importantly, baseline secretion was not affected by any of the peptides in the IL-13 (metaplastic) cultures (Fig. 4g), consistent with the absence of an effect of *Syt2* deletion on baseline secretion in mice[6].

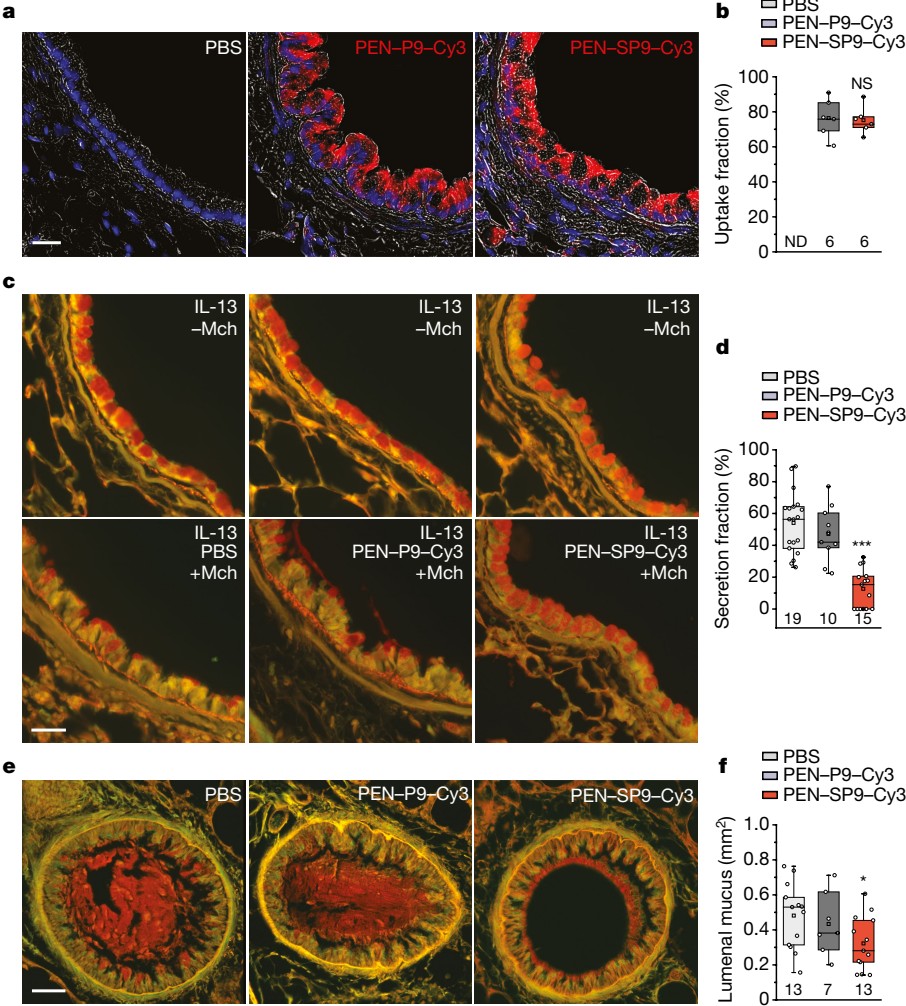

**Fig. 5 | SP9 inhibits mucin secretion and mucus occlusion in mice.**
**a**, Transverse sections of mouse bronchial airways showing intracellular uptake of peptides (200 μM microsprayer concentration; Cy3 is visualized in red). Nuclei fluoresce blue (DAPI), overlaid on bright-field images. Scale bar, 25 μm. **b**, Averaged uptake fractions for two independent experiments for a total *n* cells (indicated below each box plot) per group (Supplementary Table 1). Total of *n* = 260 (PEN–SP9–Cy3) and *n* = 361 (PEN–P9–Cy3) cells (6 sections from 3 mice per peptide). Statistical analysis was performed using a two-tailed unpaired Student's *t*-test, showing a non-significant difference (*P* = 0.83). ND, not detected. **c**, Fractional mucin secretion was measured by analysis of images of mouse airways treated with IL-13 to induce mucous metaplasia (top row), followed by stimulation of secretion with methacholine or PBS as control (bottom row). The sections were taken 3 mm distal to those in Fig. 1. Mice that were pretreated with PBS or PEN–P9–Cy3 show greater reductions in intracellular mucin content (PAFS stain (red)) in response to methacholine compared with mice that were pretreated with PEN–SP9–Cy3. Scale bar, 25 μm. **d**, Fractional mucin secretion measured as in **c**. Box plots and data points are shown for two independent experiments for a total *n* mice (indicated below each box plot) per group. Statistical analysis was performed using a two-tailed unpaired Student's *t*-test, showing a significant difference between mice that were pretreated with PEN–SP9–Cy3 compared with those that were pretreated with PBS; ***P* = 0.00000001. **e**, Airway lumenal mucus was measured by image analysis as in Fig. 1, except that the right lungs were examined instead of the left lungs. Scale bar, 25 μm. **f**, The sum of lumenal mucus cross-sectional area in the caudal lobe of the right lung measured at 500 μm intervals. Box plots and data points are shown for two independent experiments combined for a total *n* mice (indicated below each box plot) per group. Statistical analysis was performed using a two-tailed unpaired Student's *t*-test, showing a significant difference between mice that were pretreated with PEN–SP9–Cy3 compared with those that were pretreated with PBS; **P* = 0.027.

Finally, we tested the efficacy of CPP-conjugated SP9 in mice. Short-term treatment of mice with aerosolized PEN–SP9–Cy3 resulted in substantial peptide uptake into distal airway epithelial cells, and reduced methacholine-stimulated, Ca²⁺-triggered mucin secretion and airway mucus occlusion (Fig. 5), whereas the non-stapled PEN–P9–Cy3 peptide did not exhibit an inhibitory effect, again suggesting specificity. Note that the inhibitory effects of PEN–SP9–Cy3 in mice are probably greater than its measured effects on secretion and occlusion due to peptide uptake into only 76% of epithelial cells (Fig. 5a) and preferential uptake into ciliated over secretory cells (Extended Data Fig. 9d, e). Furthermore, the higher concentration of peptides required in mice compared with in cultured cells probably reflects the brief contact of the forcefully injected aerosol in mice with airway epithelial cells (Extended Data Fig. 9a).

In summary, CPP-conjugated stapled peptides can be efficiently delivered into cultured human epithelial cells and airway epithelial cells of mice, in which they markedly and specifically reduce stimulated mucin secretion and mucus occlusion of mouse airways and they could therefore serve as a starting-point therapeutic without incurring toxicity due to reduced baseline mucin secretion. Optimization of such a therapeutic would include maximizing cellular uptake without causing transepithelial systemic delivery, improving intracellular stability and increasing potency. An optimized drug could be used both as a single-dose therapy in

an acute exacerbation of airway disease (asthma, chronic obstructive pulmonary disease and cystic fibrosis), and as a drug delivered repeatedly in a patient in whom control of mucus hypersecretion is difficult to achieve with drugs directed at upstream inflammatory mediators. Therapeutic peptides derived from other host proteins have shown a very low rate of immunogenicity in clinical trials when administered systemically[33,54,55], as has recombinant human DNase administered repeatedly by aerosol[56], so this is unlikely to be a limitation of chronic therapy, although it would need to be examined. Taken together, we have shown that stimulated membrane fusion processes, such as neurotransmitter release or mucin secretion, can be manipulated pharmacologically by compounds that disrupt the interaction between the fusion proteins and $Ca^{2+}$ sensors. In view of the broad physiological significance of calcium-regulated exocytosis in neurological, endocrine and exocrine function, our research paves the way for the development of therapeutics.

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

# Methods

## Mice

All of the experiments were approved by the Institutional Animal Care and Use Committee of MD Anderson Cancer Center. *Syt2* conditionally deleted mice with the second exon flanked by *loxP* recombination sites (floxed, *Syt2^F/F^*) were obtained from T. C. Südhof[24]. These were generated on a mixed 129/Sv:C57BL/6 background and backcrossed by us for ten generations onto a C57BL/6J background. To delete *Syt2* in airway epithelial cells, *Syt2^F/F^* mice were crossed with mice in which a Cre recombinase optimized for mammalian codon usage was knocked into the secretoglobin 1A1 locus (*Scgb1a1^Cre^*) (refs. [25,26]). Half of the progeny that resulted from crossing *Syt2^F/F^* mice with *Syt2^F/F^* mice that were also heterozygous for the *Scgb1a1^Cre^* allele were *Syt2* deletant (*Syt2^D/D^*) mice, and the other half were *Syt2^F/F^* mice that served as littermate controls for the mucin-secretion experiments. Genotyping was performed by PCR using the oligonucleotide primers for *Syt2* WT and mutant alleles described in ref. [24]. Deletion of *Syt2* in airway epithelial cells was confirmed by immunohistochemical staining (Extended Data Fig. 1) using primary rabbit polyclonal antibodies against Syt2 (Abcam, ab113545, 1:1,000) and secondary donkey anti-rabbit IgG polyclonal antibodies conjugated to horseradish peroxidase (Jackson ImmunoResearch, 711-005-152, 1:5,000). Peroxidase activity was localized using a diaminobenzidine substrate kit (Vector Laboratories, SK-4100), and the slides were counterstained with haematoxylin. C57BL/6J mice were purchased from the Jackson Laboratory and used as controls to be certain the *Syt2^F/F^* allele was not hypomorphic in airway epithelium. As *Syt2^F/F^* mice did not differ from WT *Syt2* mice at the baseline or in the degree of mucous metaplasia and efficiency of stimulated secretion (Fig. 1), they were used as the primary comparator for *Syt2^D/D^* mice to minimize environmental and off-target genetic differences. Mice of both sexes were used aged 6–26 weeks. The animals were housed in specific pathogen free conditions under a 12 h–12 h light–dark cycle with food and water ad libitum. Group sizes were calculated to detect between-group differences with a power of 90% and two-tailed significance of 5%, based upon effect sizes from our numerous prior studies of mucin secretion and lumenal mucus accumulation. The number of animals used was the minimum that is consistent with scientific integrity and regulatory acceptability, with consideration given to the welfare of individual animals in terms of the number and extent of procedures to be carried out on each animal. Mice from the appropriate genotypes were randomly assigned to groups for all of the conditions. Investigators were blinded to the mouse group allocation during data collection and analysis. Furthermore, mouse airway images were analysed by investigators who were blinded to the genotype and treatment of the animals.

## Mucin secretion and airway mucus occlusion in mice

The efficiency of stimulated mucin secretion and the extent of mucus accumulation in the airway lumen of *Syt2*-mutant mice were measured as described previously[22]. In brief, to increase intracellular mucin content (that is, induce mucous metaplasia), 3 μg IL-13 (BioLegend) in 40 μl PBS was instilled every other day for a total of 3 times into the posterior pharynx of mice under isoflurane anesthesia to be aspirated during inhalation. Three days after the last instillation, mucin secretion was stimulated by exposing mice for 10 min to an aerosol of 100 mM ATP in 0.9% NaCl, then the lungs were collected 20 min later. Transverse sections of bronchial airways of mice were stained with PAFS to demonstrate mucin with red fluorescence. Fractional mucin secretion was calculated as the percentage reduction in intracellular mucin content (see below for quantification) of individual mice after sequential treatment with IL-13 and ATP or methacholine compared with the group mean mucin content of mice of the same genotype treated with only IL-13. To measure intracellular airway epithelial mucin content, the lungs were inflated through the trachea with 10% neutral buffered

formalin to 20 cm water pressure for 24 h at 4 °C, then embedded in paraffin. A single transverse 5 μm section was taken through the axial bronchus of the left lung between lateral branches 1 and 2, deparaffinized, rehydrated and stained with PAFS reagent.

Images were acquired using an upright microscope (Olympus BX 60) with a ×40 objective lens (NA 0.75), and intracellular mucin was measured around the circumferential section of the axial bronchus using ImagePro-5.1 (Media Cybernetics). Images were analysed by investigators who were blinded to mouse genotype and treatment (Supplementary Table 1).

Quantification of mucin secretion was performed according to previous protocols[22,57]. First, using images that were acquired in the red channel alone, the total area and fluorescence intensity of intracellular staining above the basement membrane were measured. Second, using the images captured under both red and green fluorescence, the total surface area of the epithelium and the length of the basement membrane in each field were measured. The volume density of mucin staining in the airway epithelium was then calculated stereologically as the ratio of surface area of staining to total surface area of the epithelium divided by a boundary length measurement, which is a product of the total epithelial surface area, the basement membrane length and the geometric constant $4/\pi$. As a result, data are presented as the epithelial mucin volume density, signifying the measured volume of mucin overlying a unit area of epithelial basal lamina.

To measure airway lumenal mucus content in *Syt2*-mutant mice, mucous metaplasia was induced as above, then mucin secretion and bronchoconstriction were induced by exposure for 10 min to an aerosol of 150 mM methacholine. Lungs were collected and fixed by immersion for 48 h at 4 °C to avoid displacement of lumenal mucus and using methanol-based Carnoy's solution (methacarn) for fixation to minimize changes in mucus volume. A single transverse 5 μm section was taken through the axial bronchus of the left lung between lateral branches 1 and 2 and stained with PAFS as above to evaluate intracellular mucin to ensure secretion had been stimulated. Then, six 5 μm sections of the paraffin blocks caudal to the initial section were taken at 500 μm intervals and stained with PAFS. Mucus in the lumens of airways was identified manually and the area summed for all twelve sections using ImagePro-5.1 (Media Cybernetics)[22].

To measure the efficiency of stimulated mucin secretion and the extent of mucus accumulation in the airway lumen of WT mice treated with peptides, the same procedures as those used for analysis of *Syt2*-mutant mice were followed, except that both outcomes were measured in the same mouse because of peptide expense by fixation of the left lung through inflation with formalin to measure mucin secretion and fixation of the right lung through immersion in methacarn to measure mucus accumulation. Stimulation of mucin secretion with a single methacholine aerosol was used for both outcomes. A MicroSprayer Aerosolizer (Penn-Century) was used for intratracheal peptide delivery (50 μl) to the airways, secretion was measured in the left axial bronchus at a site 3 mm distal to the site used in the *Syt2*-mutant mice because of greater intracellular peptide uptake in more distal airways (Extended Data Fig. 9a), and mucus accumulation was measured in the right caudal lobe taking 8–10 sections of 5 μm thickness at 500 μm intervals.

Details about experimental repeats, the number of data points *n* (that is, mice) and the number of images analysed for each mouse are provided in Supplementary Table 1. The experiments, involving all the experimental groups, were performed twice on two different occasions, weeks apart. Lung tissues from the mice from each of those independent experiments were processed separately.

## Epithelial cell uptake of peptides in mice

To measure the uptake of SP9 and control peptides conjugated to CPP in vivo, peptides were labelled with Cy3 by conjugation at the C-terminal cystine residue, as described below. Labelled peptides (200 μM microsprayer concentration) or PBS were introduced into the airways as

aerosols using a Penn-Century MicroSprayer inserted into the distal trachea under direct visualization with a laryngoscope. After 30 min, the mice were euthanized and the lungs were fixed by inflation with 10% neutral buffered formalin as described above for the measurement of mucin secretion. Transverse sections were made of the left axial bronchus, and sections were stained with DAPI to demonstrate nuclei. Fractional uptake of labelled peptides was measure as red Cy3 fluorescent staining over the number of blue DAPI-labelled nuclei in individual cells. To illustrate the relative peptide uptake by secretory and ciliated cells (Extended Data Fig. 9d), immunofluorescence staining of secretory cells was performed using primary goat polyclonal antibodies against CCSP (Millipore Sigma, ABS1673, 1:1,000), secondary donkey anti-goat IgG polyclonal antibodies conjugated to Alexa Fluor 488 (Jackson ImmunoResearch, 705-545-147, 1:1,000) and staining of nuclei with DAPI. Details about the experimental repeats, the number of data points $n$ (that is, cells) and the number of images analysed for each mouse are provided in Supplementary Table 1.

## Protein expression and purification

We used the same constructs and protocols to purify cysteine-free Stx1A, SNAP-25A, VAMP2 and Syt1 as described in ref. [41]. We used the same constructs and protocols to purify NSF, and αSNAP as described in ref. [58]. The protein sample concentrations were measured by ultraviolet light absorption at 280 nm, and aliquots were flash-frozen in liquid nitrogen and stored at −80 °C.

**Stx3.** Full-length human *STX3* was expressed in *Escherichia coli* strain BL-21 (DE3) with an N-terminal, TEV-protease-cleavable hexa-histidine tag. The expression and purification protocols were mostly identical to that of Stx1A. The protein was expressed overnight at 30 °C in 8 l of autoinducing medium. Cell pellets from 8 l of culture were suspended in 1× PBS, 1 mM EDTA, 1 mM PMSF and 8 EDTA-free protease inhibitor tablets (Roche) supplemented with lysozyme and DNase I (Sigma-Aldrich). The cells were lysed using a sonicator (Thermo Fisher Scientific) and an M-110EH microfluidizer (Microfluidics). Inclusion bodies were removed by centrifugation for 30 min at 13,000 rpm in a JA-14 rotor (Beckman Coulter), and the supernatant was centrifuged at 43,000 rpm for 1.5 h in a Ti-45 rotor (Beckman Coulter) to pellet the membrane. Membranes were resuspended in 20 mM HEPES pH 7.5, 300 mM NaCl, 10% glycerol and centrifuged at 43,000 rpm for 1 h. The pellet was resuspended once more in the same buffer, dodecylmaltoside (Anatrace) was added to 2% and the sample was stirred for 1.5 h at 4 °C. The solubilized membrane was centrifuged at 40,000 rpm for 35 min, and the supernatant was loaded onto a 5 ml column of Nickel-NTA agarose (Qiagen). The column was washed with 20 mM HEPES pH 7.5, 20 mM imidazole, 300 mM NaCl, 110 mM octyl glucoside (Anatrace), 10% glycerol and the protein was eluted with wash buffer supplemented with 450 mM imidazole and 1 M NaCl. The protein fractions were pooled, digested with 110 μg TEV protease and dialysed overnight against 20 mM HEPES pH 7.5, 50 mM NaCl, 110 mM octyl glucoside (OG) (Anatrace) and 10% glycerol. The fractions were loaded onto a MonoQ 4.6/100 PE column (GE Healthcare) that had been equilibrated with dialysis buffer. The protein was eluted with a gradient of 50 mM to 1 M NaCl over 30 column volumes. Protein concentration was measured by absorption at 280 nm and aliquots were flash-frozen in liquid nitrogen and stored at −80 °C.

**SNAP-23.** The expression and purification protocols were mostly identical to those of SNAP-25A. Cysteine free SNAP-23, in which all cysteine residues were changed to serine, was expressed in BL21(DE3) cells using autoinducing medium from a pGEX vector as an N-terminal GST tag with a thrombin protease cleavage site to remove the tag. Cells from 4.0 l of the induced culture were resuspended in 250 ml of buffer (20 mM HEPES pH 7.5, 300 mM NaCl, 4 mM DTT, 10% glycerol) containing 1 mM PMSF and 5 EDTA-free protease inhibitor tablets. Cells were lysed by sonication. The lysate was clarified by centrifugation in a Ti45

rotor for 35 min at 40,000 rpm. The supernatant was bound to 10 ml of Glutathione Sepharose beads (GE Healthcare) for 1 h with stirring at 4 °C. The beads were collected by centrifugation, poured into a column and washed with 100 ml of buffer (20 mM HEPES pH 7.5, 300 mM NaCl, 4 mM DTT, 10% glycerol). Thrombin (10 μl of 5 mg ml⁻¹) was added to the washed beads along with 5 ml of buffer (20 mM HEPES pH 7.5, 300 mM NaCl, 4 mM DTT, 10% glycerol) and the mixture was rocked overnight at 4 °C to remove the GST tag. The cleaved SNAP-23 sample was washed out of the column using buffer (20 mM HEPES pH 7.5, 300 mM NaCl, 4 mM DTT, 10% glycerol) and concentrated to 5 ml. The sample was injected onto a Superdex 200 (16/60) column (GE Healthcare) equilibrated in 20 mM HEPES pH 7.5, 100 mM NaCl and 10% glycerol. Protein-containing fractions were combined and concentrated to around 100 μM SNAP-23. The protein concentration was measured by absorption at 280 nm and aliquots were flash-frozen in liquid nitrogen and stored at −80 °C.

**VAMP8.** VAMP8 was expressed in *E. coli* strain BL-21 (DE3) with an N-terminal, TEV protease-cleavable, GST tag. The protein was expressed overnight at 25 °C in 8.0 l of autoinducing medium. Cell pellets were suspended in 20 mM HEPES pH 7.5, 300 mM NaCl, 2 mM DTT, 1 mM EDTA and 8 EDTA-free protease inhibitor tablets supplemented with lysozyme and DNase I. The cells were lysed using a sonicator (Thermo Fisher Scientific) and an M-110EH microfluidizer. Cell debris was removed by centrifugation for 30 min at 13,000 rpm in a JA-14 rotor, and the supernatant was centrifuged at 43,000 rpm for 1 h in a Ti-45 rotor to pellet the membrane. The pellet was resuspended in 20 mM HEPES pH 7.5, 300 mM NaCl, 2 mM DTT, 1 mM EDTA and 1.5% DDM and solubilized at 4 °C with stirring for 2.5 h. The solubilized membrane was centrifuged at 43,000 rpm for 35 min, the supernatant was mixed with 5 ml of Glutathione Sepharose 4B and incubated overnight at 4 °C with end-over-end mixing. The beads were washed with 20 CV of 20 mM HEPES pH 7.5, 300 mM NaCl, 2 mM DTT, 1 mM EDTA and 110 mM OG. The protein was cleaved off the column by resuspending the beads in 2 ml of wash buffer supplemented with 110 μg of TEV protease, and incubating at 4 °C for 1 h. After digestion, the column was drained and the flow through (containing cleaved VAMP8) was injected onto a Superdex 200 10/300 Increase column (GE Healthcare) equilibrated with 20 mM HEPES pH 7.5, 300 mM NaCl, 2 mM DTT and 110 mM OG. The fractions containing protein were pooled, and protein concentration was measured by absorption at 280 nm. Aliquots were flash-frozen in liquid nitrogen and stored at −80 °C.

**Syt2.** *Syt2* was expressed in BL21(DE3) cells using autoinducing medium from a pGEX vector as an N-terminal GST tag with a thrombin protease cleavage site to remove the tag. Cells from 4 l of induced culture were resuspended in 200 ml of buffer (20 mM HEPES pH 7.5, 300 mM NaCl, 1 mM EDTA and 2 mM DTT), containing 4 EDTA-free protease inhibitor tablets. Cells were lysed by three passes through the Emulsiflex C5 homogenizer (Avestin) at 15,000 p.s.i. Unlysed cells and debris were removed by centrifugation in a JA-14 rotor for 10 min at 8,000 rpm, the supernatant was centrifuged again in the same rotor for 10 min at 8,000 rpm to remove any final debris. The supernatant from the second centrifugation was then centrifuged in a Ti45 rotor for 1 h at 40,000 rpm to collect the membranes. Membranes were resuspended using a Dounce homogenizer in 100 ml of buffer (20 mM HEPES pH 7.5, 300 mM NaCl, 1 mM EDTA and 2 mM DTT) and *n*-dodecylmaltoside was added to a final concentration of 2% (w/v) to solubilize the membranes overnight at 4 °C with stirring. The extract was clarified by centrifugation using the Ti45 rotor at 40,000 rpm for 35 min. The extract was applied to a 5 ml bed of Glutathione Sepharose by stirring at 4 °C for 2 h. The column was washed with buffer (20 mM HEPES pH 7.5, 300 mM NaCl, 1 mM EDTA, 2 mM DTT) containing 110 mM β-octyl-glucoside and eluted with buffer (20 mM HEPES pH 7.5, 300 mM NaCl, 1 mM EDTA, 2 mM DTT) containing 110 mM β-octyl-glucoside and 20 mM

reduced Glutathione. The GST tag was removed by cleavage with 10 µl of 5 mg ml⁻¹ thrombin for 2 h and the Syt2 sample was purified on a monoS column equilibrated in 20 mM HEPES pH 7.5, 100 mM NaCl, 110 mM β-octyl-glucoside and 2 mM DTT (monoS buffer). After washing the column with monoS buffer (20 mM HEPES pH 7.5, 100 mM NaCl, 110 mM β-octyl-glucoside, 2 mM DTT), the protein was eluted using a linear gradient from 100 mM to 1 M NaCl. Protein-containing fractions were combined, the protein concentration was measured by absorption at 280 nm and aliquots were flash-frozen in liquid nitrogen and stored at −80 °C.

**Syt1 C2B, Syt2 C2B and Syt1 C2B(QM).** The Syt1 C2B (amino acid range 271–421), Syt2 C2B (amino acid range 272–422) domains and the Syt1 C2B(QM) (amino acid range 271–421, R281A, E295A, Y338W, R398A, R399A) mutant were expressed as GST-tagged fusion proteins in *E. coli* BL21 (DE3) cells at 30 °C overnight. After collecting the cells by centrifugation, the sample was resuspended in lysis buffer containing 50 mM HEPES-Na pH 7.5, 300 mM NaCl, 2 mM DTT and EDTA-free protease inhibitor cocktail, and then sonicated and centrifuged. The supernatant was incubated with Glutathione Sepharose beads. The resin was extensively washed with 50 ml of wash buffer I containing 50 mM HEPES-Na pH 7.5, 300 mM NaCl and 1 mM DDT, followed by 50 ml of wash buffer II containing 50 mM HEPES-Na pH 7.5, 300 mM NaCl, 1 mM DTT and 50 mM CaCl₂. The GST tag was cleaved overnight at 4 °C with PreScission protease (GE Healthcare) in cleavage buffer containing 50 mM HEPES-Na pH 7.5, 300 mM NaCl, 1 mM DTT and 2 mM EDTA. The cleaved proteins were purified by monoS column and gel filtration on Superdex 75 (GE Healthcare). The protein concentration was measured by absorption at 280 nm and aliquots were flash-frozen in liquid nitrogen and stored at −80 °C.

**Munc13-2*.** The Munc13-2* fragment of Munc13-2 (amino acid range 451–1407, that is, including the C1, C2B and the C-terminally truncated MUN domains, but excluding residues 1326–1343) was cloned into a pFastBac HTB vector with a GST tag and a PreScission cleavage site. The deletion of residues 1326–1343 in this construct prevents aggregation[39,59], and the C-terminal truncation improves solubility. Cells from 8 l of SF9 cell culture were resuspended in 200 ml resuspension buffer (RB) (50 mM Tris, pH 8.0, 500 mM NaCl, 1 mM EDTA, 0.5 mM TCEP, 10% glycerol) containing 6 EDTA-free protease inhibitor tablets. The cells were lysed by three passes through the Avestin C5 homogenizer at 15,000 p.s.i. The lysate was clarified by centrifugation for 35 min at 40,000 rpm in a Ti45 rotor. The supernatant was mixed with 15 ml Glutathione Sepharose beads at 4 °C with stirring for 2 h. The beads were washed using the Akta Prime system (GE Healthcare) with 20 ml RB, 90 ml RB + 1% Triton X-100, then eluted with RB + 50 mM reduced Glutathione. Peak fractions were pooled and then 100 µl of 10 mg ml⁻¹ PreScission protease was added and incubated overnight. The cleaved proteins were purified by gel filtration on the Superdex 200 column. The protein concentration was measured by absorption at 280 nm and aliquots were flash-frozen in liquid nitrogen and stored at −80 °C.

**Munc18-2.** Munc18-2 (amino acid range 1–594) was cloned into a pFastBac HTB vector with an N-terminal hexa-histidine tag and a TEV-cleavage site. Cells from 4.0 l of a SF9 cell culture were resuspended in 100 ml resuspension buffer (RB) (20 mM sodium phosphate, pH 8.0, 300 mM NaCl, 2 mM DTT, 10% glycerol with 1 mM PMSF) containing 6 EDTA-free protease inhibitor tablets. The cells were lysed via 3 passes through the Avestin C5 homogenizer at 15,000 p.s.i. The lysate was clarified by centrifugation for 35 min at 40,000 rpm in a Ti45 rotor. The supernatant was mixed with 3 ml Ni-NTA beads at 4 °C stirring for 1 h. The beads were washed using an Akta Prime system with 20 ml each of RB, then eluted with RB + 300 mM imidazole. Peak fractions were pooled and then 100 µl of 11 mg ml⁻¹ TEV protease was added. The mixture was dialysed overnight against 1 l of 500 ml of 20 mM HEPES pH 7.5, 300 mM NaCl, 2 mM DTT, 10% glycerol. The TEV cleaved protein was injected on a Superdex 200 column. Peak fractions were combined and the protein concentration was measured by ultraviolet light absorption at 280 nm. Aliquots of 100 µl were flash-frozen in liquid N₂ and stored at −80 °C.

### Peptide synthesis

The stapled peptide SP9 and the non-stapled peptide P0 (Fig. 2c), as well as peptide chimeras with conjugated CPPs or biotin at the N terminus, and/or conjugated fluorescent dye Cy3 labels at the C terminus (SP9–Cy3, biotin–SP9–Cy3, PEN–SP9–Cy3, PEN–P9–Cy3, TAT–SP9–Cy3 and TAT–P9–Cy3 (Fig. 4a); P0–Cy3 (Fig. 2f)) were synthesized and purified by Vivitide (formerly New England Peptide). Peptide synthesis was carried out using solid-phase peptide synthesis and Fmoc chemistry. The peptides were cleaved using trifluoroacetic acid and standard scavengers. The peptides were purified using reverse-phase high-pressure liquid chromatography (RP-HPLC). For the stapled peptides, α,α-disubstituted non-natural amino acids of olefinic side chains were synthesized (*S5-S* stereochemistry, bridging 5 amino acids).

The hydrocarbon-staple was made using Grubbs catalyst[36]. For all stapled peptides, the N termini were acetylated and the C termini were amidated. For example, SP9 was synthesized at a 0.2 mmol scale using Rink amide resin on a Liberty Blue instrument (CEM). Standard protecting groups were used for all amino acids. All amino acids were coupled using 5 equivalents of amino acid/HBTU/DIEA relative to resin loading; amino acids after *S5* were triple coupled using the same molar excess. Fmoc deprotection was performed with 20% piperidine in dimethylformamide (DMF). After final Fmoc deprotection, the N terminus was acetylated using 0.8 M acetic anhydride and 0.43 M *N*-methyl-2-pyrrolidone in DMF. Ring-closing metathesis was performed using first-generation Grubbs catalyst in dichloroethane (DCE); the reaction was allowed to proceed overnight protected from light. The resin was then rinsed with DCE, followed by 1% sodium diethyldithiocarbamate trihydrate in DMF (4 × 30 min). The resin was then rinsed with DMF and dichloromethane. The peptide was cleaved and deprotected using trifluoroacetic acid:H₂O:ethane-1,2-dithiol:thioanisole/ethylmethylsulfide (84:4:4:4:4) for 3 h, precipitated in ether and centrifuged to pellet. The pellet was resuspended in ether and centrifuged, after which the solvent was decanted. The pellet was dissolved in 1:1 acetonitrile:H₂O and lyophilized. The crude peptide was purified by RP-HPLC using a C18 column (10 µm, 120 Å, 25 × 250 mm), and a gradient of 42–58% buffer B (0.1% trifluoroacetic acid in acetonitrile) for 140 min.

The biotin-labelled stapled peptide, biotin–SP9–Cy3, was biotinylated at the N terminus by cross-linking biotin through the carbon spacer 6-aminohexaonic.

For the specified peptides, the C-terminal cystine residue was conjugated to Cy3 fluorescent dyes through maleimide reaction chemistry at pH 7.4 and a 1–2 molar ratio of dye to peptide, and after conjugation the Cy3-labelled peptides were purified again. For example, purified SP9 was dissolved in PBS:acetonitrile:DMSO (2:1:1). Cy3-maleimide (1 equivalent) was dissolved in DMSO and added to the peptide solution; the reaction was allowed to proceed for 1 h in the dark. The conjugated peptide was purified by RP-HPLC using a C18 column (10 µm, 120 Å, 25 × 250 mm), and a gradient of 46–66% buffer B for 140 min.

All of the peptides were purified to >90–95% and quality control was performed by liquid chromatography coupled with mass spectrometry (LC–MS) by the manufacturer (HPLC chromatograms and LC–MS data for SP9, TAT–SP9–Cy3, PEN–SP9–Cy3, PEN–P9–Cy3, TAT–P9–Cy3, SP9–Cy3 and P0 are provided in Supplementary Figs. 2–8, respectively). Subsequently, the peptides were lyophilized and shipped. The LC–MS quality-control data indicate that the peptides have the predicted molecular mass according to their chemical composition. Moreover, 1H 1D and 2D HSQC, HMBC, ROESY NMR experiments of 5 mg SP9 dissolved in DMSO (Supplementary Fig. 9) show that the staple (*S5*)-residues are at the expected positions in the peptide sequence and are connected

with the neighbouring residues. Although there is some spectral overlap for some of the resonances, the data are that are consistent with formation of two $S5$-$S5$ pairs. Taken together, the data show that SP9 has the expected sequence and chemical configuration.

For each group of experiments, aliquots of peptide powder were directly dissolved in the specified buffers at ~1 mM concentration using a vortexer, and then diluted to the specified peptide concentrations. For example, to prepare a stock solution of SP9, 2 mg SP9 peptide powder was weighed out. Considering the molecular mass of 2222 g mol$^{-1}$ of SP9 (Supplementary Fig. 2), this corresponds to $9 \times 10^{-7}$ mol. For the desired concentration of 1 mM SP9, we added $9 \times 10^{-7}$ mol l$^{-1} \times 10^{-3}$ mol l$^{-1}$ = 0.9 ml buffer. The concentration of the stock solution was confirmed by absorption measurement at 205 nm using a Nanodrop instrument (Thermo Fisher Scientific).

## CD spectroscopy

CD spectra were measured using the AVIV stop-flow CD spectropolarimeter at 190–250 nm using a cell with a 1 mm path length. The sample containing 100 μM of synthesized peptides in PBS buffer (137 mM NaCl, 2.7 mM KCl, 10 mM Na$_2$HPO$_4$ and 2 mM KH$_2$PO$_4$, pH 7.4) was measured at 20 °C. For the correction of the baseline error, the signal from a blank run with PBS buffer was subtracted from all the experimental spectra. The α-helical content of each peptide was calculated by dividing the mean residue ellipticity $[\varphi]222_{obs}$ by the reported $[\varphi]222_{obs}$ for a model helical decapeptide[60].

## Cryo-electron microscopy

PM and SG vesicles were separately vitrified on lacey carbon grids using the Vitrobot (Thermo Fisher Scientific), and imaged using the FEI Tecnai F20 transmission cryo-electron microscope with a field emission gun (FEI) operated at 200 kV. Images were recorded on a Gatan K2 Summit electron-counting direct detection camera (Gatan) in electron-counting mode[61]. Nominal magnifications of ×5,000 and ×9,600 (corresponding to pixel sizes of 7.4 Å and 3.8 Å) were used for airway PM and SG vesicles, respectively (Extended Data Fig. 6a). The diameters of the vesicles (Extended Data Fig. 6b) were measured using EMAN2 (ref. [62]).

## Bulk fluorescence anisotropy measurements

In the bulk fluorescence anisotropy experiments, P0 and SP9 were labelled with the fluorescent dye Cy3 at the C terminus. The fluorescence anisotropy was measured using the Tecan Infinite M1000/PRO fluorimeter using an excitation wavelength of 530 ± 5 nm and emission wavelength of 580 ± 5 nm at 27.2 °C. The fluorescent-dye-labelled samples were diluted to 10 nM concentration in TBS (20 mM Tris, pH 7.5, 150 mM NaCl, 0.5 mM TCEP) for optimal read out.

## Molecular dynamics simulations

The starting point for all of the molecular dynamics simulations was the crystal structure of the SNARE–Syt-1–complexin-1 complex at 1.85 Å resolution (PDB: 5W5C)[15]. Before the simulations, the Syt1 C2A domain, the crystallographic water molecules, Mg$^{2+}$ and glycerol molecules were deleted from the crystal structure. Specifically, the following residues were included in the simulations of the primary interface: Syt1 C2B (amino acid range 270–419), synaptobrevin-2 (amino acid range 29–66), Stx1A (amino acid range 191–244), SNAP-25A (amino acid ranges 10–74 and 141–194). Complexin-1 was not included in the simulations. For the primary interface (SNARE–Syt1 C2B) simulations, the Syt1 C2B molecule that produces the primary interface was used.

For the simulations of Syt1 C2B–P9, Syt1 C2B (amino acid range 270–419) and residues 37–53 of SNAP-25 were used (corresponding to the P9 sequence: EESKDAGIRTLVMLDEQ). For the simulations of Syt1 C2B–SP9, the Syt1 C2B–P9 complex was used as a starting point and the staples for SP9 were created by using CHARMM topology and parameter files for $S5$ and the covalent bond between $S5$ residues[63]. Initial coordinates

for the $S5$ residues were generated by mutating the native residues into Lys using PyMol v.2.5.1 (Schrödinger), and then using the VMD mutate command[64] to change Lys into $S5$. The SP9 and P9 peptides were simulated with an acetylated N terminus, and an amidated C terminus. For all of the simulations, the NAMD program was used[65].

As a control, five 1 μs molecular dynamics simulations of the primary interface were performed in a solvated environment (Extended Data Fig. 2d, e). For these simulations, the starting models were placed in a 113 × 125 × 116 Å periodic boundary condition box. The empty space in the box was filled with 50,420 water molecules using the VMD solvate plugin. The system has a total of 157,833 atoms. The system was charge-neutralized and ionized by addition of 155 potassium and 138 chloride ions, corresponding to a salt concentration of ~145 mM using the VMD autoionize plugin.

For the simulations with P9 and SP9, the starting models were placed in a 80 × 80 × 80 Å periodic boundary condition box. The empty space in the box was filled with ~15,200 water molecules using the VMD solvate plugin. The system has a total of 48,486 atoms. The system was charge-neutralized and ionized by the addition of 42 potassium and 44 chloride ions, corresponding to a salt concentration of ~145 mM using the VMD autoionize plugin.

The CHARMM22 (P9–Syt1 C2B and SP9–Syt1 C2B simulations) or CHARMM36 (primary interface simulations) all-hydrogen force fields and parameters[66] were used with a non-bonded cut-off of 11 Å. A constant-pressure method was used by adjusting the size of the box. The particle mesh Ewald method was used to accelerate the calculation of long-range electrostatic non-bonded energy terms. Langevin dynamics (with a friction term and a random force term) was used to maintain the temperature of the simulation. All hydrogen-heavy-atom bonds were kept rigid using the Rattle method as implemented in NAMD.

For the simulations with stapled peptides, in the relaxation step, dihedral angle restraints were added to restrain the $S5$–$S5$ CE–CE double bond in the *cis* conformation, the $S5$ olefinic side chains in the *trans* conformation, and all α-helices in the α-helical conformation (using the ssrestraints plugin for VMD). In all of the subsequent steps (heating steps and production runs), all these dihedral angle restraints were turned off. For all of the other simulations with peptides without staples, in the relaxation step, α-helical (secondary structure) restraints were added for all α-helices (using the ssrestraints plugin for VMD). In all of the subsequent steps (heating steps and production runs), all these dihedral angle restraints were turned off. The system was equilibrated by the following procedure: (1) relaxation step, ramping up the temperature from 0 to 50 K for 50 ps with a 1 fs time step; (2) first heating step, ramping up the temperature from 50 to 100 K for 50 ps with a 1 fs time step; (3) second heating step, ramping up the temperature from 100 to 250 K for 150 ps with a 1 fs time step. For all simulations, 1 ns chunks were run at a temperature of 300 K with a time step of 1 fs. Five independent 1 μs simulations were performed for each system (primary interface, SP9–Syt1 C2B, P9–Syt1 C2B) by using different initial random number seeds. As expected, the primary interface is stable in these simulations.

One simulation of P9–Syt1 C2B resulted in a dissociation event (Fig. 2g (green, right)). Interestingly, the dissociated peptide P9 is highly dynamic, revealing a variety of distorted, partially helical conformations. Presumably, the increased dynamics of the non-stapled P9 peptide resulted in the destabilization of the interactions with Syt1 C2B, producing the rather different binding poses of P9 (Fig. 2g).

All of the simulations were performed on the Stanford Sherlock Cluster using 4 nodes, each node consisting of dual ten-core CPU 2.4 Ghz Intel processors, that is, a total of 80 CPUs were used for each simulation. The MPI-parallel NAMD2 2.14b1 executable was used. To visualize the results, only protein components are shown, and all of the structures were fitted to each other, and displayed using PyMOL v.2.5.1.

## Vesicle reconstitution

For the ensemble lipid mixing assay, the lipid composition of the SV vesicles was phosphatidylcholine (PC) (46%), phosphatidylethanolamine (PE) (20%), phosphatidylserine (PS) (12%), cholesterol (20%) and 1,1′-dioctadecyl-3,3,3′,3′-tetramethylindodicarbocyanine perchlorate (DiD) (Invitrogen) (2%); for the both neuronal and airway PM vesicles, the lipid composition was brain total lipid extract supplemented 3.5 mol% PIP2, 0.1 mol% biotinylated PE and 2 mol% 1,1′-dioctadecyl-3,3,3′,3′-tetramethylindocarbocyanine perchlorate (DiI) (Invitrogen). All the lipids are from Avanti Polar Lipids.

For single-vesicle content mixing assay, the lipid composition of the SV, SG, VAMP2, or VAMP8 vesicles was PC (48%), PE (20%), PS (12%) and cholesterol (20%); for both the neuronal and airway PM vesicles, the lipid composition was brain total lipid extract supplemented 3.5 mol% PIP2 and 0.1 mol% biotinylated PE.

The reconstitution method for neuronal PM and SV vesicles is described in detail in refs. [41,67,68]. The same methods were used for airway PM, SG, VAMP2 and VAMP8 vesicles. Dried lipid films were dissolved in 110 mM OG buffer containing purified proteins at protein-to-lipid ratios of 1:200 for VAMP2 and Stx1A for SV and neuronal PM vesicles, respectively (or 1:200 for VAMP8 and Stx3 for SG and airway PM vesicles, respectively), and 1:800 for Syt1 for SV vesicles (or 1:1,200 for Syt2 for SG vesicles).

A three to fivefold excess of SNAP-25A or SNAP-23 (with respect to Stx1A or Stx3) was added to the protein–lipid mixture for neuronal or airway PM vesicles. Detergent-free buffer (20 mM HEPES pH 7.4, 90 mM NaCl and 0.1% 2-mercaptoethanol) was added to the protein–lipid mixture until the detergent concentration was at (but not lower than) the critical micelle concentration of 24.4 mM, that is, vesicles did not yet form. For the preparation of SV, SG, VAMP2, or VAMP8 vesicles for the single-vesicle content mixing assay, 50 mM sulforhodamine B (Thermo Fisher Scientific) was added to the protein–lipid mixture. The vesicles subsequently formed during size-exclusion chromatography using a Sepharose CL-4B column, packed under near constant pressure by gravity with a peristaltic pump (GE Healthcare) in a 5.5 ml column with a ~5 ml bed volume that was equilibrated with buffer V (20 mM HEPES pH 7.4 and 90 mM NaCl) supplemented with 20 µM EGTA and 0.1% 2-mercaptoethanol. The eluent was dialysed into 2 l of detergent-free buffer V supplemented with 20 µM EGTA, 0.1% 2-mercaptoethanol, 5 g of Bio-beads SM2 (Bio-Rad) and 0.8 g l⁻¹ Chelex 100 resin (Bio-Rad). After 4 h, the buffer was exchanged with 2 l of fresh buffer V supplemented with 20 µM EGTA, 0.1% 2-mercaptoethanol and Bio-beads, and the dialysis was continued overnight for another 12 h. We note that the chromatography equilibration and elution buffers did not contain sulforhodamine, so the effective sulforhodamine concentration inside SV, SG, VAMP2 or VAMP8 vesicles is considerably (up to tenfold) lower than 50 mM. For the ensemble lipid mixing assay, the reconstitution method is the same as that for the single-vesicle content mixing assay, except that 50 mM sulforhodamine B was omitted for all the steps.

As described previously[67], the presence and purity of reconstituted proteins in the airway system was confirmed by SDS–PAGE of the vesicle preparations and the directionality of the membrane proteins (facing outward) was assessed by chymotrypsin digestion followed by SDS–PAGE gel electrophoresis. The size distributions of the airway PM and SG vesicles were analysed by cryo-electron microscopy (Extended Data Fig. 6a, b) as described previously[69].

## Single-molecule counting experiments with SP9–Cy3

PEG-coated flow chambers were prepared using the same protocol as for the single-vesicle content mixing experiments. Freshly synthesized SP9–Cy3 powder was first dissolved in an imaging buffer (20 mM HEPES pH 7.4, 90 mM NaCl and 0.5 mM TCEP) at a concentration of 50 µM, then centrifuged at around 16,000 g for 10 min to remove potential insoluble materials. The sample's concentration was remeasured by

absorption by Cy3 at 550 nm before serial dilution to concentrations of 100 nM, 10 nM, 1 nM and 0.5 nM. Diluted sample (5 µl) was injected into a flow chamber on the quartz slides followed by an immediate (~500 µl) wash with imaging buffer. There is some degree of non-specific binding of SP9–Cy3 to the imaging surface, enabling the counting of molecules in fluorescent spots by observing single-molecule photobleaching events[70]. After quickly focusing, the sample stage was moved to a fresh location within the same sample chamber distant from prior illumination and the recording was started before exciting SP9–Cy3 by green (532 nm) laser light at an excitation power of ~8 mW. Multiple recordings were performed at fresh locations within the same chamber.

The number of SP9–Cy3 molecules in a fluorescent spot was counted by observing sequential stepwise photobleaching events. Fluorescent spots were automatically detected by smCamera and time traces for each spot generated. The time traces were automatically analysed by Hidden Markov modelling[71,72] using a script written for MATLAB. We applied constraint-based clustering to initiate the Hidden Markov model (HMM) and calculated the probability matrices of transition and emission iteratively. The most probable state sequences were then reconstructed with a standard Viterbi algorithm. The time traces and automatic HMM fits were manually inspected. For many traces, there were distinct stepwise decreases in fluorescence intensity where the stepwise decreases were approximately as recognized by HMM (Extended Data Fig. 3c). Traces were selected that showed distinct stepwise fluorescence intensity decreases and that had undergone complete photo-bleaching at the end of the observation period. Histograms of the number of photo-bleaching steps (also known as the number of SP9–Cy3 molecules per fluorescent spot) were then generated (Extended Data Fig. 3d).

## Single-vesicle content mixing experiments

All single-vesicle fusion experiments were performed on a prism-type total internal reflection fluorescence microscope using 532 nm (green) laser (CrystaLaser) and 637 nm (red) laser (OBIS) excitation. Two observation channels were created by a 640 nm single-edge dichroic beamsplitter (FF640-FDi01-25x36, Semrock): one channel was used for the fluorescence emission intensity of the content dyes and the other channel for that of the Cy5 dye that is part of the injected Ca²⁺ solution. The two channels were recorded on two adjacent rectangular areas (45 × 90 µm²) of a charge-coupled device camera (iXon+ DV 897E, Andor Technology). The imaging data were recorded and analysed using the smCamera program[73] developed by K. Suk Lee and T. Ha. Fluorescent peaks were automatically detected using smCamera and time traces were saved in smCamera format as well as in plain text (scripts to convert the smCamera files to tiff and text files were provided by M. Hyn Jo). Candidates for fusion events in the time traces were detected using a script written for MATLAB and then confirmed by manual inspection.

Flow chambers were assembled by creating a 'sandwich' consisting of a quartz slide and a glass coverslip that were both coated with polyethylene glycol (PEG) molecules, including 0.1% (w/v) biotinylated-PEG except when stated otherwise, and using double-sided tape to create up to five flow chambers. Coating the surface with PEG molecules alleviates non-specific binding of vesicles. The same protocol and quality controls (surface coverage and non-specific binding) were used as described previously[67,74] except that PEG-SVA (Laysan Bio) instead of mPEG-SCM (Laysan Bio) was used as it has a longer half-life. The flow chambers were incubated with neutravidin for 30 min (0.1 mg ml⁻¹).

For the single-vesicle fusion experiments described in Fig. 3 and Extended Data Figs. 4, 5 and 7, biotinylated neuronal or airway PM vesicles (100× dilution) were tethered to the imaging surface by incubation at room temperature (25 °C) for 30 min followed by three rounds of washing with 120 µl buffer V to remove unbound neuronal or airway PM vesicles; each buffer wash effectively replaces the 3 µl flow chamber volume more than 100 times.

For the complete reconstitution (Fig. 3), to form airway SM vesicles with reconstituted Stx3–Munc18-2 complex, we added the 'disassembly factors' (1 μM Munc18-2, 0.5 μM NSF, 5 μM αSNAP, 3 mM ATP and 3 mM $Mg^{2+}$) to tethered airway PM vesicles (Fig. 3b), according to previous work with neuronal proteins[41]. This procedure results in tethered SM vesicles. Next, the flow chamber with the tethered SM vesicles was washed with buffer V along with 0.5 μM Munc-13-2* and 2 μM SNAP-23.

For all of the reconstitution experiments, after the start of illumination and recording of the fluorescence from a particular field of view of the flow chamber, SV, SG, VAMP2 or VAMP8 vesicles (diluted 100 to 1,000 times; including peptides at the specified concentration, 0.5 μM Munc-13-2* and 2 μM SNAP-23, if applicable) were loaded into the flow chamber to directly monitor vesicle association of SG, SV, VAMP2 or VAMP8 vesicles to neuronal or airway PM vesicles for 1 min. When peptide was included in a particular experiment, it was mixed with the SV, SG, VAMP2 or VAMP8 vesicles before loading into the flow chamber. Thus, the peptide would have a chance to bind to Syt1 or Syt2 in the SV or SG vesicles before loading them into the flow chamber. While continuing the recording, the flow chamber was washed three times with 120 μl of buffer V (including peptides at the specified concentration, 0.5 μM Munc-13-2* and 2 μM SNAP-23, if applicable) to remove unbound vesicles.

For the complete reconstitution (Fig. 3), note that Munc13-2* will catalyse the transfer of Stx3 from the Stx3–Munc18-2 complex into the ternary SNARE complex with SNAP-23 and VAMP8; we therefore call the tethered vesicles again as PM vesicles after this transfer (Fig. 3b).

Subsequently, we continued recording for another minute to monitor spontaneous fusion events. To initiate $Ca^{2+}$-triggered fusion events within the same field of view, a solution consisting of buffer V, 500 μM $Ca^{2+}$ or 50 μM $Ca^{2+}$, 500 pM Cy5 dye molecules (used as an indicator for the arrival of $Ca^{2+}$ in the evanescent field) and, if applicable, peptide was injected into the flow chamber. The injection was performed at a speed of 66 μl s$^{-1}$ by a motorized syringe pump (Harvard Apparatus) using a withdrawal method similar to the one described previously[74].

## Multiple acquisition rounds and repeats for the single-vesicle content mixing experiments

To increase the throughput of the assay and make better use of the vesicle samples, after intensive washing (3 × 120 μl) with buffer V (which includes 20 μM EGTA to remove $Ca^{2+}$ from the sample chamber), we repeated the entire acquisition sequence (SV, SG, VAMP2 or VAMP8 vesicle loading, counting the number of freshly associated vesicle-vesicle pairs, monitoring of $Ca^{2+}$-independent fusion, $Ca^{2+}$-injection and monitoring of $Ca^{2+}$-triggered fusion) in a different imaging area within the same flow chamber. Five such acquisition rounds were performed with the same sample chamber. SV, SG, VAMP2 or VAMP8 vesicles were diluted 1,000× for the first and second acquisition rounds, 200× for the third and fourth acquisition rounds, and 100× for the fifth acquisition round to offset the slightly increasing saturation of the surface with SG, SV, VAMP2 or VAMP8 vesicles. The entire experiment (each with five acquisition rounds) was then repeated several times (Supplementary Table 2) (referred to as repeat experiment). Among the specified number of repeats, there are at least three different protein preparations and vesicle reconstitutions, so the variations observed in the bar charts reflect sample variations as well as variations among different flow chambers. At least two independent reconstitutions were performed for each condition, and multiple technical repeats were performed using different imaging areas, so the number n refers to the number of repeats combining at least two independent reconstitutions for each condition; all of the repeats were successful, and the number of repeats was deemed to be sufficient to reach significance between different conditions.

## Cell culture

Primary HAE cells from several donors were obtained from Promocell at passage 2 or isolated from fresh tissues that were obtained during tumour resections or lung transplantation with fully consent of patients (Ethics approval: ethics committee Medical School Hannover, project no. 2701-2015). Cells were isolated according to the protocol by ref. [57], aliquots were maintained in liquid nitrogen until use. HAE cells from individual donors were thawed and expanded in a T75 flask (Sarstedt) in Airway Epithelial Cell Basal Medium supplemented with Airway Epithelial Cell Growth Medium Supplement Pack (both Promocell) and with 5 μg ml$^{-1}$ Plasmocin prophylactic, 100 μg ml$^{-1}$ Primocin and 10 μg ml$^{-1}$ Fungin (all from InvivoGen). Growth medium was replaced every two days. After reaching 90% confluence, HAE cells were detached using DetachKIT (Promocell) and seeded into 6.5 mm Transwell filters with a 0.4 μm pore size (3470, Corning Costar). The filters were precoated with collagen solution (StemCell Technologies) overnight and irradiated with ultraviolet light for 30 min before cell seeding for collagen cross-linking and sterilization. Cells (3.5 × 10$^4$) in 200 μl growth medium were added to the apical side of each filter, and an additional 600 μl of growth medium was added basolaterally. The apical medium was replaced after 48 h. After 72–96 h, when cells reached confluence, the apical medium was removed and basolateral medium was switched to differentiation medium ± 10 ng ml$^{-1}$ IL-13 (IL012; Merck Millipore). Differentiation medium consisted of a 50:50 mixture of DMEM-H and LHC Basal (Thermo Fisher Scientific) supplemented with Airway Epithelial Cell Growth Medium Supplement Pack as previously described[75] and was replaced every 2 days. Air lifting (removal of apical medium) defined day 0 of ALI culture, and cells were grown at ALI conditions until experiments were performed at day 25 to 28. To avoid mucus accumulation on the apical side, HAE cell cultures were washed apically with Dulbecco's phosphate buffered solution (DPBS) for 30 min every 3 days from day 14 onwards.

## Mucin-secretion assay in HAE cells

Mucin-secretion experiments under static, that is, non-perfused, conditions were conducted as described previously[51,52,76] with modifications for the peptide treatments. In brief, for the 24 h peptide treatment, 20 μl of DMEM ± 100 μM peptides was added to the apical surface 24 h before stimulation. On the day of the assay, cells were washed five times with 100 μl DMEM for 1 h for each wash on the apical side. Apical supernatants were collected after every wash (wash 1–5), then 100 μl of DMEM ± 100 μM peptides was added to the apical surface and HAE cells incubated for 15 min before collecting the supernatant (baseline wash). HAE cells were then incubated for an additional 15 min with 100 μl DMEM ± 100 μM ATP (Sigma-Aldrich) before collecting the supernatants (experimental washes) (Fig. 4e). After sample collection, cells were lysed in 100 μl of lysis buffer (lysate) containing 50 mM Tris-HCl pH 7.2, 1 mM EDTA, 1 mM EGTA, 1% Triton-X (Sigma-Aldrich), protease inhibitor cOmplete mini EDTA-free and phosphatase inhibitor PhosSTOP (Roche). The protocol was adapted for 30 min peptide treatment as follows. After wash 5, 100 μl of DMEM ± 10 μM or 100 μM of peptides (Fig. 4e) was added to the apical surface and HAE cells incubated for 30 min (baseline wash). HAE cells were then incubated for 30 min with 100 μl DMEM ± 100 μM ATP to collect experimental washes.

All of the samples were diluted 1:10 in PBS (washes and cell lysates) and 50 μl of each sample was vacuum-aspirated onto a 0.45 μm pore nitrocellulose membrane using the Bio-Dot Microfiltration Apparatus (Bio-Rad). Subsequently, membranes were incubated with Intercept blocking buffer (Li-Cor) for 1 h before probing with anti-MUC5AC (MA1–21907, Invitrogen) added at 1:250 in Intercept blocking buffer for 1 h. Membranes were then washed four times for 10 min in PBS-Tween-20 (PBST) before incubation with the IRDye secondary antibodies (926–33212 or 926–68072; Li-Cor) diluted at 1:10,000 in Intercept blocking for 1 h. All of the steps were performed at room temperature. Fluorescent signals were acquired using the Odyssey Fc Imaging System (Li-Cor) and quantified using ImageJ (v.2.0.0; NIH). Equal volumes of samples were loaded on the gels for control and peptide treatments (all of the raw gels are provided in Supplementary Fig. 1). Differences in total

MUC5AC signal result from differences in IL-13 induced metaplasia between individual filters. Stimulated secretion was therefore normalized to baseline secretion within individual filters to account for filter-to-filter heterogeneities.

To account for donor heterogeneity, all of the relevant experiments were performed in HAE cell ALI cultures generated from at least four individual donors. Complete sets of control and experimental conditions were conducted in ALI cultures from the same donor. Donors were selected randomly from our depository. Individual ALI cultures from the same donor were then randomly allocated to a control treatment groups. Thus, covariates including sex, age and clinical history were identical in all of the conditions. No blinding was performed. Donor numbers are indicated in the respective figure legends. Donor participant sex, age and smoking status is listed in Supplementary Table 3.

### Immunofluorescence staining in HAE cells for CPP uptake experiments

HAE cells grown on Transwell filters were incubated with 20 µl of DMEM ± 100 µM specified peptides (Fig. 4a) on day 28 of establishing ALI. Then, 24 h later, cells were fixed for 20 min in 2% paraformaldehyde in DPBS. Cells were then permeabilized for 10 min with 0.2% saponin and 10% FBS (Thermo Fisher Scientific) in DPBS. Cells were washed twice with DPBS and stained with anti-MUC5AC (45M1, MA1-21907, Thermo Fisher Scientific) antibodies diluted 1:100 in DPBS, 0.2% saponin and 10% FBS overnight at 4 °C. Subsequently, cells were washed twice with DPBS and incubated for 1 h at room temperature in DPBS, 0.2% saponin and 10% FBS containing AlexaFluor-488-labelled anti-mouse secondary antibodies (1:500; Thermo Fisher Scientific) and DAPI (1:5,000; Thermo Fisher Scientific). Images were taken on an inverted confocal microscope (Leica TCS SP5) using a ×40 lens (Leica HC PL APO CS2 40x1.30 OIL). Images for the blue (DAPI), green (AlexaFluor 488) and red (Cy3) channels were taken in sequential mode using appropriate excitation and emission settings.

### Image analysis for analysis of CPP uptake in HAE cells

Serial sections of images along the basolateral to apical cell axis ($z$ axis) were acquired with a 0.28 µm distance between individual $z$-sections to analyse the distribution of intracellular Cy3 fluorescence (Extended Data Fig. 8). Fluorescence intensity profiles along the $z$ axis in individual cells were calculated for all of the channels using the Lecia LAS X software (Leica). In brief, fluorescence intensities of DAPI, AlexaFluor 488 (MUC5AC) and Cy3 were analysed within individual cells at each $z$-section, normalized and fluorescence intensity traces were calculated along the basolateral to apical cell axis. Traces were exported to GraphPad Prism 7 for graph plotting. For quantitative analysis of intracellular Cy3 fluorescence intensities, maximum projections of all $z$-sections were calculated using the Leica LAS X software and average fluorescence intensities were analysed for individual MUC5AC$^+$ cells. Experiments to analyse peptide uptake were performed in HAE cell ALI cultures from two individual donors and complete sets of experimental conditions were conducted in ALI cultures from both donors.

For binding of biotin–SP9–Cy3 to bacterial toxins, C2 and CRM197 were conjugated to streptavidin. Biotin–SP9–Cy3 and streptavidin-conjugated toxins were mixed at a 10:1 ratio at 30 °C for 30 min before adding to cells.

### Quantification and statistical analysis

Origin, MATLAB and Prism were used to generate all curves and graphs. The fusion experiments were conducted at least three times with different protein preparations and vesicle reconstitutions, and properties were calculated as the mean ± s.e.m. Two-tailed Student's $t$-tests were used to test statistical significance in Figs. 1, 3 and 5 and Extended Data Figs. 4, 5 and 7 with respect to the specified reference experiment. Statistical significance in Fig. 4 and Extended Data Fig. 8 was assessed using ANOVA followed by post hoc Dunnett's test or by two-tailed Student's $t$-tests, where appropriate.

Box plots are defined as follows: the whiskers show the minimum and maximum values (excluding outliers), the box limits show the 25% and 75% percentiles, the square point denotes the mean, and the centre line denotes the median.

### Software and code

The HAE cell data collection was performed using the Leica LAS X v3.1.5.16308, Li-Cor Odyssey Fc Imaging System v.5.2. The data for the single-vesicle fusion experiments and single molecule counting experiments were collected by the smCamera program developed by T. Ha.

Data analysis was performed for the HAE cell experiments using MS Excel for Mac v.16.36, GraphPad Prism v.7 and NIH ImageJ v.2.0.0-rc69. For the mouse experiments, ImagePro-5.1 (Media Cybernetics) was used. For the single-vesicle fusion experiments, single-molecule counting experiments, fluorescence anisotropy experiments and circular dichroism experiments, OriginPro 8 and MATLAB-2021b were used. EMAN2-2.91 was used to analyse the Cryo-EM images in Extended Data Fig. 6a. NAMD2 v.2.14b1 was used for the molecular dynamics simulations. Pymol v.2.5.1 was used for modelling mutations and visualization.

### Animal statement

All the mouse work was conducted in accordance with the UT MD Anderson Cancer Center IACUC guidelines, and under the IACUC supervision; protocol no. 00001214-RN02.

### Reporting summary

Further information on research design is available in the Nature Research Reporting Summary linked to this paper.

## Data availability

The imaging data for the single-vesicle fusion experiments, the studies of primary HAE cells, the mucin-secretion and airway mucus occlusion studies in mice, and the NAMD input files and trajectories of the molecular dynamics simulations are available in the Dryad repository https://doi.org/10.5061/dryad.dz08kprz7. Full versions of the blots are provided in Supplementary Fig. 1. Source data are provided with this paper.

## Code availability

MATLAB analysis scripts for the single-vesicle fusion, single-molecule counting experiments and smCamera file conversions are available in the Zenodo repository https://doi.org/10.5281/zenodo.6370585.

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

**Acknowledgements** We thank R. Adachi, M. Singh, T. Südhof, T. Velkov and C. Wang for discussions; T. Südhof for providing the *Syt2* conditional knockout mice; Z. Azzegagh and V. Oren for technical assistance with mucin secretion in mice; H. Barth and M. Fellermann for providing bacterial toxins (C2 and CRM197); P. Braubach for providing donor tissue; C. Schilpp for isolation of HAE cells; S. Lynch for performing the NMR experiments for SP9; H. Chong and colleagues at Vivitide for their expertise in peptide synthesis and collaborative efforts; K. Marcus for providing script files for the molecular dynamics simulations; and C. Mayne for providing parameters for the molecular dynamics simulations. We acknowledge the Deutsche Forschungsgemeinschaft (DFG, German Research Foundation) (251293561 (SFB1149) and 175083951, to M.F.), the National Institutes of Health (NIH) (R37MH63105 to A.T.B.; R01 HL129795 and R21 AI137319 to B.F.D.) and the Cystic Fibrosis Foundation (DICKEY18G0 and DICKEY19P0) for support. The contents of this publication are solely the responsibility of the authors and do not necessarily represent the official views of NIGMS or NIH.

**Author contributions** Y.L., G.F., M.F., M.J.T., B.F.D. and A.T.B. designed experiments. Y.L. performed fusion, binding and CD experiments. J.L. and J.P. wrote MATLAB scripts to analyse the fusion experiments. G.F. and M.F. performed HAE cell experiments. Y.L., M.J.T., P.J. and Q.Z. performed inhibitor design. K.Y. performed EM experiments. K.Y. and J.L. performed single-molecule counting experiments, and Y.L. and Y.Z. analysed these data. R.A.P. and L.E. assisted with protein purification. M.J.T. performed mouse *Syt2* knockout experiments. J.R.F. performed mouse experiments and analyses. Y.L., G.F., M.F., B.F.D. and A.T.B. wrote the manuscript.

**Competing interests** The authors declare no competing interests.

**Additional information**
**Correspondence and requests for materials** should be addressed to Ying Lai, Manfred Frick, Burton F. Dickey or Axel T. Brunger.

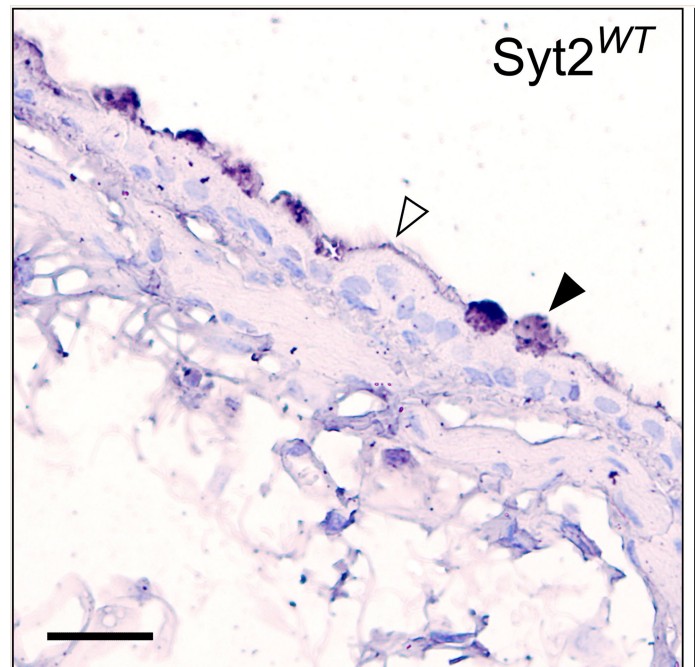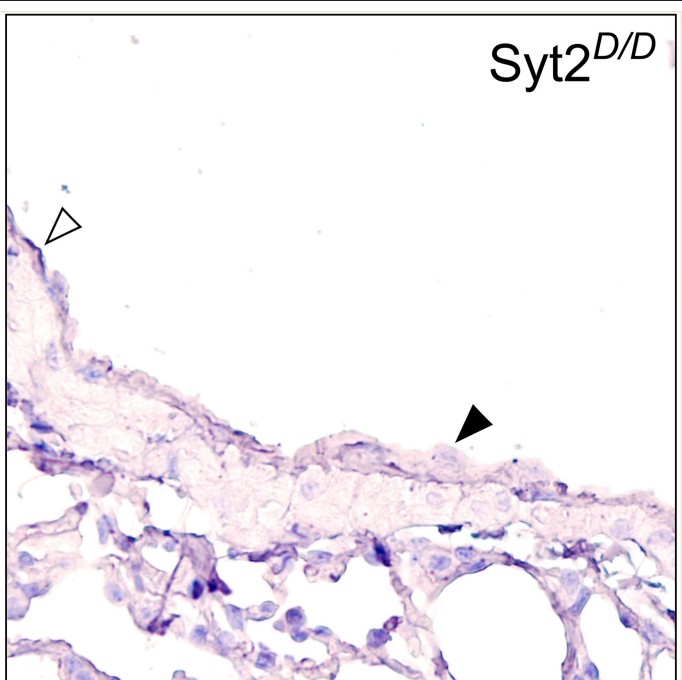

**Extended Data Fig. 1 | Related to Figure 1. Efficiency of Syt2 deletion in airway secretory cells.** Bronchial airways of Syt2$^{WT}$ and Syt2$^{D/D}$ mice were stained with antibodies to Syt2 and secondary antibodies conjugated to horseradish peroxidase (brown colour, see Methods). Secretory cells in Syt2 WT mice (left panel) have a domed appearance with their apical poles staining intensely for Syt2 (filled triangle). There was less intense linear staining of tufted ciliated cells in the region of ciliary basal bodies (open triangle), which often stain non-specifically. Secretory cells in Syt2$^{D/D}$ mice did not stain for Syt2 (filled triangle), but linear staining of ciliated cells was similar to that in Syt2$^{WT}$. We enumerated secretory cells in 3 mice of each genotype and found 46% in Syt2$^{WT}$ and 48% in Syt2$^{D/D}$, ±4%, which did not differ significantly, indicating there was no loss of viability of secretory cells in Syt2$^{D/D}$ mice. In Syt2$^{WT}$ mice, 91% of secretory cells stained for Syt2, whereas in Syt2$^{D/D}$ mice, only 7% stained for Syt2, indicating a deletion efficiency ~92%. Results for Syt2$^{F/F}$ mice were indistinguishable from those for Syt2$^{WT}$. Scale bar, 50 μm. Experiments were repeated twice with similar results.

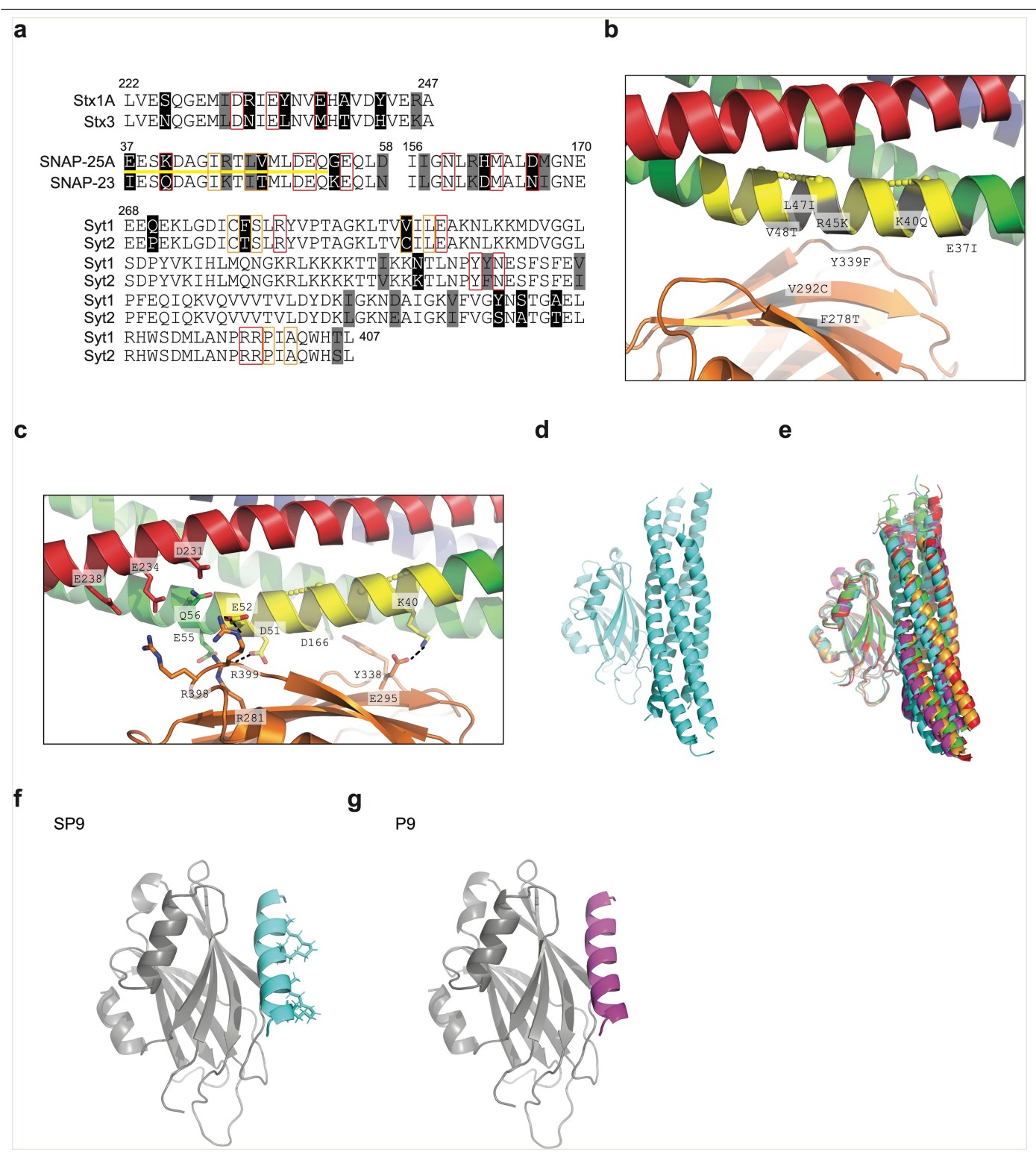

**Extended Data Fig. 2 | Related to Figure 2. The conservation of the primary interface. a**, Primary sequence alignments between neuronal and airway systems (Stx1A vs. Stx3, SNAP-25A vs. SNAP-23, and Syt1 vs. Syt2). White: absolutely conserved, grey: similar, black: not conserved. Red boxes indicate residues involved in salt bridges and hydrogen bonds, orange boxes indicate residues involved in hydrophobic interactions in the primary interface. The yellow line indicates the residues of SP9 shown as the yellow region in panel **b** and **c. b**, Close-up view of the primary interface (PDB ID 5W5C) with grey colour and labels indicating the locations of sequence differences in the primary interface between the neuronal and airway epithelial systems (SNAP-25A vs. SNAP-23, Syt1 vs. Syt2); the corresponding labels indicate the sequence

differences. Yellow: region that corresponds to SP9 with staples shown as dumbbells. **c**, Close-up view of the primary interface with residues shown as sticks that are important for the primary interface, including R281, E295, Y338, R398, R399 in Syt1 C2B (also corresponding to residues mutated in Syt1(QM)) and K40, D51, E52, E55, Q56, D166 in SNAP-25A and D231, E234, E238 in Stx1A. Yellow: region that corresponds to SP9 with staples shown as dumbbells. **d**, Starting point of the molecular dynamics simulations of the primary interface. **e**, End points of five independent 1-µsec simulations (colours) of the primary interface. **f**, Starting point of the SP9–Syt1-C2B simulations. **g**, Starting point of the P9–Syt1-C2B simulations. All starting points were derived from the crystal structure with PDB ID 5W5C.

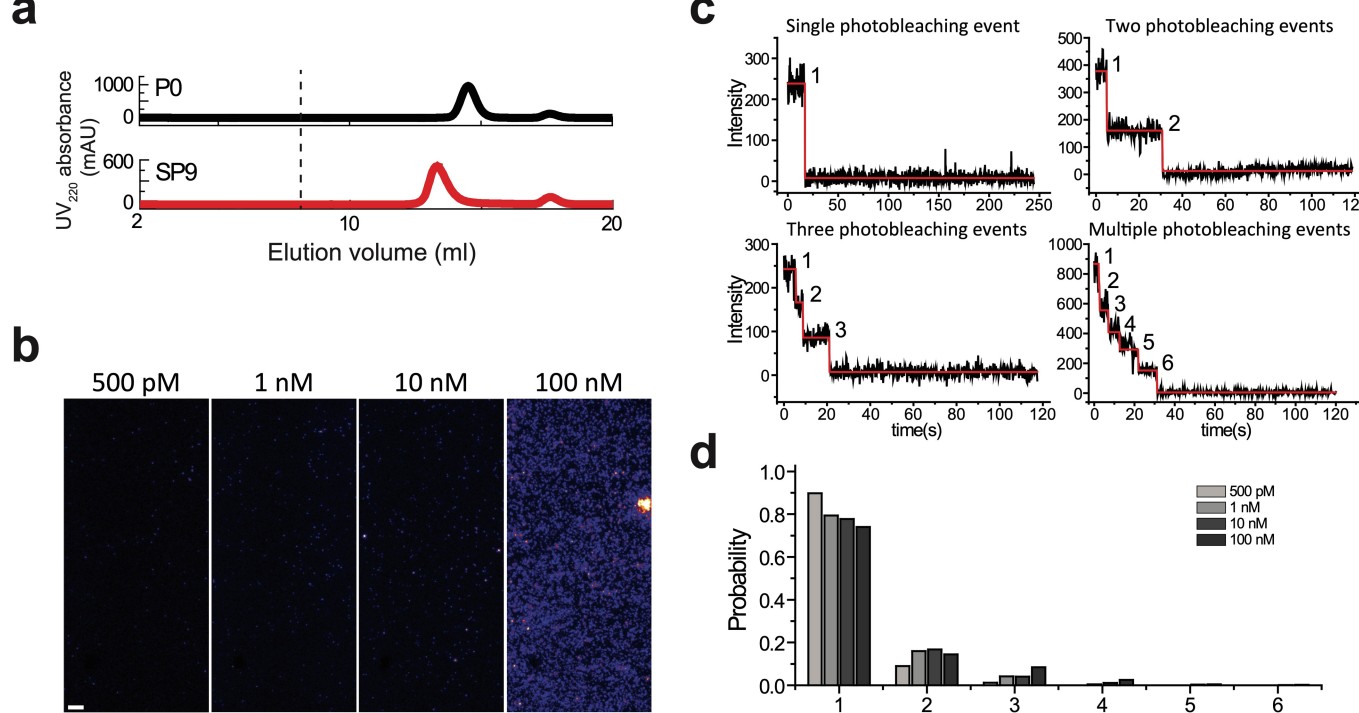

**Extended Data Fig. 3 | Related to Figure 2. Characterization of the oligomeric state of SP9. a**, Size exclusion chromatography (SEC) profiles of peptides. Each peptide was filtered with a 0.2 micrometer filter and then loaded on a Superdex 75 column in buffer V (20 mM HEPES, pH 7.4, 90 mM NaCl). The dashed line indicates the border of the void volume at ~8 ml. The difference in retention times for P0 and SP9 may be related to the conformations of the peptides. **b**, Representative TIRF images of immobilized SP9-Cy3 at specified concentrations. Scale bar, 410 μm. **c**, Representative time traces showing single-molecule stepwise photobleaching events of SP9-Cy3.

Black lines correspond to the fluorescence intensity of SP9-Cy3 and red lines correspond to the idealized trajectory obtained by Hidden Markov Model analysis (HMM) (Methods). **d**, Distribution of multiple SP9-Cy3 molecules in (diffraction limited) fluorescent spots at specified concentrations. Fluorescent spots were automatically selected by smCamera. The number of SP9-Cy3 molecules per fluorescent spot was determined from the observed fluorescence intensity time traces by HMM and verified by manual inspection (Methods). Bar graphs were calculated from 167, 675, 937, and 520 selected traces at concentrations of 0.5, 1, 10, and 100 nM SP9-Cy3, respectively.

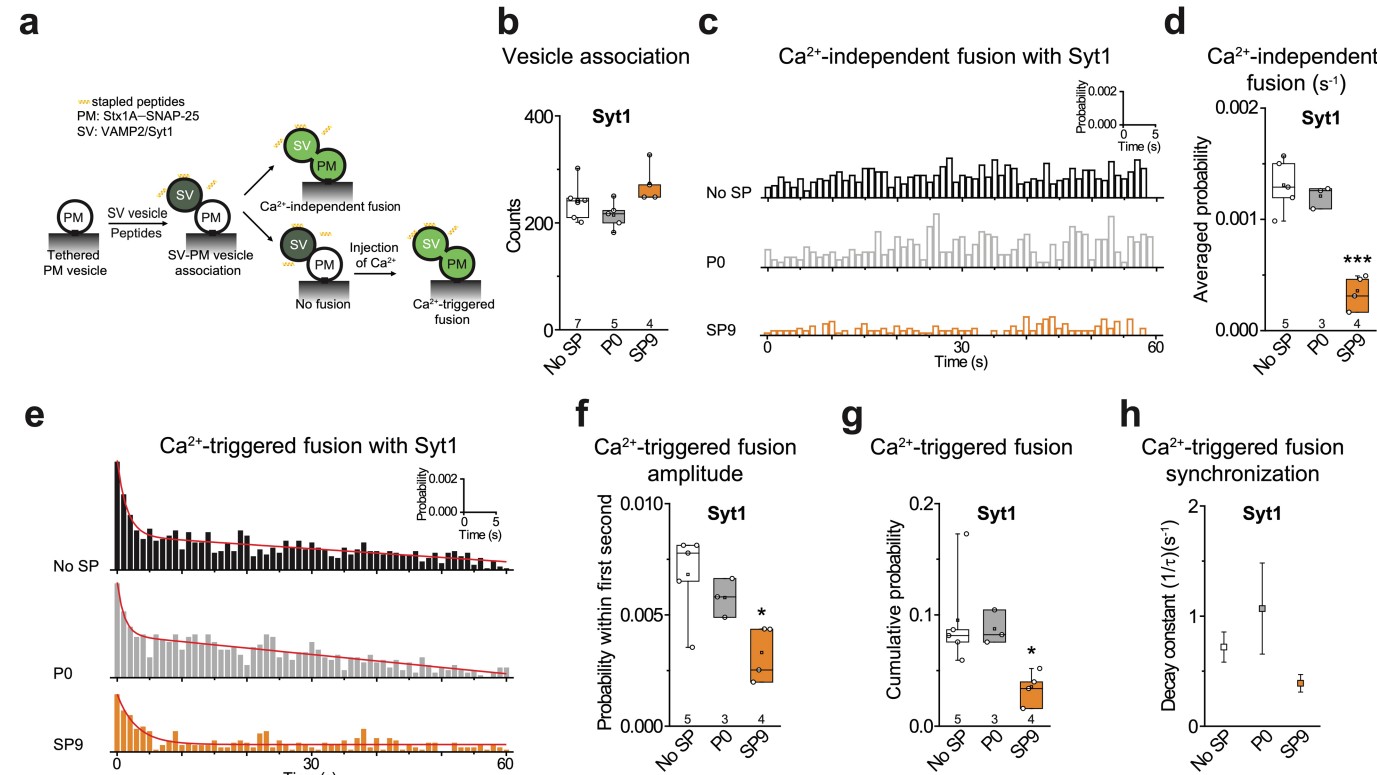

**Extended Data Fig. 4 | Related to Figure 3. SP9 inhibits both Ca²⁺-independent and Ca²⁺-triggered vesicle fusion with reconstituted neuronal SNAREs and Syt1. a**, Schematic of the single vesicle content mixing assay. Neuronal PM: plasma membrane mimic vesicles with reconstituted Stx1A and SNAP-25A; SV: synaptic vesicle mimic with reconstituted VAMP2 and Syt1. After SV - neuronal PM vesicle association, vesicle pairs either undergo Ca²⁺-independent fusion or remain associated until fusion is triggered by Ca²⁺ addition. 10 µM of P0 or SP9 was added together with SV vesicles and was present in all subsequent stages. **b**, Effect P0 and SP9 on vesicle association. **c**, Corresponding Ca²⁺-independent fusion probabilities. **d**, Corresponding average probabilities of Ca²⁺-independent fusion events per second (*** p = 0.00022). **e**, Corresponding Ca²⁺-triggered fusion probabilities. **(f–h)**

Corresponding Ca²⁺-triggered fusion amplitudes of the first 1-sec time bin upon 500 µM Ca²⁺-injection (**f**) (* p = 0.017), the cumulative Ca²⁺-triggered fusion probability within 1 min (**g**) (* p = 0.039), and the decay rate (1/τ) of the Ca²⁺-triggered fusion histogram (**h**). The fusion probabilities and amplitudes were normalized to the number of analysed neuronal SV - neuronal PM vesicle pairs (Supplementary Table 2). Panels **b**, **d**, **f**, **g** show box plots and data points for n (indicated below each box plot) independent repeat experiments (Supplementary Table 2). Two-tailed Student's t-tests were used for SP9 vs. No SP. Decay constants (boxes) and error estimates (bars) in panels **h** computed from the covariance matrix upon fitting the corresponding histograms combining all repeats with a single exponential decay function using the Levenberg-Marquardt algorithm.

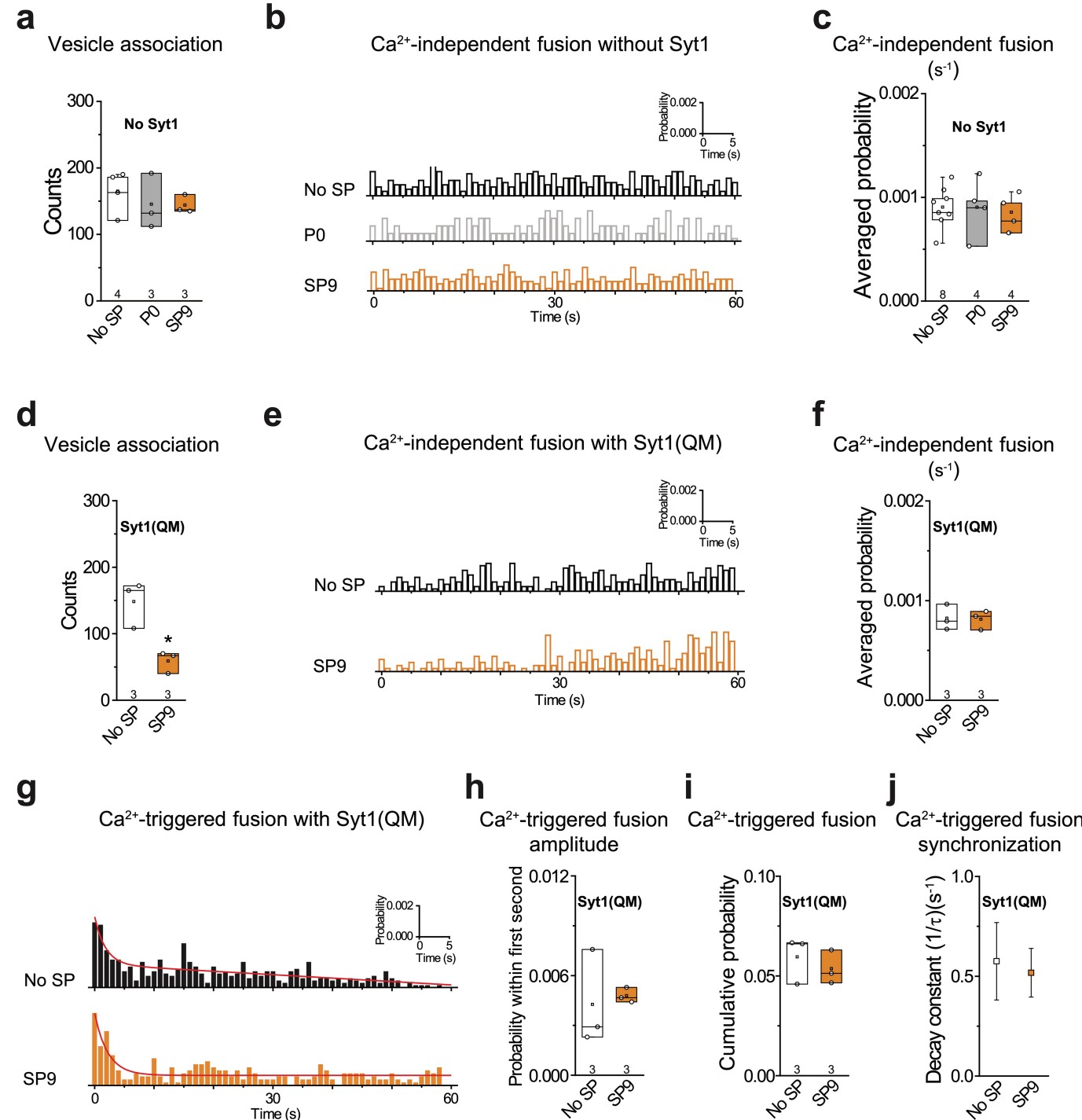

**Extended Data Fig. 5 | Related to Figure 3. SP9 has no effect on vesicle fusion mediated by neuronal SNAREs alone, or by Syt1(QM). a–c**, SP9 has no effect on vesicle fusion mediated by neuronal SNAREs alone. **a**, Effects of 10 μM of P0 or SP9 on vesicle association. **b**, Corresponding Ca²⁺-independent fusion probabilities. **c**, Corresponding average probabilities of Ca²⁺-independent fusion events per second. **d–j**, the quintuple Syt1(QM) mutant alleviates the inhibitory effect of SP9 on neuronal synaptic vesicle fusion. **d**, Effect of 10 μM of SP9 on vesicle association (* p = 0.01629). **e**, Corresponding Ca²⁺-independent fusion probabilities. **f**, Corresponding average probabilities of Ca²⁺-independent fusion events per second. **g**, Corresponding Ca²⁺-triggered fusion probabilities. (**h–j**) Corresponding Ca²⁺-triggered fusion amplitude of the first 1-sec time bin upon 500 μM Ca²⁺-injection (**h**), the cumulative Ca²⁺-triggered fusion probability within 1 min (**i**), and the decay rate (1/τ) of the Ca²⁺-triggered fusion histogram (**j**). The fusion probabilities and amplitudes were normalized to the number of analysed neuronal SV - neuronal PM vesicle pairs (Supplementary Table 2). Panels **a**, **c**, **d**, **f**, **h**, **i** show box plots and data points for n (indicated below each box plot) independent repeat experiments (Supplementary Table 2). Two-tailed Student's t-tests were used for SP9 vs. No SP. Decay constants (boxes) and error estimates (bars) in panels **j** computed from the covariance matrix upon fitting the corresponding histograms combining all repeats with a single exponential decay function using the Levenberg-Marquardt algorithm.

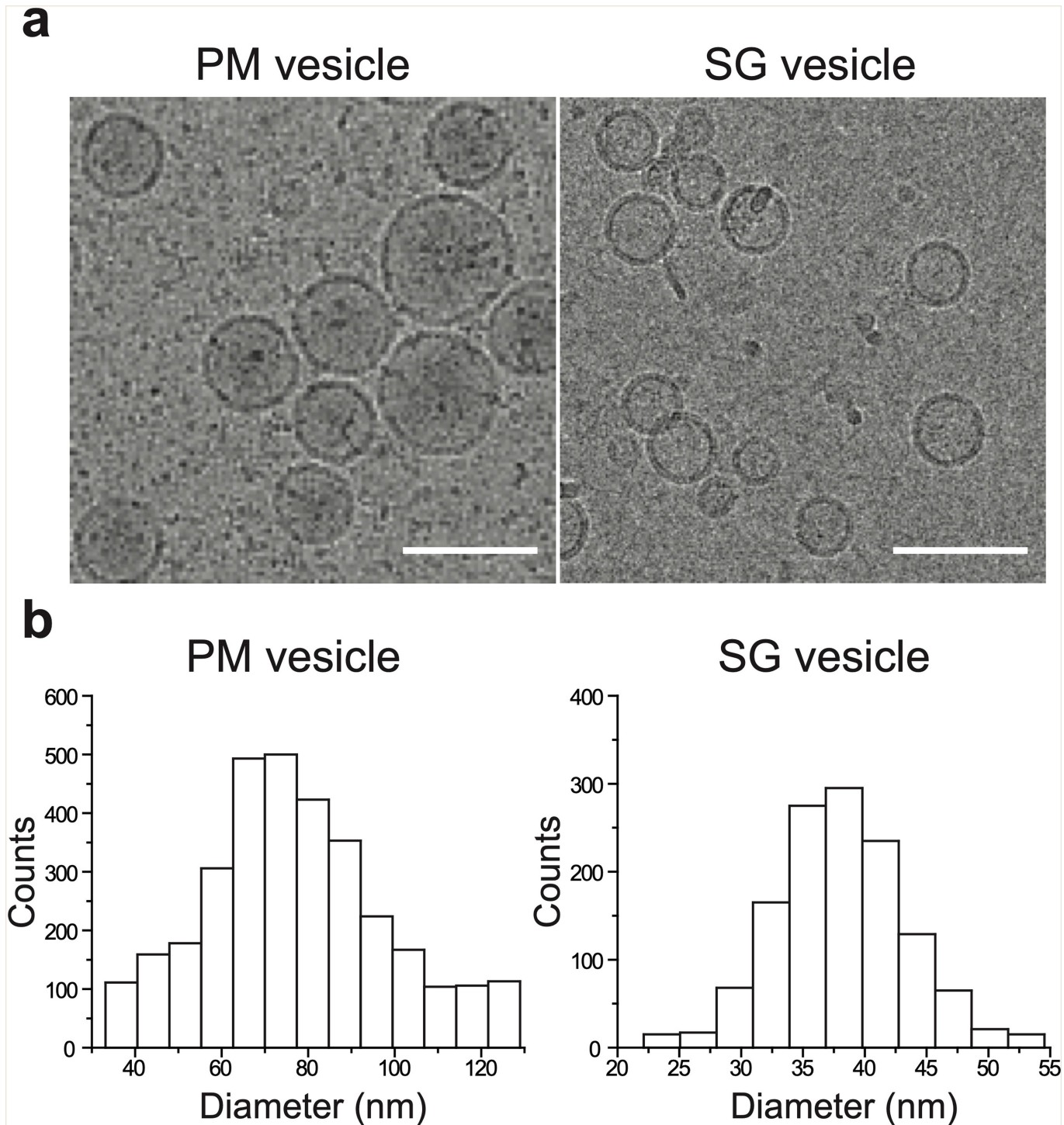

**Extended Data Fig. 6 | Related to Figure 3. Airway PM and SG vesicle preparation. a**, Cryo-EM images of airway PM and SG vesicles as defined in Methods. Scale bar, 100 nm. **b**, Diameter distributions for airway PM and SG vesicles.

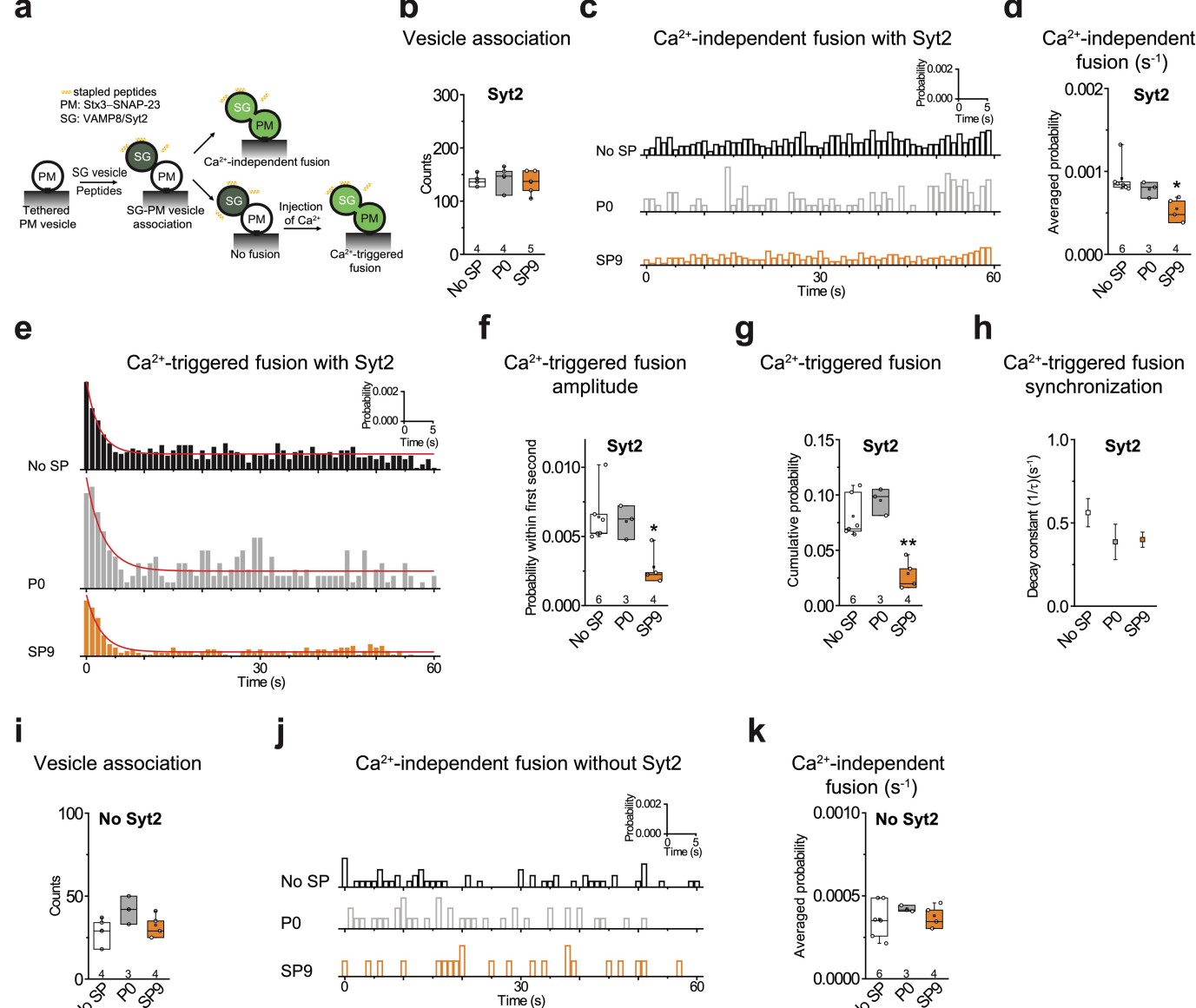

**Extended Data Fig. 7 | Related to Figure 3. SP9 inhibits Ca²⁺-triggered vesicle fusion with reconstituted airway epithelial SNAREs and Syt2.**
**a**, Schematic of the single vesicle content mixing assay. Airway PM: plasma membrane mimic vesicles with reconstituted airway Stx3 and SNAP-23; SG: secretory granule mimics with reconstituted VAMP8 and Syt2. After SG - airway PM vesicle association, vesicle pairs either undergo Ca²⁺-independent fusion or remain associated until fusion is triggered by Ca²⁺ addition. 10 μM of P0 or SP9 was added together with SG vesicles and was present during all subsequent stages. **b**, Effects of P0 or SP9 on vesicle association. **c**, Corresponding Ca²⁺-independent fusion probabilities. **d**, Corresponding average probabilities of Ca²⁺-independent fusion events per second (* p = 0.014). **e**, Corresponding Ca²⁺-triggered fusion probabilities. (**f**–**h**) Corresponding Ca²⁺-triggered fusion amplitudes of the first 1-sec time bin upon 500 μM Ca²⁺-injection (**f**)

(* p = 0.012), the cumulative Ca²⁺-triggered fusion probability within 1 min (**g**) (** p = 0.0018), and the decay rate (1/τ) of the Ca²⁺-triggered fusion histogram (**h**). **i**–**k**, SP9 has no effect on vesicle fusion mediated by airway SNAREs alone. **i**, Effects of 10 μM of P0 or SP9 on vesicle association using the assay described above. **j**, Corresponding Ca²⁺-independent fusion probabilities. **k**, Corresponding average probabilities of Ca²⁺-independent fusion events per second. Panels **b**, **d**, **f**, **g**, **i**, **k** show box plots and data points for n (indicated below each box plot) independent repeat experiments (Supplementary Table 2). Two-tailed Student's t-tests were used for SP9 vs. No SP. Decay constants (boxes) and error estimates (bars) in panel **h** computed from the covariance matrix upon fitting the corresponding histograms combining all repeats with a single exponential decay function using the Levenberg-Marquardt algorithm.

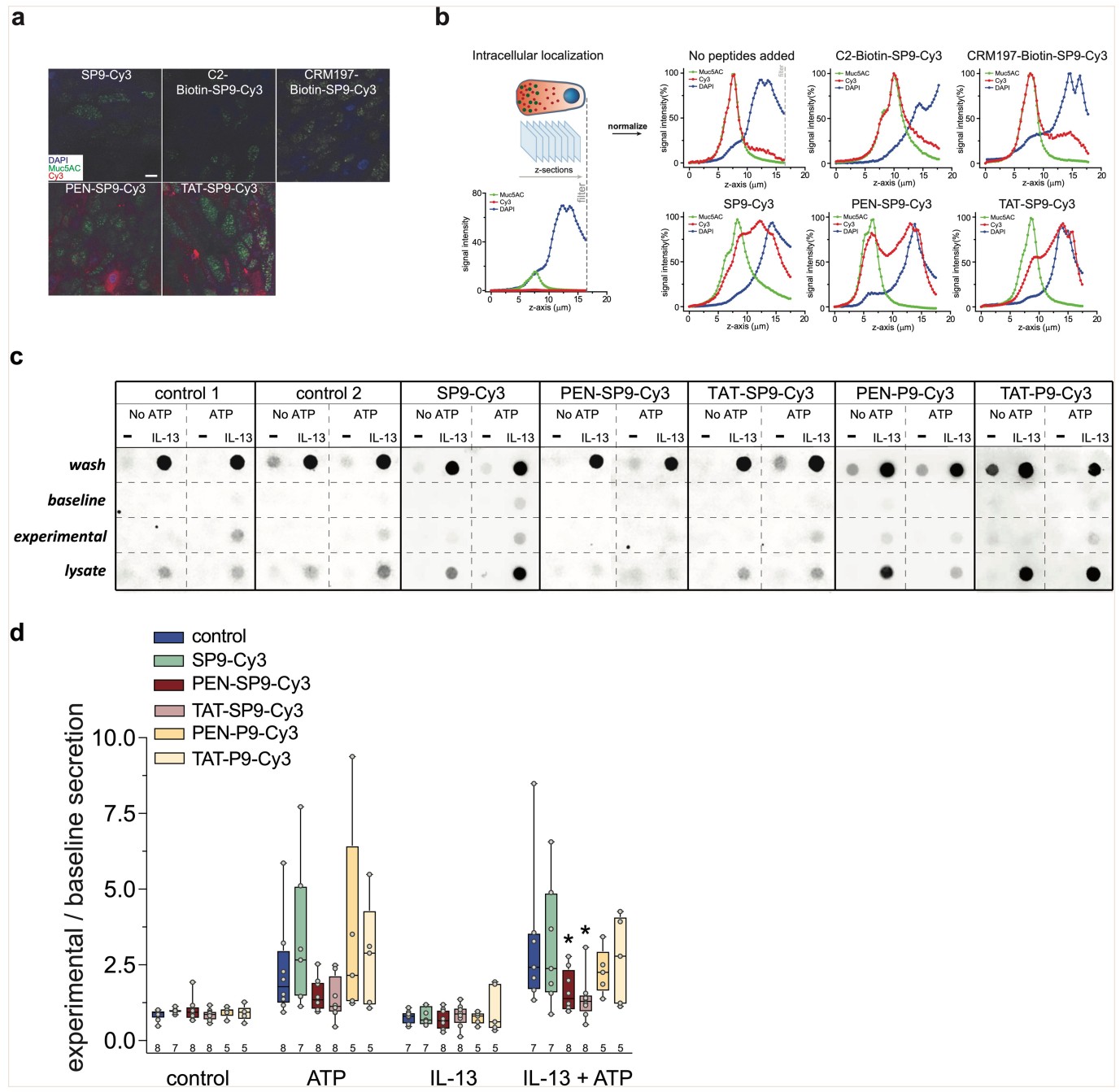

**Extended Data Fig. 8 | Related to Figure 4. SP9 penetrates epithelial cells when conjugated to CPPs and inhibit mucin secretion from airway epithelium cells. a**, Representative confocal images (z-sections) of fixed HAE cells treated with SP9-Cy3 or SP9-Cy3 conjugated to CPPs or biotin. Biotin-SP9-Cy3 was bound to streptavidin-conjugated C2 or CRM197. The experiment was repeated twice with ALI cultures from different donors with similar results. Scale bar, 10 μm. **b**, The diagram illustrates the analysis of intracellular localization of MUC5AC, Cy3, and DAPI in airway secretory cells. Fluorescence intensities of DAPI, AlexaFluor 488 (MUC5AC) and Cy3 were analysed within individual MUC5AC+ cells at each z-section, normalized and fluorescence intensity traces calculated along the basolateral to apical cell axis. **c**, Representative western blot immunofluorescence images for MUC5AC on apical surface of untreated HAE cells (control 1 and 2) or HAE cells treated with 100 μM SP9-Cy3, PEN-SP9-Cy3, TAT-SP9-Cy3, PEN-P9-Cy3, or TAT-P9-Cy3 for 24 h before stimulation. *Wash* represents MUC5AC accumulated during

culture and before start of experiment. *Baseline* represents unstimulated MUC5AC secretion during a 15 min period after removal of accumulated MUC5AC and *experimental* represents MUC5AC secreted within 15 min of stimulation with (ATP) or without (no ATP) 100 μM ATP. *Lysate* represents MUC5AC within HAE cells at the end of the experiment. Cells were treated with IL-13 to induce mucous metaplasia. All original blots are shown in Supplementary Fig 1b. **d**, Box plots and data points show the ratio of experimental / baseline secretion (fold increase of stimulated secretion over baseline secretion) following 24 h preincubation with 100 μM of the respective peptides. Numbers below box-plots indicate n for each condition, representing individual ALI cultures derived from 4 donors for each condition. * p = 0.046 for HAE cells treated with 100 μM PEN-SP9-Cy3, and p = 0.016 for HAE cells treated with 100 μM TAT-SP9-Cy3, assessed by two-way ANOVA followed by post-hoc Dunnett`s test.

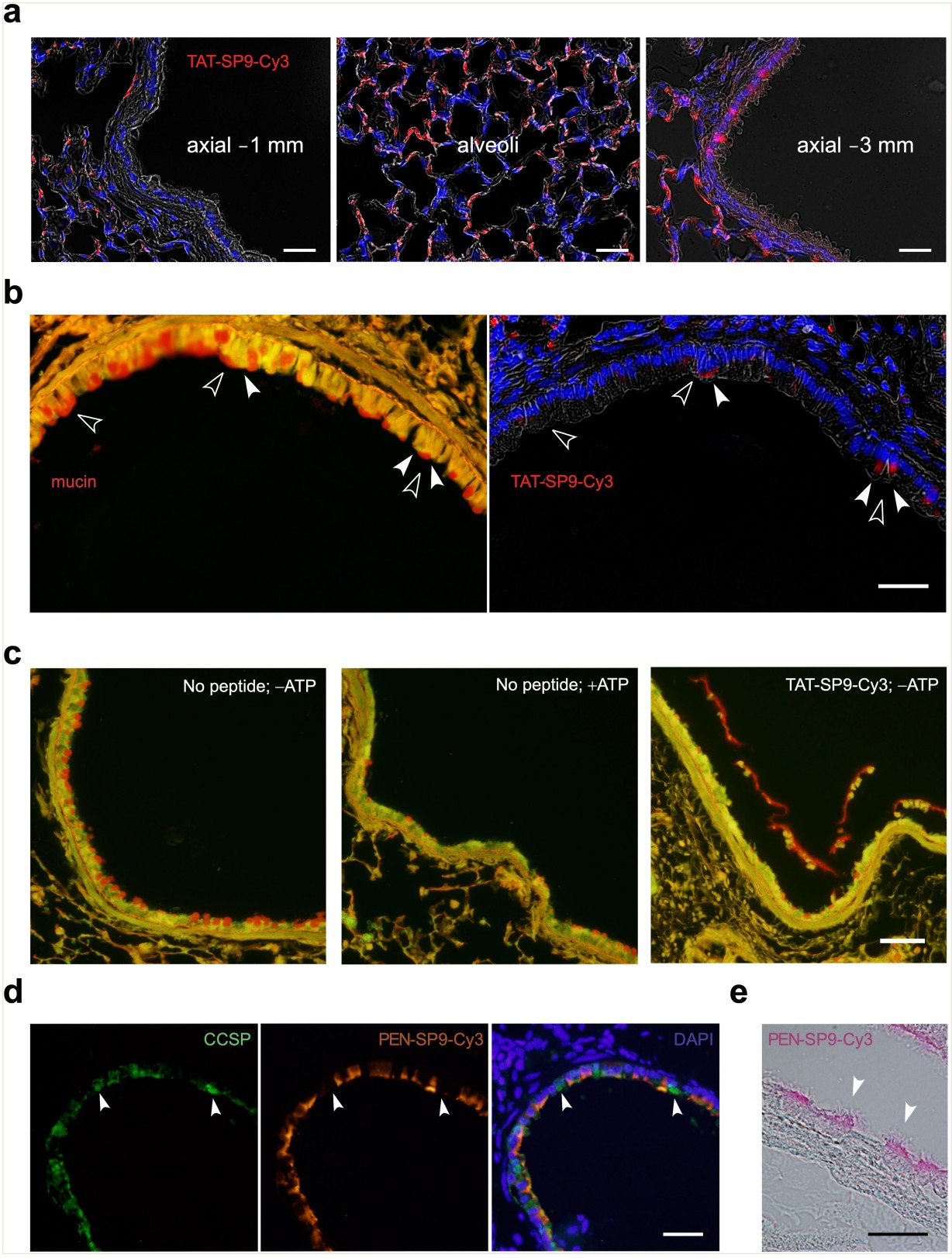

**Extended Data Fig. 9** | See next page for caption.

**Extended Data Fig. 9 | Related to Figure 5. Uptake of labelled SP9 conjugated to CPP into mouse airway epithelial cells. a**, Predominant delivery of 100 μM TAT-SP9-Cy3 to distal airways of mice using a microsprayer. The left panel shows a lack of uptake of the labelled peptide in the proximal axial bronchus, sectioned 1 mm below our usual site of transverse section between lateral branches 1 and 2, though uptake in alveoli surrounding the airway can be seen. The middle panel shows a section through the periphery of the lung, with extensive uptake of the labelled peptide in alveolar epithelial cells. The right panel shows patchy uptake of the labelled peptide in the distal axial bronchus, sectioned 3 mm below lateral branches 1 and 2. Scale bar, 50 μm. **b**, Two serial transverse sections of the left axial bronchus of a mouse with mucous metaplasia induced by prior instillation of IL-13, then treated with 300 μM aerosolized TAT-SP9-Cy3. The left panel is stained with PAFS (red) to demonstrate intracellular mucin, and shows some cells with high mucin content (open arrowheads) and other cells with low mucin content (closed arrowheads), presumably due to induced mucin secretion. The right panel shows blue fluorescent staining of nuclei with DAPI and red fluorescent staining of epithelial cells that have internalized TAT-SP9-Cy3. The same cells indicated in the left panel are also indicated in the right panel, showing that cells that internalize TAT-SP9-Cy3 tend to have low intracellular mucin content, possibly due to the cell-penetrating peptide conjugate allowing calcium entry into the cytoplasm to induce secretion. Scale bar, 20 μm. **c**, Sections of the left axial bronchus of mice with mucous metaplasia induced by prior instillation of IL-13, and not further treated (left panel), or subsequently treated with aerosolized 100 mM ATP (middle panel) or 1 mM TAT-SP9-Cy3 (right panel). These show high intracellular mucin content (red: PAFS stain) in the mice not treated with an aerosolized drug, extensive secretion of intracellular mucin in the mice treated with ATP, and extensive apocrine mucin secretion in the mice treated with TAT-SP9-Cy3, possibly due to disruption of mucin granule membrane integrity causing intracellular mucin swelling at this high peptide concentration. Scale bar, 50 μm. **d**, Transverse section of the left axial bronchus of a mouse taken 30 min after treatment with aerosolized 20 μM PEN-SP9-Cy3. Immunofluorescent staining for CCSP shows green secretory cells (arrowheads), red PEN-SP9-Cy3 fluorescence, and blue nuclei stained with DAPI. Green secretory cells are observed to not visibly take up PEN-SP9-Cy3, in contrast to ciliated cells, the other major airway epithelial cell type that is not labelled here with a lineage marker, but avidly internalize red PEN-SP9-Cy3 fluorescence . Scale bar, 50 μm. **e**, Section of the left axial bronchus of a mouse treated as in "**c**" (with PEN-SP9-Cy3), showing red fluorescent staining of ciliated cells with ciliary tufts clearly visible by differential interference microscopy (arrowheads), but no staining of intervening secretory cells. Scale bar, 20 μm. Similar results were obtained multiple times.

# Reporting Summary

Nature Research wishes to improve the reproducibility of the work that we publish. This form provides structure for consistency and transparency in reporting. For further information on Nature Research policies, see Authors & Referees and the Editorial Policy Checklist.

## Statistics

For all statistical analyses, confirm that the following items are present in the figure legend, table legend, main text, or Methods section.

| n/a | Confirmed | |
|---|---|---|
| ☐ | ☒ | The exact sample size (*n*) for each experimental group/condition, given as a discrete number and unit of measurement |
| ☐ | ☒ | A statement on whether measurements were taken from distinct samples or whether the same sample was measured repeatedly |
| ☐ | ☒ | The statistical test(s) used AND whether they are one- or two-sided *Only common tests should be described solely by name; describe more complex techniques in the Methods section.* |
| ☐ | ☒ | A description of all covariates tested |
| ☐ | ☒ | A description of any assumptions or corrections, such as tests of normality and adjustment for multiple comparisons |
| ☐ | ☒ | A full description of the statistical parameters including central tendency (e.g. means) or other basic estimates (e.g. regression coefficient) AND variation (e.g. standard deviation) or associated estimates of uncertainty (e.g. confidence intervals) |
| ☐ | ☒ | For null hypothesis testing, the test statistic (e.g. *F*, *t*, *r*) with confidence intervals, effect sizes, degrees of freedom and *P* value noted *Give P values as exact values whenever suitable.* |
| ☒ | ☐ | For Bayesian analysis, information on the choice of priors and Markov chain Monte Carlo settings |
| ☒ | ☐ | For hierarchical and complex designs, identification of the appropriate level for tests and full reporting of outcomes |
| ☒ | ☐ | Estimates of effect sizes (e.g. Cohen's *d*, Pearson's *r*), indicating how they were calculated |

*Our web collection on statistics for biologists contains articles on many of the points above.*

## Software and code

Policy information about availability of computer code

| | |
|---|---|
| Data collection | The HAEC data collection was performed with Leica LAS X v3.1.5.16308, Li-Cor® Odyssey® Fc Imaging System v5.2. The data for the single vesicle fusion experiments and single molecule counting experiments were collected by smCamera developed by Taekjip Ha, Johns Hopkins University, Baltimore. |
| Data analysis | Data analysis was performed for the HAEC experiments with MS Excel for Mac v16.36, GraphPad Prism 7, NIH ImageJ v.2.0.0-rc69. For the mouse experiments, ImagePro 5.1 (Media Cybernetics) was used. For the single vesicle fusion experiments, single molecule counting experiments, fluorescence anisotropy experiments, and circular dichroism experiments, OriginPro 8, Matlab-2021b were used. EMAN2-2.91 was used to analyze the Cryo-EM images in Extended Data Fig. 6a. NAMD2 2.14b1 was used for the molecular dynamics simulations. <br><br> Matlab analysis scripts for the single vesicle fusion, single molecule counting experiments, and smCamera file conversions are available in the Zenodo repository https://doi.org/10.5281/zenodo.637058 . |

For manuscripts utilizing custom algorithms or software that are central to the research but not yet described in published literature, software must be made available to editors/reviewers. We strongly encourage code deposition in a community repository (e.g. GitHub). See the Nature Research guidelines for submitting code & software for further information.

## Data

Policy information about availability of data

All manuscripts must include a data availability statement. This statement should provide the following information, where applicable:

- Accession codes, unique identifiers, or web links for publicly available datasets
- A list of figures that have associated raw data
- A description of any restrictions on data availability

The imaging data for the studies of primary human airway epithelial cells (HAECs), for the mucin secretion and airway mucus occlusion studies in mice, and the NAMD input files and trajectories of the molecular dynamics simulations are available in the Dryad repository doi:10.5061/dryad.dz08kprz7 . Full versions of blots

are provided in Supplementary Fig. 1. Excel spreadsheet files with all data points and analyses are provided as Source Data (Fig1.xlsx, Fig2.xlsx, Fig3.xlsx, Fig4.xlsx, Fig5.xlsx, ED_Fig3.xlsx, ED_Fig4.xlsx, ED_Fig5.xlsx, ED_Fig6.xlsx, ED_Fig7.xlsx, ED_Fig8.xlsx). A model of SP9 is provided as Source Data (SP9.pdb), and endpoints of the molecular dynamics simulations of Syt1 C2B : SP9 and Syt1 C2B : P9 are provided as Source Data (SP9_simulations.pdb, P9_simulations.pdb, respectively).

# Field-specific reporting

Please select the one below that is the best fit for your research. If you are not sure, read the appropriate sections before making your selection.

☒ Life sciences ☐ Behavioural & social sciences ☐ Ecological, evolutionary & environmental sciences

For a reference copy of the document with all sections, see nature.com/documents/nr-reporting-summary-flat.pdf

# Life sciences study design

All studies must disclose on these points even when the disclosure is negative.

| | |
|---|---|
| Sample size | For HAEC experiments peptide and control treatments were conducted within ALI cultures from the same donor. ALI cultures were prepared from 2 donors for peptide internalisation assays and at least 4 different donors for each set of mucin secretion experiments (i.e., different doses, incubation times). 4 donors were considered sufficient to account for donor variation in mucin secretion assays. Donor subject sex, age and smoking status is listed in Supplementary Table 2. Sample size for HAEC experiments was based on previous experience and common practice (this included repeating all experiments in > 3 individual donors for mucin secretion assays). For studies with mice (Fig. 1, 5, Extended Data Fig. 1, 9), two independent experiments were combined to give a total of 10-11 mice per group. For single vesicle fusion experiments (Fig. 3, Extended Data Fig. 4, 5, 7) see Supplementary Table 1 for details about sample size and measurements. |
| Data exclusions | For HAEC experiments, ALI cultures were excluded when ALI was disturbed on day of experiment (i.e. epithelial leakage) and in very rare cases when analysis of dot blot signal was impaired. For single vesicle content mixing experiments, PM vesicles with multiple associated SV, SG, VAMP2, VAMP8 vesicle were excluded. |
| Replication | For HAEC experiments analysing mucin secretion all experiment were conducted in individual ALI filters coming from a miniumum of 4 individual donors, n numbers represent the number of ALI filters. For peptide internalisation experiments in HAEC experiments were repeated twice in ALI cultures from 2 donors. For experiments with mice (Figure 1 and 5), twice. For the single vesicle fusion experiments ((Fig. 3, Extended Data Fig. 4, 5, 7) see Supplementary Table 1 for details for details about repeats. |
| Randomization | For HAEC experiments, donors were selected randomly from our depository. Individual ALI cultures from the same donor were then randomly allocated to control an treatment groups. Control experiments and all experimental conditions were conducted in samples (ALI cultures) from the same donor. Hence, covariates including sex, age, clinical history were identical in all conditions. For animal experiments, mice from appropriate genotypes were randomly assigned to groups for all the conditions. |
| Blinding | For HAEC experiments, none. The entire analysis pipeline to quantify blot intensities was defined upfront (i.e. ROI size and location for detection of signal areas and background) and maintained identically for all analyses. For animal experiments investigators were blinded to the mouse group allocation during data collection and analysis. Also, mouse airway images were analyzed by investigators blinded to animal's genotype and treatment. |

# Reporting for specific materials, systems and methods

We require information from authors about some types of materials, experimental systems and methods used in many studies. Here, indicate whether each material, system or method listed is relevant to your study. If you are not sure if a list item applies to your research, read the appropriate section before selecting a response.

## Materials & experimental systems

| n/a | Involved in the study |
|---|---|
| ☐ | ☒ Antibodies |
| ☒ | ☐ Eukaryotic cell lines |
| ☒ | ☐ Palaeontology |
| ☐ | ☒ Animals and other organisms |
| ☒ | ☐ Human research participants |
| ☒ | ☐ Clinical data |

## Methods

| n/a | Involved in the study |
|---|---|
| ☒ | ☐ ChIP-seq |
| ☒ | ☐ Flow cytometry |
| ☒ | ☐ MRI-based neuroimaging |

## Antibodies

| | |
|---|---|
| Antibodies used | Anti-MUC5AC Mouse Monoclonal Antibody clone 45M1; cat.# MA1-21907, Thermo Scientific (dilution: 1:100 for immunocytochemistry; 1:250 for dot blots)<br>AlexaFluor 488-labelled anti-mouse secondary antibody, cat.# A-11001, Thermo Scientific (dilution 1:500)<br>IRDye® 800CW Donkey anti-Mouse IgG Secondary Antibody, cat.# 926-33212, Li-Cor (dilution 1:10000)<br>IRDye® 680RD Donkey anti-Mouse IgG Secondary Antibody , cat.# 926-68072, Li-Cor(dilution 1:10000) |

Anti-Syt2 rabbit polyclonal antibody ab113545 from Abcam, validated by manufacturer, confirmed by experimentation on the Syt2-deletant tissues
Goat Anti-Rabbit (HRP-conjugated) ab205718 from Abcam
THE HRP ANTIBODY IS NOT MENTIONED IN THE TEXT, THE ANTI-CCSP ANTIBODY (EXT FIG 9D) IS NOT LISTED HERE

Validation

The specificity of the commercially available anti-Muc5AC Antibody from Thermo scientific has been validated in many different publications (available on vendor website) and it has been used on human samples with many different techniques (IF,WB,Elisa). Validation of anti-Mucin and anti-Syt2 antibodies in mice was done using the relevant mouse knockouts. A

# Animals and other organisms

Policy information about studies involving animals; ARRIVE guidelines recommended for reporting animal research

Laboratory animals

Mice:
Mutant mice were received on a 129/Sv:C57BL/6 background, as referenced in the manuscript. They were subsequently backcrossed 10 times to C57BL/6J background to match the standard C57BL/6J wild type mice.

The animals were housed in specific pathogen-free conditions on a 12-hr light/dark cycle with food and water ad libitum. The number of animals used was the minimum that is consistent with scientific integrity and regulatory acceptability, consideration having been given to the welfare of individual animals in terms of the number and extent of procedures to be carried out on each animal.

Wild animals

No wild animals were used in this study.

Field-collected samples

No field-collected samples were used in this study.

Ethics oversight

All the mice work was conducted in accordance with the UT MD Anderson Cancer Center Institutional Animal Care and Use Committee (IACUC) guidelines, and under the IACUC supervision; protocol No 00001214-RN02.

Note that full information on the approval of the study protocol must also be provided in the manuscript.

