## [Peer Review File · Nature]

Manuscript Title: Inhibition of Ca²⁺ 1 -triggered secretion by hydrocarbon-stapled peptides

Reviewer Comments & Author Rebuttals

Reviewer Reports on the Initial Version:

Referees' comments:

Referee #1 (Remarks to the Author):

The authors take, as a point of departure, their 2015 Nature paper reporting the structure of a neuronal SNARE complex bound to the calcium sensor synaptotagmin. This structure revealed that the primary interface between the SNAREs and synaptotagmin was composed mostly – on the SNARE side – of a segment of the N-terminal SNAP-25A SNARE domain. The idea here is to inhibit the SNARE-synaptotagmin interaction by competition with a peptide version (SP9) of this SNAP-25A region. SP9 is ‘stapled’ by introducing two i, i+4 crosslinks, locking it into a helical conformation, and binds synaptotagmin with a K_d of about 30 μM. The inhibitory function of SP9 is validated in various assays using neuronal proteins and (more importantly for this work) in a rather complete reconstitution system for airway secretory granule fusion. Finally, by attaching a cell-penetrating peptide to SP9, the authors were able to demonstrate its ability to reduce mucin secretion/mucus occlusion in cultured human cells/mouse airway epithelium. Encouragingly, baseline mucin secretion – which might cause toxicity – was unaffected. Given the important role of stimulated mucus hypersecretion in respiratory disease, this evidence of SP9’s therapeutic potential is intriguing.

1. Although the “SP9 regions” of SNAP-25A (required for neurotransmitter release) and SNAP-23 (required for mucin secretion) are similar, they differ in at least one residue that contacts synaptotagmin (Fig. S2a). It would therefore seem to be a high priority to test ‘SNAP-23 SP9’ for inhibitory activity in vitro and in vivo.

2. A surprising and somewhat concerning aspect of this work is that SP9 oligomerizes as judged by Fig. S3. Fig. S3 also suggests a correlation between oligomerization and efficacy. Have the authors ruled out the possibility that the monomers are covalently crosslinked? Are the oligomers SNARE-bundle-like tetramers? Is oligomerization important for efficacy? Mass spectrometry and analytical ultracentrifugation should easily answer the first two questions. Answering the third question might be trickier but perhaps the authors have already figured out how to crystallize SP9?

3. The authors write (p. 5) that, “from a therapeutic perspective, peptide-based strategies have been successfully applied to inhibit virus-host membrane fusion. However, the molecular mechanisms are quite different since viral membrane fusion is mediated by formation of a six-helix bundle whereas Ca²⁺-triggered membrane fusion is mediated by formation of four-helix SNARE bundles in cooperation with synaptotagmins and other factors.” In my view, the biggest difference is that peptide-based viral inhibitors function extracellularly whereas peptide-based secretory/synaptic vesicle inhibitors need to act intracellularly. But even on its own terms the analogy seems inapt,

because there is no effort here (at least, I don't think there is) to inhibit helix bundle formation.

4. I'm unconvinced that the molecular dynamics simulations advance the narrative. It seems obvious based on entropic considerations that the stapled peptide would bind with higher affinity to synaptotagmin (unless of course the staples block the binding site), and a simulation that did not capture that effect would be intrinsically suspect.

5. A nice contribution of this paper is to develop a more authentic reconstitution for airway SNARE/synaptotagmin-mediated fusion, incorporating Munc13-2, Munc18-2, etc., and to establish that SP9 has a potent effect specifically on calcium triggered fusion. The presentation however is more convoluted than it needs to be. Rather than asking the reader (pp. 10-11) to compare multiple panels in supplementary figures to multiple panels in Fig. 3, the authors should devise a concise way to contrast the 'simple' and 'complete' assays and present the key evidence in a single panel in Fig. 3.

Referee #2 (Remarks to the Author):

The authors probed whether intracellular delivery of stapled peptides which inhibit vesicle fusion machinery could block mucin hypersecretion, a cause of airway obstruction. They designed a series of hydrocarbon-stapled peptides based on a fragment of SNAP-25 that participates in the SNARE-Synaptotagmin complex, and used these to inhibit mucin secretion. The authors showed using several complementary approaches that stimulated membrane fusion processes, such as neurotransmitter release or mucin secretion, can be manipulated pharmacologically by their stapled peptides that disrupt the interaction between the fusion proteins and Ca²⁺ sensors. The manuscript demonstrates a wide range of experimental approaches supporting their conclusions and is written clearly.

Major comments:

- 1 Could the high concentration of peptides used be a major block to therapeutic applications? The authors should reflect on the amounts of the material to be applied (milligrams or grams?) and provide an example to support such high peptide usage in humans.
2. The authors should discuss whether they envisage the stapled peptide application to be a one-off or a frequent/on-demand procedure and in these regards discuss pertinent immune/autoimmune aspects of their peptides. They should provide at least one example of a successful application of stapled peptides in humans. Alternatively, it will be helpful to the reader to know current obstacles to translation.

Minor suggestions:

P4 - we tested whether deletion of Syt2 in airway epithelial cells –

this can be improved 'we tested whether deletion of Syt2 specifically in airway epithelial cells'

P4 – "we crossed Syt2F/F mice" – Please explain abbreviations Syt2F/F and Syt2D/D, namely F/F and D/D

P5 'From a therapeutic perspective, peptide-based strategies have been successfully applied

to inhibit virus-host membrane fusion 27–31. However, the molecular mechanisms are quite different since viral membrane fusion is mediated by formation of a six-helix bundle 11 whereas Ca²⁺-triggered membrane fusion is mediated by formation of four-helix SNARE bundles in cooperation with synaptotagmins and other factors'

–The authors should point out here also extracellular versus intracellular drug delivery and that the successful application was presumably in the former case.

Extended Data Figure 2 – related to Figure 2, the panels f and g labels: SP9 and P9 seems to be mixed up.

References: The authors overlooked previous literature describing delineation of the SNAP25-Synaptotagmin interaction interface and an alternative pharmacological approach which disrupts this critical interaction - inositol hexakisphosphate.

J Biol Chem. 2004 Mar 26;279(13):12574-9. doi: 10.1074/jbc.M310710200.

Mol Biol Cell. 2006 Jan;17(1):283-94. doi: 10.1091/mbc.e05-07-0620.

Referee #3 (Remarks to the Author):

1. Summary of the key results: The authors set out to pharmacologically disrupt Ca²⁺- triggered membrane fusion and secretion of mucins from airway epithelial cells. Specifically, they designed a hydrocarbon-stapled peptide (SP9) to Ca²⁺-triggered membrane fusion by interfering with the interface between the SNARE complex and synaptotagmin-1. SP9 suppressed Ca²⁺-triggered fusion and a version conjugated to cell penetrating peptides reduced stimulated mucin secretion in IL-13 stimulated human airway epithelial cells in culture and in IL-13 ex[posed mouse airway epithelial cells in vivo. They propose potential therapeutic benefit for this pharmacologic approach in mucus-associated lung diseases.

2. Originality and significance: There are no available drugs to inhibit mucin secretion in pulmonary medicine. Mucus-related lung disease is a large clinical problem with significant unmet clinical need. The use of a peptide inhibitor strategy to inhibit mucin secretion from airway goblet cells is novel and the significance of this work is the proof of technical success in cell and in rodents. The work to show that the SP9 peptide is stabilized by staples appears carefully done and elegant.

3. Data & methodology: The utility of IL-13 as a cytokine to generate goblet cell metaplasia in cell sheets and in the murine airway epithelium is fully leveraged here and it is a highly relevant and useful model to deploy. The data quality is generally good although the quantitative methods used to measure mucin stores in epithelial cells in the cell cultures and in the mice are investigator lab based methods (and heavily Image J dependent), and not the most rigorous of methods that can be used (stereology approaches for example). So it requires some degree of trust that the lab did this in an unbiased way. The blinding strategy for measurement used in the murine studies is a good feature. In terms of the human airway epithelial cell culture the methods refers to "several donors". More detail should be provided here – there is donor variability in the response to IL-13 stimulation and experienced readers will want to know the number of separate donors used for each of the key experiments.

4. Appropriate use of statistics and treatment of uncertainties: there is a good number of replicates

including in the animal data.

5. Conclusions: robustness, validity, reliability: The paper reads well in terms of how the authors build their argument and provide multiple layers of data to support the utility of the SP9 strategy to prevent mucin secretion from goblet cell mucin granules.

6. Suggested improvements: experiments, data for possible revision: It would help the reader if the authors comment on whether the drug concentrations (micrograms) are in a range that can be feasibly delivered to humans (interspecies allometric scaling).

7. References: appropriate credit to previous work: yes.

8. Clarity and context: lucidity of abstract/summary, appropriateness of abstract, introduction and conclusions; the last sentence of the paper is "Our work may pave the way for the development of novel therapeutics across a wide range of areas" is very vague for such a specifically crafted paper about airway mucin secretion. Consider revising.

Author Rebuttals to Initial Comments:

Referees' comments are in plain font, our responses are in italic font.

Referee #1 (Remarks to the Author):

The authors take, as a point of departure, their 2015 Nature paper reporting the structure of a neuronal SNARE complex bound to the calcium sensor synaptotagmin. This structure revealed that the primary interface between the SNAREs and synaptotagmin was composed mostly – on the SNARE side – of a segment of the N-terminal SNAP-25A SNARE domain. The idea here is to inhibit the SNARE-synaptotagmin interaction by competition with a peptide version (SP9) of this SNAP-25A region. SP9 is ‘stapled’ by introducing two i, i+4 crosslinks, locking it into a helical conformation, and binds synaptotagmin with a Kd of about 30 μ M. The inhibitory function of SP9 is validated in various assays using neuronal proteins and (more importantly for this work) in a rather complete reconstitution system for airway secretory granule fusion. Finally, by attaching a cell-penetrating peptide to SP9, the authors were able to demonstrate its ability to reduce mucin secretion/mucus occlusion in cultured human cells/mouse airway epithelium. Encouragingly, baseline mucin secretion – which might cause toxicity – was unaffected. Given the important role of stimulated mucus hypersecretion in respiratory disease, this evidence of SP9’s therapeutic potential is intriguing.

We thank the reviewer for the positive assessment of our work.

1. Although the “SP9 regions” of SNAP-25A (required for neurotransmitter release) and SNAP-23 (required for mucin secretion) are similar, they differ in at least one residue that contacts synaptotagmin (Fig. S2a). It would therefore seem to be a high priority to test ‘SNAP-23 SP9’ for inhibitory activity in vitro and in vivo.

We thank the reviewer for the suggestion. Although we agree that testing other sequences, including that of [SNAP-23]-SP9, would be of interest, we kindly like to point out that the one direct contact between Syt1/Syt2 and SNAP-25 that the reviewer is referring to and that is different for SNAP-23 (K40Q, see red boxes in Extended Data Figure 2a) may potentially weaken the interaction between [SNAP-23]-SP9 and Syt1/Syt2. We have shown that SP9 binds to both Syt1 and Syt2 with comparable affinity within experimental error (Figure 2f). Moreover, the molecular dynamics simulations suggest dynamical interactions at the N-terminal end of SP9 (Figure 2g). Thus, potentially weakening the interaction by changing the SP9 sequence to the SNAP-23 sequence may not be the best strategy for an inhibitor that disrupts the SNARE-Syt2 interaction by binding to Syt2, and, rather, other sequences may be more successful. However, we believe that such lead optimization would go well beyond the scope of this paper. Also, we kindly would like to point out that the reconstitution, cell-based, and in vivo experiments presented in this work required more than a year to complete, so repeating these experiments with other peptide sequences would practically not be feasible within a reasonable time frame. We have added the point about the K40Q substitution to the main text.

2. A surprising and somewhat concerning aspect of this work is that SP9 oligomerizes as judged by Fig. S3. Fig. S3 also suggests a correlation between oligomerization and efficacy. Have the authors ruled out the possibility that the monomers are covalently crosslinked? Are the oligomers SNARE-bundle-like tetramers? Is oligomerization important for efficacy?

Mass spectrometry and analytical ultracentrifugation should easily answer the first two questions. Answering the third question might be trickier but perhaps the authors have already figured out how to crystallize SP9?

We thank the reviewer for the suggestion. We sincerely apologize for implying that the peptides may form dimers or oligomers based on the size exclusion chromatography (Extended Data Fig. 3). Upon further reflection, it is more correct to state that none of the peptides showed aggregation as assessed by size exclusion chromatography, and that shorter retention times for some of the peptides may be consistent with non-globular extended shapes of these peptides. We attempted to assess the oligomeric state by SEC-MALS and DLS. Unfortunately, the light scattering signal is too weak to obtain a reliable estimate of the molecular mass. We therefore performed single molecule counting experiments of Cy3-labeled SP9 (i.e., SP9-Cy3) (new Extended Data Fig. 3b-d and Methods). These experiments suggest that the peptide is primarily in monomeric form with minor fractions of higher order oligomers. The quality control measures provided by the company (Vivitide, formerly New England Peptide) suggest a highly pure samples without covalent cross links (representative data are in the new Supplementary Figures 2-7).

We attempted to crystallize the complex between Syt1 and SP9 to no avail. This is perhaps not surprising considering the somewhat dynamic nature of the interaction as suggested by the molecular dynamics simulations.

3. The authors write (p. 5) that, “from a therapeutic perspective, peptide-based strategies have been successfully applied to inhibit virus-host membrane fusion. However, the molecular mechanisms are quite different since viral membrane fusion is mediated by formation of a six-helix bundle whereas Ca²⁺-triggered membrane fusion is mediated by formation of four-helix SNARE bundles in cooperation with synaptotagmins and other factors.” In my view, the biggest difference is that peptide-based viral inhibitors function extracellularly whereas peptide-based secretory/synaptic vesicle inhibitors need to act intracellularly. But even on its own terms the analogy seems inapt, because there is no effort here (at least, I don’t think there is) to inhibit helix bundle formation.

We are very grateful for the comment and have revised the text accordingly. Indeed, our work addresses for the first time if a peptide inhibitor strategy can be applied to disrupt Ca²⁺-triggered membrane fusion.

4. I’m unconvinced that the molecular dynamics simulations advance the narrative. It seems obvious based on entropic considerations that the stapled peptide would bind with higher affinity to synaptotagmin (unless of course the staples block the binding site), and a simulation that did not capture that effect would be intrinsically suspect.

We thank the reviewer for the comment. On the other hand, the simulations suggest possible future strategies by stabilizing the N-terminal side of the SP9-Syt1/Syt2 interaction (Figure 2g). We have considerably shortened the discussion of the simulations and removed one of the panels and supplementary video S3 (dissociation event) since they are not relevant for our conclusions.

5. A nice contribution of this paper is to develop a more authentic reconstitution for airway SNARE/synaptotagmin-mediated fusion, incorporating Munc13-2, Munc18-2, etc., and to establish that SP9 has a potent effect specifically on calcium triggered fusion. The

presentation however is more convoluted than it needs to be. Rather than asking the reader (pp. 10-11) to compare multiple panels in supplementary figures to multiple panels in Fig. 3, the authors should devise a concise way to contrast the ‘simple’ and ‘complete’ assays and present the key evidence in a single panel in Fig. 3.

We thank the reviewer for the suggestion. We have added the results of the “simple” reconstitution to Fig. 3 e, g-i for easier comparison.

Referee #2 (Remarks to the Author):

The authors probed whether intracellular delivery of stapled peptides which inhibit vesicle fusion machinery could block mucin hypersecretion, a cause of airway obstruction. They designed a series of hydrocarbon-stapled peptides based on a fragment of SNAP-25 that participates in the SNARE-synaptotagmin complex, and used these to inhibit mucin secretion. The authors showed using several complementary approaches that stimulated membrane fusion processes, such as neurotransmitter release or mucin secretion, can be manipulated pharmacologically by their stapled peptides that disrupt the interaction between the fusion proteins and Ca²⁺ sensors. The manuscript demonstrates a wide range of experimental approaches supporting their conclusions and is written clearly.

We thank the reviewer for the positive assessment of our work.

Major comments:

1 Could the high concentration of peptides used be a major block to therapeutic applications? The authors should reflect on the amounts of the material to be applied (milligrams or grams?) and provide an example to support such high peptide usage in humans.

The high concentration of peptides used in our experiments was mostly driven by the fact that the solution was injected into the airway under pressure to generate a droplet aerosol. The solution would be expected to be in contact with the airway epithelium only briefly, and to mostly end up in the alveolar region. Serial lung sections after introduction of fluorescent labeled peptide confirmed this expectation. These data were not included in the original manuscript because of space limitations, but are now shown in a revised Extended Data Fig. 9a, and briefly elaborated in the last section of Results and in the Discussion. It is not possible to infer from these experiments what concentration of peptide would be required in a conventional therapeutic delivery device, such as a jet nebulizer, metered dose inhaler, or dry powder inhaler. Since delivery of peptides to mice using such devices was prohibitively expensive, we only used the microsyringe with a small volume of solution.

*That being said, we can make inferences about the concentration of PEN-SP9 that would be required in a jet nebulizer based upon the concentration required to achieve a therapeutic effect in cultured airway epithelial cells (Figure 4, 10 μ M, rather than the 200 μ M in mice). Such correlations have been extensively studied by us and by others, and the concentration of drug in the airway surface liquid can be predicted to be approximately 20% that in the nebulizer (Knust J et al., *Anat Rec*, 292:113, 2009; Patton JS et al., *J Aerosol Med Pulm Drug Delivery*, 23:S71, 2010; Durairaj L, *Respir Res*, 5:13, 2004). This assumes an approximate deposition of 0.1% of the nebulized drug per mouse, and an average airway surface liquid volume in mice of 20 μ l. These assumptions were tested in our work with an aerosolized drug we discovered (Evans SE, et al. *Inhaled innate immune ligands to prevent**

pneumonia. Br J Pharmacol, 2011, 163:195, PMID 21250981), and found to be approximately correct (Alfaro VY, et al. Safety, tolerability, and biomarkers of the treatment of mice with aerosolized Toll-like receptor ligands. Front Pharmacol, 2014, 5:8, PMID 24567720; Cleaver JO, et al. Lung epithelial cells are essential effectors of inducible resistance to pneumonia. Mucosal Immunol. 2014 7:78, PMID 23632328). These issues are now briefly addressed in the Discussion.

It should also be noted that we view PEN-SP9 as a lead compound, and not as a drug ready for clinical trials. Optimization to improve cellular uptake, intracellular stability, and potency could result in a drug that could be administered at substantially lower concentration (see below, response to Reviewer 3).

2. The authors should discuss whether they envisage the stapled peptide application to be a one-off or a frequent/on-demand procedure and in these regards discuss pertinent immune/autoimmune aspects of their peptides. They should provide at least one example of a successful application of stapled peptides in humans. Alternatively, it will be helpful to the reader to know current obstacles to translation.

We envisage an optimized drug being used both as a single-dose therapy in an acute exacerbation of airway disease (asthma, COPD, CF) in which airway mucus occlusion is thought to play a part, and as a drug delivered repeatedly in a patient with airway disease in whom control of mucus hypersecretion is difficult to achieve with drugs directed at upstream inflammatory mediators. Therapeutic peptides (MW < 5,000, amino acids < 50) derived from other host proteins have shown a very low rate of immunogenicity in clinical trials, so this is unlikely to be a limitation, though this would need to be examined for SP9 itself in clinical trials. Pseudo-allergic reactions due to mast cell activation can be screened in vitro. A data set collected in 2018 found 484 therapeutic peptides in development, with 12% approved, and 32% in clinical trials (AM Ali, et al. Stapled peptide inhibitors: A new window for drug discovery. Comput Struct Biotechnol J, 2019, 17:263, PMID 30867891). Other useful review articles include the following:

- *Morrison C. Constrained peptides' time to shine? Nat Rev Drug Discov. 2018, 17:531. PMID 30057410.*
- *Lau JL, Dunn MK. Therapeutic peptides: Historical perspectives, current development trends, and future directions. Bioorg Med Chem, 2018, 26:2700. PMID 28720325.*
- *Fosgerau K, Hoffmann T. Peptide therapeutics: current status and future directions. Drug Discov Today, 2015, 20:122. PMID 25450771.*
- *Moiola M, et al. Stapled peptides – a useful improvement for peptide-based drugs. Molecules, 2019, 24:3654. PMID 31658723.*

The most highly cited stapled peptide used in humans has been an inhibitor of the interaction of p53 with MDM2 and MDMX (Chang YS, et al. Stapled α -helical peptide drug development: a potent dual inhibitor of MDM2 and MDMX for p53-dependent cancer therapy. Proc Natl Acad Sci U S A, 2013, 110:E3445. PMID 23946421).

Regarding obstacles to translation of PEN-SP9 into the clinic, we believe the major goals of optimization will be to maximize cellular uptake without causing transepithelial systemic delivery, improving intracellular stability, and increasing potency. This has now been added to the Discussion, along with reference to some of the review articles cited above.

Minor suggestions:

P4 - we tested whether deletion of Syt2 in airway epithelial cells – this can be improved ‘we tested whether deletion of Syt2 specifically in airway epithelial cells’

Done.

P4 – ‘we crossed Syt2F/F mice’ – Please explain abbreviations Syt2F/F and Syt2D/D, namely F/F and D/D

These abbreviations are used both in the Results section on page 5, and in the legends to Figure 1 and Extended Data Figure 1. These abbreviations are widely used in the literature to designate “floxed” and tissue-specific “deletant”, though other abbreviations are also widely used such as “Fl” and the Greek symbol for delta. We now define the abbreviations more completely in Results and the legends to Fig. 1 and Extended Data Fig. 1, and state more fully on page 5, line 6, “the airway epithelial-specific deletant progeny of this cross”. If alternative abbreviations are preferred by the editors, we are happy to change the figures and text.

P5 ‘From a therapeutic perspective, peptide-based strategies have been successfully applied to inhibit virus-host membrane fusion 27–31. However, the molecular mechanisms are quite different since viral membrane fusion is mediated by formation of a six-helix bundle 11 whereas

Ca²⁺-triggered membrane fusion is mediated by formation of four-helix SNARE bundles in cooperation with synaptotagmins and other factors’

–The authors should point out here also extracellular versus intracellular drug delivery and that the successful application was presumably in the former case.

We thank the reviewer for the suggestion which was also made by Reviewer 1 (see above). We have now pointed out this major difference between drug delivery in the text.

Extended Data Figure 2 – related to Figure 2, the panels f and g labels: SP9 and P9 seems to be mixed up.

Thank you. This has been fixed.

References: The authors overlooked previous literature describing delineation of the SNAP25-Synaptotagmin interaction interface and an alternative pharmacological approach which disrupts this critical interaction - inositol hexakisphosphate.

J Biol Chem. 2004 Mar 26;279(13):12574-9. doi: 10.1074/jbc.M310710200.

Mol Biol Cell. 2006 Jan;17(1):283-94. doi: 10.1091/mbc.e05-07-0620.

We thank the reviewer for the suggestion. However, we believe that inositol hexakisphosphate is probably not a good starting point for specific drug development to disrupt stimulated fusion. Moreover, our crystal structures of synaptotagmin/SNARE complexes (Zhou et al. , Nature 2015; NMR 2017), along with recent NMR and fluorescence binding studies by Voleti et al. Elife 9, (2020) suggest that the polybasic region is primarily involved in membrane

interactions and it is unclear if inositol hexakisphosphate will also interfere with the primary interface.

Referee #3 (Remarks to the Author):

1. Summary of the key results: The authors set out to pharmacologically disrupt Ca²⁺-triggered membrane fusion and secretion of mucins from airway epithelial cells. Specifically, they designed a hydrocarbon-stapled peptide (SP9) to Ca²⁺-triggered membrane fusion by interfering with the interface between the SNARE complex and synaptotagmin-1. SP9 suppressed Ca²⁺-triggered fusion and a version conjugated to cell penetrating peptides reduced stimulated mucin secretion in IL-13 stimulated human airway epithelial cells in culture and in IL-13 exposed mouse airway epithelial cells in vivo. They propose potential therapeutic benefit for this pharmacologic approach in mucus-associated lung diseases.

2. Originality and significance: There are no available drugs to inhibit mucin secretion in pulmonary medicine. Mucus-related lung disease is a large clinical problem with significant unmet clinical need. The use of a peptide inhibitor strategy to inhibit mucin secretion from airway goblet cells is novel and the significance of this work is the proof of technical success in cell and in rodents. The work to show that the SP9 peptide is stabilized by staples appears carefully done and elegant.

We thank the reviewer for the positive assessment of our work.

3. Data & methodology: The utility of IL-13 as a cytokine to generate goblet cell metaplasia in cell sheets and in the murine airway epithelium is fully leveraged here and it is a highly relevant and useful model to deploy. The data quality is generally good although the quantitative methods used to measure mucin stores in epithelial cells in the cell cultures and in the mice are investigator lab based methods (and heavily Image J dependent), and not the most rigorous of methods that can be used (stereology approaches for example). So it requires some degree of trust that the lab did this in an unbiased way. The blinding strategy for measurement used in the murine studies is a good feature.

As noted by the reviewer, the Dickey laboratory has used image analysis to study mucin secretion for many years. The laboratory has considered alternative methods, but in order to assess the effect of gene deletion or pharmacologic inhibition on secretory function, it is critical to first accurately measure the intracellular mucin content, and we know of no practical way to do this other than image analysis. To measure baseline secretory function, it is possible to measure total lung mucin content by Western blotting, and to infer a secretion defect from the extent of spontaneous mucin accumulation. This method is highly quantitative, and agrees well with image analysis (Jaramillo AM, et al. Different Munc18 proteins mediate baseline and stimulated airway mucin secretion. JCI Insight. 2019, 4:e124815, PMID 30721150). However, to measure stimulated secretory function by the same method, it would be essential to clear the airways of secreted mucin (i.e., luminal mucus), but this cannot be done because induced secretion of overproduced mucins results in mucus impaction in the airways, as illustrated in Figs. 1 and 5. We have published our methodology extensively (e.g., Piccotti L, Dickey, BF, Evans CM. Assessment of intracellular mucin content in vivo. Methods Mol Biol 2012, 842:279-95. PMID 22259143), and we have further attempted to prevent bias by building a device to stereotypically section the lung at the same point in the axial bronchus between experiments (Jaramillo AM, et al., ibid.), using standardized imaging settings, and blinding investigators both during tissue processing and

image analysis. In unpublished work, we compared our method using ImageJ or ImagePro with results obtained by stereology, and found no significant difference. However, the image analysis was faster and easier for operators in the laboratory, so we continue to use it.

In terms of the human airway epithelial cell culture the methods refers to “several donors”. More detail should be provided here – there is donor variability in the response to IL-13 stimulation and experienced readers will want to know the number of separate donors used for each of the key experiments.

We are very pleased, that the reviewer appreciates the relevance and usefulness of our IL-13 model system. We agree with the reviewer that there is some donor variability within human airway epithelial cell cultures in the response to IL-13 stimulation. All relevant experiments were therefore performed in HAEC ALI cultures generated from several (4 – 8) individual donors. Donor numbers are indicated in the figure legends. Information on the respective donor subjects’ sex, age and smoking status has been added to the methods section.

4. Appropriate use of statistics and treatment of uncertainties: there is a good number of replicates including in the animal data.

5. Conclusions: robustness, validity, reliability: The paper reads well in terms of how the authors build their argument and provide multiple layers of data to support the utility of the SP9 strategy to prevent mucin secretion from goblet cell mucin granules.

We thank the reviewer for the positive assessment of statistics, treatment of uncertainties, robustness, validity, and reliability.

6. Suggested improvements: experiments, data for possible revision: It would help the reader if the authors comment on whether the drug concentrations (micrograms) are in a range that can be feasibly delivered to humans (interspecies allometric scaling).

*As described in the response to Point 1 of Reviewer 2, the high drug concentrations used in the current experiments in mice are thought to mostly reflect short-lived contact with the airway epithelium rather than an intrinsic adverse property of the peptide. When synthesis of the peptide is scaled up during drug development, it will be delivered by a conventional aerosol generation technique, and dose-response experiments will be conducted. In terms of allometric scaling for human drug delivery, it is generally believed that concentrations of drug in a jet nebulizer can be kept constant since the lung size of mammals, and hence the inspiratory volume, scales with size. This is certainly true with aerosol drug delivery to human children and adults using the same device and formulation. Consistent with this, in our own experiments, we found that precisely the same concentration of drug in a jet nebulizer in mice administered for a defined period of time that fell on the upper inflection point for a pharmacodynamic marker (Alfaro VY, et al. Safety, tolerability, and biomarkers of the treatment of mice with aerosolized Toll-like receptor ligands. *Front Pharmacol* 2014, 5:8. PMID 24567720) also fell on the upper inflection point of a biomarker in a human dose-escalation study (NCT 02124278). For clinical use, many types of inhalers are now available, some of which deliver drug to the conducting airways with high efficiency. For all these reasons, we do not think there will be problems with delivery of the candidate drug, and we have addressed the issue in the text as indicated in the response to Reviewer 2.*

7. References: appropriate credit to previous work: yes.

8. Clarity and context: lucidity of abstract/summary, appropriateness of abstract, introduction and conclusions; the last sentence of the paper is “Our work may pave the way for the development of novel therapeutics across a wide range of areas” is very vague for such a specifically crafted paper about airway mucin secretion. Consider revising.

We thank the reviewer for the suggestion. We have strengthened this sentence.

Reviewer Reports on the First Revision:

Referees' comments:

Referee #1 (Remarks to the Author):

I would like to follow up on my first point, i.e., that the manuscript would benefit from using the proper cognate sequence for SP9. Briefly, SP9 was designed based on the sequence of neuronal SNAP25A to inhibit the interaction between SNAP25A and Syt1 (which it does) and neurotransmitter release (which it also does in a cell-free reconstitution). This work is founded on the assumption that the interface between SNAP-25A and Syt1 is conserved in the cognate proteins, SNAP-23 and Syt2, responsible for stimulated mucin secretion. I don't think there is much room for doubt that SP9 based on SNAP-23 would be as good, or probably better, as an inhibitor, and probably more selective as well.

But the authors seem to push back against this idea, and I don't understand their logic. First, they incorrectly (in my view) state that the only divergent residue in the interface is K40Q [lines 130-131]. Because this seemed unlikely based on Extended Data Figure 2B, I double-checked and discovered that both L47I and V48T are also in the interface (based on visual inspection and on analysis of PDB file 5CCG using PISA to identify residues that are more buried in the complex than in the individual proteins). These substitutions should not be swept under the rug.

Second, the authors make the odd assertion that, "although it would be desirable to test other sequences in future work, using the SNAP-23 sequence itself might weaken the interaction with Syt1 or Syt2..." [lines 136-137]. While I certainly agree that it might weaken the interaction with Syt1 (the non-cognate neuronal synaptotagmin), it should surely be neutral or more likely enhance the interaction with Syt2 which, as demonstrated in the preceding section of the manuscript, is the relevant synaptotagmin for stimulated mucin secretion.

The authors state in their rebuttal that it would not be practically feasible to repeat these experiments with other peptide sequences within a reasonable time frame. Perhaps not, and the current manuscript certainly represents a massive amount of effort already. Still, at a minimum, the authors need to be more forthcoming and about the merits and shortcomings of using SNAP-25A-based SP9 in this work.

I am also skeptical of the authors' response to my second point. If SP9 were indeed monomeric, it would be expected to be among the most compact of the peptides studied and should therefore display one of the longest retention times in size exclusion chromatography; instead, it displays one of the shortest (Extended Data Fig. 3a). Evaluating the oligomerization state of immobilized Cy3-labeled SP9 (Extended Data 3b-e) is creative but the gold standard method for evaluating oligomerization state in solution is analytical ultracentrifugation; I think the authors ought to perform this experiment.

The authors' responses to my other points are satisfactory.

Referee #2 (Remarks to the Author):

The authors significantly improved the manuscript showing that stapled peptides that disrupt Ca²⁺-triggered membrane fusion may allow therapeutic modulation of mucin secretory pathways. Conjugation of cell-penetrating peptides to the stapled peptide resulted in delivery into airway epithelium, where it reduced stimulated mucin secretion and attenuated mucus occlusion of mouse airways. The results are original and point to a new therapeutic approach for treatments of mucin disorders. Further work needs to be done on maximizing cellular uptake, improving intracellular stability, and increasing potency.

Referee #3 (Remarks to the Author):

The authors have responded comprehensively to reviewer comments. A minor remaining issue would be that the references in the latter part of the discussion regarding stapled proteins as drugs are not specific to inhaled stapled peptides. Inhaled muco-active drugs that are proteins (eg rhDNAse) are delivered without immune consequence but it remains possible that presentation of peptides directly to the airways and lungs could generate an immune response not seen with non aerosol delivery.

Referee #4 (Remarks to the Author):

The submitted manuscript describes work that goes some way to validating disruption of the neuronal SNARE complex and synaptotagmin-1 protein-protein interaction using stapled peptides as a strategy to inhibit Ca²⁺-triggered membrane fusion. This is original work that builds on the authors previous research in this area. Importantly the work indicates for the first time that targeting this protein-protein interaction has therapeutic value and thus is of significant value to the field. The manuscript is generally well written and the methods and data valid.

Characterisation data is provided for the lead peptide SP9 and the analogue peptides described in figure 4, however no characterisation data has been provided for the initial peptides described in Figure 2. Presumably this is because the peptides used in this study were sourced from two different institutions, namely 'Chemistry Core at the MD Anderson Cancer Center' and 'Vivitide'. i,i+7 stapled peptides are not trivial to synthesise because both cis and trans isomers of the alkene are produced in the ring closing metathesis reaction and must be separated by HPLC and characterised as cis or trans by NMR. As such, LCMS and NMR characterisation data should be provided for those compounds.

It is not entirely clear how peptide stock concentrations were determined. Method line 621 states that peptide powder was dissolved to give a 1 mM stock solution and then diluted. This does not take into account peptide content. Stock solution concentration should be determined using UV, or NMR if no chromophore is present in the peptide.

Other minor corrections that would improve the manuscript include:

- 1) Supplementary videos S1 and S2 do not help with making sense of the binding interactions as the side-chains are not present on the structure. The key interactions should be included.

- 2) Figure 2b provides a cartoon schematic diagram for stapled peptide synthesis. Although the reaction is catalysed by Grubbs 1st generation catalyst the reaction is more commonly described as a ring closing metathesis reaction and thus should be described as such. For example, line 129 - 'generate hydrocarbon-stapled peptides by a Grubbs catalyst to interfere with the primary interface', suggest change to 'generate hydrocarbon-stapled peptides by a ring closing metathesis reaction using Grubbs catalyst to interfere with the primary interface'.

- 3) 'Sn or Rn indicates S or R stereochemistry' R and S should be in italics.

- 4) line 140 – ' α,α -distributed', should read ' α,α -disubstituted', see else where, line 600 etc.

- 5) Line 598 – 'Fmoc chemistry' should read 'Fmoc chemistry'

- 6) line 142 – 'i and i+4', i needs to be italics (see throughout).

- 7) Line 690 – 'restrain the S5 CE-CE double bond in the cis conformation, the S5 aliphatic chains in the trans conformation', This should read *i,i+4* stapled peptides in the cis configuration, *i,i+7* stapled peptides in the trans configuration?

- 8) The experimental methods provided for peptide synthesis in the methods section are not detailed enough to facilitate replication of the work. For example, no details are provided on the deprotection, coupling or ring closing metathesis reactions, or RP-HPLC purification methods. These should be included.

Author Rebuttals to First Revision:

Referees' comments are in plain font, our responses are in italic font.

Referee #1 (Remarks to the Author):

I would like to follow up on my first point, i.e., that the manuscript would benefit from using the proper cognate sequence for SP9. Briefly, SP9 was designed based on the sequence of neuronal SNAP25A to inhibit the interaction between SNAP25A and Syt1 (which it does) and neurotransmitter release (which it also does in a cell-free reconstitution). This work is founded on the assumption that the interface between SNAP-25A and Syt1 is conserved in the cognate proteins, SNAP-23 and Syt2, responsible for stimulated mucin secretion. I don't think there is much room for doubt that SP9 based on SNAP-23 would be as good, or probably better, as an inhibitor, and probably more selective as well.

But the authors seem to push back against this idea, and I don't understand their logic. First, they incorrectly (in my view) state that the only divergent residue in the interface is K40Q [lines 130-131]. Because this seemed unlikely based on Extended Data Figure 2B, I double-checked and discovered that both L47I and V48T are also in the interface (based on visual inspection and on analysis of PDB file 5CCG using PISA to identify residues that are more buried in the complex than in the individual proteins). These substitutions should not be swept under the rug.

We thank the reviewer for the comment and sincerely apologize for the implying that K40Q is the only divergent residue involved in the neuronal primary interface. Rather, we were referring to divergent residues that form salt bridges or hydrogen bonds in the primary interface. In the interest of completeness, we have revised the text and included L47I and V48T as potentially affecting the interaction between the stapled peptide and Syt1 C2B, clarified the meaning of the red boxes in the caption of Extended Data Figure 2a, and added orange boxes to indicate hydrophobic interactions.

Second, the authors make the odd assertion that, "although it would be desirable to test other sequences in future work, using the SNAP-23 sequence itself might weaken the interaction with Syt1 or Syt2..." [lines 136-137]. While I certainly agree that it might weaken the interaction with Syt1 (the non-cognate neuronal synaptotagmin), it should surely be neutral or more likely enhance the interaction with Syt2 which, as demonstrated in the preceding section of the manuscript, is the relevant synaptotagmin for stimulated mucin secretion.

We thank the reviewer for the suggestion. We agree that it is possible that the divergent residues may affect the interaction with Syt1 or Sty2 C2B. We revised the text accordingly. We would like to kindly point out, however, that the interface is conserved on the synaptotagmin side, in other words, there appear to be no complementary substitutions C2B Syt1 → C2B Syt2 that might mirror the SNAP-25 → SNAP23 substitutions.

The authors state in their rebuttal that it would not be practically feasible to repeat these experiments with other peptide sequences within a reasonable time frame. Perhaps not, and the current manuscript certainly represents a massive amount of effort already. Still, at a minimum, the authors need to be more forthcoming and about the merits and shortcomings of using SNAP-25A-based SP9 in this work.

We thank for the reviewer for the comment. We believe that the revised text (page 6) addresses the concern:

“It would be desirable to test other sequences and strategies in future work, including the SNAP-23 sequence itself that might strengthen (or weaken) the interaction with Syt1 or Syt2 considering the K40Q, L47I, and V48T substitutions (Extended Data Fig. 2a).”

I am also skeptical of the authors’ response to my second point. If SP9 were indeed monomeric, it would be expected to be among the most compact of the peptides studied and should therefore display one of the longest retention times in size exclusion chromatography; instead, it displays one of the shortest (Extended Data Fig. 3a).

The precise origin of the shorter retention times of the stapled peptides is uncertain. It could be related to extended shape of the stapled peptide helix or possible interactions with the resin of the size exclusion column. However, we would like to kindly point out that none of our conclusions depend on this particular characterization.

Evaluating the oligomerization state of immobilized Cy3-labeled SP9 (Extended Data 3b-e) is creative but the gold standard method for evaluating oligomerization state in solution is analytical ultracentrifugation; I think the authors ought to perform this experiment.

We thank the reviewer for the comment. However, regretfully, there is no analytical ultracentrifuge at Stanford University, and there are no facilities accessible to us that have an analytical ultracentrifuge and the expertise to perform such experiments. This was the reason that we had attempted SEC-MALS as mentioned in our previous response (to no avail), and then developed the single molecule counting experiment to observe the oligomeric state(s) of SP9. The single molecule counting experiment (Extended Data Fig. 3b-e) suggest that SP9 is primarily monomeric, and that it is unlikely a tightly associated oligomer. Moreover, we kindly would like to point out that our in vitro (reconstitution), cell-based, and mouse experiments and several controls show that the effect of SP9 is specific. Finally, the conclusions of this manuscript do not depend on the precise distribution of the oligomeric states of SP9.

The authors’ responses to my other points are satisfactory.

We thank the reviewer for the comment.

Referee #2 (Remarks to the Author):

The authors significantly improved the manuscript showing that stapled peptides that disrupt Ca²⁺-triggered membrane fusion may allow therapeutic modulation of mucin secretory pathways. Conjugation of cell-penetrating peptides to the stapled peptide resulted in delivery into airway epithelium, where it reduced stimulated mucin secretion and attenuated mucus occlusion of mouse airways. The results are original and point to a new therapeutic approach for treatments of mucin disorders. Further work needs to be done on maximizing cellular uptake, improving intracellular stability, and increasing potency.

We thank the reviewer for the positive assessment of our work. We agree that future follow-up work would be desirable to improve the stapled peptide, but we feel that this is outside the scope of this work.

Referee #3 (Remarks to the Author):

The authors have responded comprehensively to reviewer comments. A minor remaining issues would be that the references in the latter part of the discussion regarding stapled proteins as drugs are not specific to inhaled stapled peptides. Inhaled muco-active drugs that are proteins (eg rhDNase) are delivered without immune consequence but it remains possible that presentation of peptides directly to the airways and lungs could generate an immune response not seen with non aerosol delivery.

We thank the reviewer for the comment. As regards a possible immune response, we have revised the text in the concluding paragraph:

“Therapeutic peptides derived from other host proteins have shown a very low rate of immunogenicity in clinical trials when administered systemically^{33,53,54}, as has rhDNase administered repeatedly by aerosol⁵⁵, so this is unlikely to be a limitation of chronic therapy though it would need to be examined.”

We added reference (55) that includes the studies involving rhDNase.

Referee #4 (Remarks to the Author):

The submitted manuscript describes work that goes some way to validating disruption of the neuronal SNARE complex and synaptotagmin-1 protein-protein interaction using stapled peptides as a strategy to inhibit Ca²⁺-triggered membrane fusion. This is original work that builds on the authors previous research in this area. Importantly the work indicates for the first time that targeting this protein-protein interaction has therapeutic value and thus is of significant value to the field. The manuscript is generally well written and the methods and data valid.

Characterisation data is provided for the lead peptide SP9 and the analogue peptides described in figure 4, however no characterisation data has been provided for the initial peptides described in Figure 2. Presumably this is because the peptides used in this study were sourced from two different institutions, namely „Chemistry Core at the MD Anderson Cancer Center“ and „Vivitide“. $i,i+7$ stapled peptides are not trivial to synthesise because both cis and trans isomers of the alkene are produced in the ring closing metathesis reaction and must be separated by HPLC and characterised as cis or trans by NMR. As such, LCMS and NMR characterisation data should be provided for those compounds.

We thank the reviewer for the positive assessment of our work. As discussed with the Nature Editors, we have removed the studies with the early peptides since the quality control data are regrettably not accessible anymore (the Chemistry Core at MD Anderson Cancer Center has been dissolved), and these initial studies are not essential for the conclusions of this work.

It is not entirely clear how peptide stock concentrations were determined. Method line 621 states that peptide powder was dissolved to give a 1 mM stock solution and then diluted. This does not take into account peptide content. Stock solution concentration should be determined using UV, or NMR if no chromophore is present in the peptide.

We thank the reviewer for the comment and added detailed information about the stock concentrations in the Methods.

Other minor corrections that would improve the manuscript include:

1) Supplementary videos S1 and S2 do not help with making sense of the binding interactions as the side-chains are not present on the structure. The key interactions should be included.

We thank the reviewer for the comment. We have now included side-chains in the videos. We have also provided pdb files as source data that correspond to the structures shown in these videos.

2) Figure 2b provides a cartoon schematic diagram for stapled peptide synthesis. Although the reaction is catalysed by Grubbs 1st generation catalyst the reaction is more commonly described as a ring closing metathesis reaction and thus should be described as such. For example, line 129 - „generate hydrocarbon-stapled peptides by a Grubbs catalyst to interfere with the primary interface“, suggest change to „generate hydrocarbon-stapled peptides by a ring closing metathesis reaction using Grubbs catalyst to interfere with the primary interface“.

Thank you for the comment. We have changed the text accordingly.

3) „Sn or Rn indicates S or R stereochemistry“ R and S should be in italics.

Done.

4) line 140 – „ α,α -distributed“, should read „ α,α -disubstituted“, see else where, line 600 etc.

Done.

5) Line 598 – „FMOC chemistry“ should read „Fmoc chemistry“

Done.

6) line 142 – „i and i+4“, i needs to be italics (see throughout).

Done.

7) Line 690 – „restrain the S5 CE-CE double bond in the cis conformation, the S5 aliphatic chains in the trans conformation“, This should read *i,i+4 stapled peptides in the cis configuration, i,i+7 stapled peptides in the trans configuration?*

Since the initial peptides have been removed from the manuscript, there are no examples of i,i+7 peptides anymore and we have revised the text accordingly.

8) The experimental methods provided for peptide synthesis in the methods section are not detailed enough to facilitate replication of the work. For example, no details are provided on the deprotection, coupling or ring closing metathesis reactions, or RP-HPLC purification methods. These should be included.

We have included the methods for peptide synthesis in the Methods.

Reviewer Reports on the Second Revision:

Referees' comments:

Referee #1 (Remarks to the Author):

Overall, I am satisfied by the authors' responses. Two comments:

1. The authors now acknowledge that there are 4 residues in the Syt1/SNAP-25 interface that differ from those in the Syt2/SNAP-23 interface. Three of these residues — Syt1 V292, SNAP-25 L47, and SNAP-25 V48 — are in fairly close proximity. Therefore, I don't entirely agree with the statement in the rebuttal letter that "the interface is conserved on the synaptotagmin side, in other words, there appear to be no complementary substitutions C2B Syt1 -> C2B Syt2 that might mirror the SNAP-25 -> SNAP23 substitutions." But since this statement does not appear in the text of the manuscript, and all four mutations are now mentioned explicitly, I am ready to let the matter rest.

2. As for the unavailability of an analytical ultracentrifuge at Stanford, I can only note in passing that poor Arnold Beckmann would be turning in his grave to think of this unfortunate situation!

Referee #4 (Remarks to the Author):

The authors have responded to all of my comments and made appropriate corrections. I agree that removing the initial peptide discovery section provides a more focused and generally accessible manuscript. Congratulations on some very nice science.

Author Rebuttals to Second Revision:

Referees' comments:

Referee #1 (Remarks to the Author):

Overall, I am satisfied by the authors' responses. Two comments:

1. The authors now acknowledge that there are 4 residues in the Syt1/SNAP-25 interface that differ from those in the Syt2/SNAP-23 interface. Three of these residues — Syt1 V292, SNAP-25 L47, and SNAP-25 V48 — are in fairly close proximity. Therefore, I don't entirely agree with the statement in the rebuttal letter that "the interface is conserved on the synaptotagmin side, in other words, there appear to be no complementary substitutions C2B Syt1 -> C2B Syt2 that might mirror the SNAP-25 -> SNAP23 substitutions." But since this statement does not appear in the text of the manuscript, and all four mutations are now mentioned explicitly, I am ready to let the matter rest.

2. As for the unavailability of an analytical ultracentrifuge at Stanford, I can only note in passing that poor Arnold Beckmann would be turning in his grave to think of this unfortunate situation!

We thank the reviewer for the positive assessment. We note that no changes were requested in the manuscript.

Referee #4 (Remarks to the Author):

The authors have responded to all of my comments and made appropriate corrections. I agree that removing the initial peptide discovery section provides a more focused and generally accessible manuscript. Congratulations on some very nice science.

We thank the reviewer for the positive assessment of our work.